# PREFERENCE DIFFUSION FOR RECOMMENDATION

**Shuo Liu**[1,2]   **An Zhang**[2]*   **Guoqing Hu**[3]   **Hong Qian**[1]   **Tat-Seng Chua**[2]
[1]East China Normal University, China
[2]National University of Singapore, Singapore
[3]University of Science and Technology of China, China
`shuoliu@stu.ecnu.edu.cn, anzhang@u.nus.edu, hl15671953077@`
`ustc.mail.edu.cn, hqian@cs.ecnu.edu.cn, dcscts@nus.edu.sg`

## ABSTRACT

Recommender systems aim to predict personalized item rankings by modeling user preference distributions derived from historical behavior data. While diffusion models (DMs) have recently gained attention for their ability to model complex distributions, current DM-based recommenders typically rely on traditional objectives such as mean squared error (MSE) or standard recommendation objectives. These approaches are either suboptimal for personalized ranking tasks or fail to exploit the full generative potential of DMs. To address these limitations, we propose **Prefer-Diff**, an optimization objective tailored for DM-based recommenders. PreferDiff reformulates the traditional Bayesian Personalized Ranking (BPR) objective into a log-likelihood generative framework, enabling it to effectively capture user preferences by integrating multiple negative samples. To handle the intractability, we employ variational inference, minimizing the variational upper bound. Furthermore, we replace MSE with cosine error to improve alignment with recommendation tasks, and we balance generative learning and preference modeling to enhance the training stability of DMs. PreferDiff devises three appealing properties. First, it is the first personalized ranking loss designed specifically for DM-based recommenders. Second, it improves ranking performance and accelerates convergence by effectively addressing hard negatives. Third, we establish its theoretical connection to Direct Preference Optimization (DPO), demonstrating its potential to align user preferences within a generative modeling framework. Extensive experiments across six benchmarks validate PreferDiff's superior recommendation performance. Our codes are available at `https://github.com/lswhim/PreferDiff`.

## 1 INTRODUCTION

The recommender system endeavors to model the user preference distribution based on their historical behaviour data (He & McAuley, 2016; Wang et al., 2019; Rendle, 2022) and predict personalized item rankings. Recently, diffusion models (DMs) (Sohl-Dickstein et al., 2015; Ho et al., 2020; Yang et al., 2024) have gained considerable attention for their robust capacity to model complex data distributions and versatility across a wide range of applications, encompassing diverse input styles: texts (Li et al., 2022; Lovelace et al., 2023), images (Dhariwal & Nichol, 2021; Ho & Salimans, 2022) and videos (Ho et al., 2022a;b). As a result, there has been growing interest in employing DMs as recommenders in recommender systems.

These DM-based recommenders utilize the diffusion-then-denoising process on the user's historical interaction data to uncover the potential target item, typically following one of three approaches: modeling the distribution of the next item (Yang et al., 2023b; Wang et al., 2024b; Li et al., 2024), capturing the user preference distribution (Wang et al., 2023b; Zhao et al., 2024; Hou et al., 2024a; Zhu et al., 2024), or focusing on the distribution of time intervals for predicting the user's next action (Ma et al., 2024a). However, prevalent DM-based recommenders often routinely rely on standard generative loss functions, such as mean squared error (MSE), or blindly adapt established recommendation objectives, such as Bayesian personalized ranking (BPR) (Rendle et al., 2009) and (binary) cross entropy (Sun et al., 2019) without any modification. Despite their empirical

---

*An Zhang is the corresponding author.

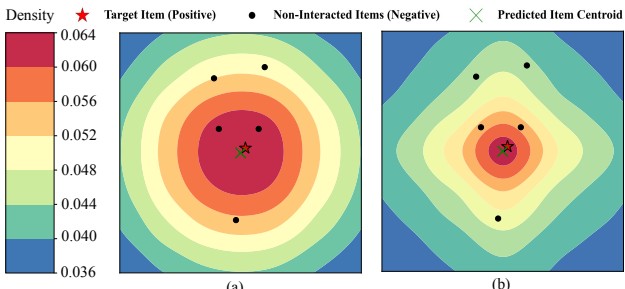

Figure 1: Illustration of user preference distributions modeled by DM-based recommenders. (a) Neglecting the negative item distribution leads to predicted items potentially being closer to negative items. (b) Incorporating the negative sampling enhances the understanding of user preferences.

success, two key limitations in their training objectives have been identified, which may hinder further advancements in this field:

• **DM-based recommenders inheriting generative objective functions (Yang et al., 2023b) lack a comprehensive understanding of user preference sequences.** They model user behavior by considering only the items users have interacted with, neglecting the critical role of negative items in recommendations (Chen et al., 2023a; 2024; Zhang et al., 2024). As illustrated in Figure 1(a), although the predicted item centroid is close to the positive item, the sampling process of the DMs may tend to obtain the final predicted item embedding in high-density regions (red in Figure 1(a)(b)). This can result in the predicted item embedding being too close to negative items, thereby affecting the personalized ranking performance. Enabling DMs to understand what users may dislike can help alleviate this issue, as illustrated in Figure 1(b).

• **DM-based recommenders simply employ standard recommendation training objectives, hindering their generative ability.** This type of DM-based recommenders treats DMs primarily as noise-resistant models that focus on ranking or classification rather than on generation. While this approach can mitigate the impact of noisy interactions inherent in recommender systems (Wang et al., 2023b; Li et al., 2024), it may not fully exploit the generative and generalization capabilities of DMs, whose primary objective is to maximize the data log-likelihood.

To better understand and redesign a diffusion optimization objective that is specially tailored to model user preference distributions for personalized ranking, we aim to simultaneously encode user dislikes and enhance the generative capability of the ranking objective. Our approach involves extending the classical and widely-adopted BPR objective to incorporate multiple negative samples, while also clarifying its connection to likelihood-based generative models, exemplified by DMs (Yang et al., 2024). BPR only seeks to maximize the rating margin between positive and negative items, which may result in high score negative ratings. In contrast, our core idea focuses on modeling user preference distributions, where the distribution of positive items diverges from that of negative items, conditioned on the user's personalized interaction history.

To this end, we propose a training objective specifically designed for DM-based recommenders, called **PreferDiff**, which effectively integrates negative samples to better capture user preference distributions. Specifically, by applying softmax normalization, we transform BPR from a rating ranking into log-likelihood ranking, leading to the formulation of $\mathcal{L}_{\text{BPR-Diff}}$. However, since DMs are latent variable models (Ho et al., 2020), direct optimization through gradient descent is intractable. To address this intractability, we derive a variational upper bound for $\mathcal{L}_{\text{BPR-Diff}}$ using variational inference, which serves as a surrogate optimization target. Furthermore, we replace the original MSE with cosine error (Hou et al., 2022b), allowing generated items to better align with the similarity calculations in recommendation tasks and controlling the scale of embeddings (Chen et al., 2023c). Additionally, we extend $\mathcal{L}_{\text{BPR-Diff}}$ to incorporate multiple negative samples, enabling the model to inject richer preference information during training while implementing an efficient strategy to prevent redundant denoising steps from excessive negative samples. Finally, we balance generation learning and preference learning to achieve a trade-off that enhances both training stability and model performance, culminating in the final objective function, $\mathcal{L}_{\text{PreferDiff}}$.

Benefiting from a comprehensive understanding of user preference distributions, **PreferDiff** has three appealing properties: First, PreferDiff is the first personalized ranking loss specifically designed for DM-based recommenders, incorporating multiple negatives to model the user preference distributions. Second, gradient analysis reveals that PreferDiff handles hard negatives by assigning higher gradient weights to item sequences, where DM incorrectly assigns a higher likelihood to negative items than positive ones (Chen et al., 2022; Fan et al., 2023; Zhang et al., 2023))(cf. Section 3.2). This not only improves recommendation performance but also accelerates training (cf. Section 4.1). Third, from a preference learning perspective, we find that PreferDiff is connected to Direct Preference Optimization (Rafailov et al., 2023) under certain conditions, indicating its potential to align user preferences through generative modeling in diffusion-based recommenders (cf. Section 3.2).

We evaluate the effectiveness of PreferDiff through extensive experiments and comparisons with baseline models using six widely adopted public benchmarks (cf. Section 4.1). Furthermore, by simply replacing item ID embeddings with item semantic embeddings via advanced text-embedding modules, PreferDiff shows strong generalization capabilities for sequential recommendations across untrained domains and platforms, without introducing additional components (cf. Section 4.2).

## 2 PRELIMINARY

In this section, we begin by formally introducing the task of sequential recommendation and then introduce the foundations of DM-based recommenders who model the next-item distribution.

**Sequential Recommendation.** Suppose each user has a historical interaction sequence $\{i_1, i_2, \ldots, i_{n-1}\}$, representing their interactions in chronological order and $i_n$ is the next target item. For each sequence, we randomly sample negative items from batch or candidate set result in $\mathcal{H} = \{i_v\}_{v=1}^{|\mathcal{H}|}$. Moreover, each item $i$ is associated with a unique item ID or additional descriptive information (e.g., title, brand and category). Via ID-embedding or text-embedding module, items can be transformed into its corresponding vectors $\mathbf{e} \in \mathbb{R}^{1 \times d}$. Therefore, the historical interaction sequence and negative items' set can be transformed to $\mathbf{c} = \{\mathbf{e}_1, \mathbf{e}_2, \ldots, \mathbf{e}_{n-1}\}$ and $\mathcal{H} = \{\mathbf{e}_v\}_{v=1}^{V}$. The goal of sequential recommendation is to give the personalized ranking on the whole candidate set, namely, predict the next item $i_n$ user may prefer given the sequence $\mathbf{c}$ and negative items' set $\mathcal{H}$.

**Diffusion models for Sequential Recommendation.** In this section, we introduce the use of guided DMs to model the conditional next-item distribution $p(i_n \mid i_{<n})$, following the DreamRec (Yang et al., 2023b). For clarity, we denote the vector representation of the next item $i_n$ as $\mathbf{e}_0^+$ instead of $\mathbf{e}_n$ and negative items $i_v$ as $\mathbf{e}_0^{-v}$ result in $\mathcal{H} = \{\mathbf{e}_0^{-v}\}_{v=1}^{|\mathcal{H}|}$. The subscript denotes the timesteps in DM, where "0" indicates that no noise has been added, and the superscript represents whether the item is positive or negative, denoted by "+" or "-" respectively in recommendation. Notably, these notations will be used consistently in the subsequent sections.

• **Forward Process.** DMs add Gaussian noise to the positive item embedding $\mathbf{e}_0^+$ with noise scale $\{\alpha_1, \alpha_2, \cdots, \alpha_T\}$ over the pre-defined timesteps $T$, namely, $q(\mathbf{e}_t^+ \mid \mathbf{e}_0^+) = \mathcal{N}(\sqrt{\bar{\alpha}_t}\mathbf{e}_0^+, (1 - \bar{\alpha}_t)\mathbf{I})$. If $T \to +\infty$, $e_T^+$ asymptotically converges to the standard Gaussian distribution. $q(\mathbf{e}_t^+ \mid \mathbf{e}_0^+)$ can be easily derived through applications of the reparameterization trick (Kingma & Welling, 2014).

• **Reverse Process.** The reverse process aims to recover the target item embedding $\mathbf{e}_0^+$ from the standard Gaussian distribution through the denoising process with the personalized guidance $\mathbf{c}$. Concretely, following the classical DMs' paradigm introduced in DDPM (Ho et al., 2020), we choose the simple objective which minimizes the KL divergence between the true denoising transition $q(\mathbf{e}_{t-1}^+ \mid \mathbf{e}_t^+, \mathbf{e}_0^+)$ and the intractable denoising transition $p_\theta(\mathbf{e}_{t-1}^+ \mid \mathbf{e}_t^+, \mathbf{c})$. Leveraging the favorable properties of the Gaussian distribution, we can derive the following closed-form objective:

$$\mathcal{L}_{\text{Simple}} = \mathbb{E}_{(\mathbf{e}_0^+, \mathbf{c}, t)} \left[ \left\| \mathcal{F}_\theta(\mathbf{e}_t^+, t, \mathcal{M}(\mathbf{c})) - \mathbf{e}_0^+ \right\|_2^2 \right], \tag{1}$$

where $\mathbf{e}_0^+, \mathbf{c}$ come from the training data. $t \sim \mathcal{U}(1, T)$ is the sampled timestep. $\mathcal{M}(\cdot)$ denotes the arbitrary sequence encoder utilized in sequential recommendation (e.g., GRU (Hidasi et al., 2016), Transformer (Kang & McAuley, 2018), Bert (Sun et al., 2019)). $\mathcal{F}_\theta(\cdot)$ serves as denoising network which is commonly parameterized by a simple MLP and $\theta$ denotes the trainable parameters. Classifier-free guidance scheme (Ho & Salimans, 2022) can be utilized here to replace $\mathcal{M}(\mathbf{c})$ with

dummy token $\Phi$ with probability $p_u$ to achieve the training of unconditional DM. Furthermore, some works (Li et al., 2024) utilize the recommendation objective (binary) cross entropy instead of MSE.

• **Inference and Recommend.** During the inference stage, we first derive the representation of a given user's historical sequence, denoted as $\mathcal{M}(\mathbf{c})$. Starting from pure Gaussian noise, we then utilize the denoising network $\mathcal{F}_\theta(\cdot)$ to iteratively generate latent embeddings, following arbitrary samplers (e.g., DDIM (Song et al., 2021a)) in DMs, until the inferred next item embedding $\hat{\mathbf{e}}_0$ is obtained. More details can be found in Algorithm 2 and Appendix B. Finally, we recommend the top-K items with the highest dot product between $\hat{\mathbf{e}}_0$ and the item embeddings in the candidate set.

## 3 METHODOLOGY: THE PROPOSED PREFERDIFF

In this section, we introduce **PreferDiff**, a novel loss for DM-based recommenders that can instill preference information. First, we extend the classical BPR loss to a probabilistic one, defining a new loss $\mathcal{L}_{\text{BPR-Diff}}$. To address the inherent intractability, we derive a variational upper bound $\mathcal{L}_{\text{Upper}}$ for $\mathcal{L}_{\text{BPR-Diff}}$ and optimize this bound instead. Furthermore, we explore the incorporation of multiple negative samples and propose an efficient strategy by lowering the likelihood of the negative samples' centroid, which avoids multiple denoising steps. Lastly, we make a trade-off between learning generation and learning preference to ensure training stability, resulting in the final loss $\mathcal{L}_{\text{PreferDiff}}$.

### 3.1 CONNECT DIFFUSION MODELS WITH BAYESIAN PERSONALIZED RANKING

In this subsection, we explore the integration of DMs with the classical BPR loss (Rendle et al., 2009), which has been proven to be highly effective in real-world industrial recommendation scenarios. As BPR is designed to optimize personalized ranking by modeling user preferences in a pairwise fashion, it has been extensively applied in contemporary recommendation researches (Kang & McAuley, 2018; He et al., 2020). It can be formulated as

$$\mathcal{L}_{\text{BPR}} = -\mathbb{E}_{(\mathbf{e}_0^+, \mathbf{e}_0^-, \mathbf{c})} \left[ \log \sigma \left( f_\theta(\mathbf{e}_0^+ \mid \mathbf{c}) - f_\theta(\mathbf{e}_0^- \mid \mathbf{c}) \right) \right] , \tag{2}$$

where $\mathbf{e}_0^+, \mathbf{e}_0^-$ represents the positive item and one negative item in $\mathcal{H}$, we omit $v$ for brevity. $\mathbf{c}$ represents the historical item sequences. $\sigma$ is the Sigmoid function. $f_\theta(\mathbf{e}_0 \mid \mathbf{c})$ is the predicted rating of item $\mathbf{e}_0$ conditioned on the historical item sequence $\mathbf{c}$. As DMs are part of the family of likelihood-based generative models (Yang et al., 2024) and are employed here to maximize the log-likelihood of the next item distribution $\log p_\theta(\mathbf{e}_0^+ \mid \mathbf{c})$, it is clear that equation 2 does not meet this need. Therefore, we put forward to change the rating to the probability distribution.

**From Rating to Probability Distribution.** Here, we define the probability distribution of the next-item $\mathbf{e}_0$ given historical item sequences $\mathbf{c}$ via a softmax over the arbitrarily flexible, parameterizable, rating function $f_\theta(\cdot)$. It can be formulated as $p_\theta(\mathbf{e}_0 \mid \mathbf{c}) = \frac{\exp(f_\theta(\mathbf{e}_0 \mid \mathbf{c}))}{Z_\theta}$, where $Z_\theta$ is normalizing constant (a.k.a, partition function), defined as $\int \exp(f_\theta(\mathbf{e} \mid \mathbf{c})) \, d\mathbf{e}$. Then, by substituting it into equation 2, we obtain the following result, which we refer to as $\mathcal{L}_{\text{BPR-Diff}}$, as we utilize the DMs to model that distribution. The detailed derivation is provided in Appendix C.1.

$$\mathcal{L}_{\text{BPR-Diff}}(\theta) = -\mathbb{E}_{(\mathbf{e}_0^+, \mathbf{e}_0^-, \mathbf{c})} \left[ \log \sigma \left( \log p_\theta(\mathbf{e}_0^+ \mid \mathbf{c}) - \log p_\theta(\mathbf{e}_0^- \mid \mathbf{c}) \right) \right] . \tag{3}$$

Intuitively, $\mathcal{L}_{\text{BPR-Diff}}$ seeks to widen the gap between the log-probability distributions of positive and negative items given $\mathbf{c}$. However, the challenge is that equation 3 is intractable due to the need to marginalize over all possible diffusion paths as DMs are latent variable models. Therefore, like previous work (Sohl-Dickstein et al., 2015; Ho et al., 2020), we propose to minimize the $\mathcal{L}_{\text{BPR-Diff}}$ via variational inference through minimizing the derived variational upper bound.

**Minimize $\mathcal{L}_{\text{BPR-Diff}}$ through Variational Upper Bound.** Therefore, like previous work (Sohl-Dickstein et al., 2015; Ho et al., 2020), we introduce latent variables $(\mathbf{e}_1, \ldots, \mathbf{e}_T)$, resulting in $p_\theta(\mathbf{e}_0 \mid \mathbf{c}) = \int p_\theta(\mathbf{e}_{0:T} \mid \mathbf{c}) \, d\mathbf{e}_{1:T}$. Then, we substitute $p_\theta(\mathbf{e}_{1:T} \mid \mathbf{e}_0)$ with $q(\mathbf{e}_{1:T} \mid \mathbf{e}_0)$ which is typically modeled as a Gaussian distribution with predefined mean and variance at each timestep, due to the intractability of directly sampling from the former distribution. The objective can be expressed as follows:

$$\mathcal{L}_{\text{BPR-Diff}}(\theta) = -\mathbb{E}_{(\mathbf{e}_0^+, \mathbf{e}_0^-, \mathbf{c})} \left[ \log \sigma \left( \log \mathbb{E}_{q(\mathbf{e}_{1:T}^+ \mid \mathbf{e}_0^+)} \frac{p_\theta(\mathbf{e}_{0:T}^+ \mid \mathbf{c})}{q(\mathbf{e}_{1:T}^+ \mid \mathbf{e}_0^+)} - \log \mathbb{E}_{q(\mathbf{e}_{1:T}^- \mid \mathbf{e}_0^-)} \frac{p_\theta(\mathbf{e}_{0:T}^- \mid \mathbf{c})}{q(\mathbf{e}_{1:T}^- \mid \mathbf{e}_0^-)} \right) \right] . \tag{4}$$

By applying Jensen's inequality and leveraging the convexity of the logarithmic function, we can move the expectation operator outside. Consequently, after further mathematical derivations, we can establish an upper bound for $\mathcal{L}_{\text{BPR-Diff}}$ as equation 5.

$$\mathcal{L}_{\text{BPR-Diff}}(\theta) \leq -\mathbb{E}_{(\mathbf{e}_0^+, \mathbf{e}_0^-, \mathbf{c})} \mathbb{E}_{q(\mathbf{e}_{1:T}^+ | \mathbf{e}_0^+), q(\mathbf{e}_{1:T}^- | \mathbf{e}_0^-)} \left[ \log \sigma (\log \frac{p_\theta(\mathbf{e}_{0:T}^+ \mid \mathbf{c})}{q(\mathbf{e}_{1:T}^+ \mid \mathbf{e}_0^+)} - \log \frac{p_\theta(\mathbf{e}_{0:T}^- \mid \mathbf{c})}{q(\mathbf{e}_{1:T}^- \mid \mathbf{e}_0^-)}) \right]. \tag{5}$$

Following the derivation of classical DMs (Ho et al., 2020; Song et al., 2021a; Luo, 2022), we can simplify the above equation through algebra, yielding the following result:

$$\begin{aligned} \mathcal{L}_{\text{BPR-Diff}}(\theta) \leq -\mathbb{E}_{(\mathbf{e}_0^+, \mathbf{e}_0^-, \mathbf{c})} \Bigg[ \log \sigma \Bigg( - \Bigg( \sum_{t=1}^{T} \mathbb{E}_{q(\mathbf{e}_t^+ | \mathbf{e}_0^+)} \left[ D_{\text{KL}} \left( q(\mathbf{e}_{t-1}^+ | \mathbf{e}_t^+, \mathbf{e}_0^+) \parallel p_\theta(\mathbf{e}_{t-1}^+ | \mathbf{e}_t^+) \right) \right] \\ - \sum_{t=1}^{T} \mathbb{E}_{q(\mathbf{e}_t^- | \mathbf{e}_0^-)} \left[ D_{\text{KL}} \left( q(\mathbf{e}_{t-1}^- | \mathbf{e}_t^-, \mathbf{e}_0^-) \parallel p_\theta(\mathbf{e}_{t-1}^- | \mathbf{e}_t^-) \right) \right] + C_1 \Bigg) \Bigg) \Bigg], \end{aligned} \tag{6}$$

where $C_1$ is a constantthath is independent of the model parameter $\theta$. As introduced in the Preliminary, by applying Bayes' theorem and leveraging the additivity property of Gaussian distributions, the final trainable objective on stochastic samples over timestep is expressed as follows:

$$\mathcal{L}_{\text{Upper}}(\theta) = -\mathbb{E}_{(\mathbf{e}_0^+, \mathbf{e}_0^-, \mathbf{c}), t \sim U(1, T)} \left[ \log \sigma(-(S(\hat{\mathbf{e}}_0^+, \mathbf{e}_0^+) - S(\hat{\mathbf{e}}_0^-, \mathbf{e}_0^-))) \right]. \tag{7}$$

Here, $\hat{\mathbf{e}}_0^+ = \mathcal{F}_\theta(\mathbf{e}_t^+, t, \mathcal{M}(\mathbf{c})), \hat{\mathbf{e}}_0^- = \mathcal{F}_\theta(\mathbf{e}_t^-, t, \mathcal{M}(\mathbf{c}))$. $S(\cdot)$ denotes the function that quantifies the distance between the prediction and the true next item embedding, typically MSE in previous works. As retrieval during the inference stage is conducted via maximal inner product search for ranking and MSE shows sensitivity to vector norms and dimensionality (Friedman, 1997; Hou et al., 2022b), we propose using cosine error instead. Since $\mathcal{L}_{\text{Upper}}$ serves as an upper bound for $\mathcal{L}_{\text{BPR-Diff}}$, minimizing $\mathcal{L}_{\text{Upper}}$ implicitly minimizes $\mathcal{L}_{\text{BPR-Diff}}$. Intuitively, equation 7 is designed such that, given a user's historical item sequence, the denoising network $\mathcal{F}(\cdot)$ tends to recover the positive item rather than the negative item. A detailed derivation can be found in Appendix C.3.

## 3.2 ANALYSIS OF $\mathcal{L}_{\text{BPR-DIFF}}$

In this subsection, we demonstrate the two properties of $\mathcal{L}_{\text{BPR-Diff}}$ by analyzing the gradient with respect to $\theta$ and connecting it with recent popular direct preference optimization. We also reveal the connection between the rating function and the score function in Appendix equation C.2 which bridges the objective of recommendation with generative modeling in DMs.

**Gradient Analysis.** Here, we analyze the gradients of $\mathcal{L}_{\text{BPR-Diff}}$ to understand their impact on the training process of DMs for sequential recommendation.

$$\frac{\partial \mathcal{L}_{\text{BPR-Diff}}(\theta)}{\partial \theta} = -\mathbb{E}_{(\mathbf{e}_0^+, \mathbf{e}_0^-, \mathbf{c})} [w_\theta (\underbrace{\nabla_\theta \log p_\theta(\mathbf{e}_0^+ \mid \mathbf{c})}_{\text{Increase Likelihood on Positive Item}} - \underbrace{\nabla_\theta \log p_\theta(\mathbf{e}_0^- \mid \mathbf{c})}_{\text{Decrease Likelihood on Negative Item}})], \tag{8}$$

where $w_\theta = 1 - \sigma \left( \log p_\theta(\mathbf{e}_0^+ \mid \mathbf{c}) - \log p_\theta(\mathbf{e}_0^- \mid \mathbf{c}) \right)$ represents the gradient weight. Obviously, if given certain item sequences, the DM incorrectly assigns a higher likelihood to the negative items than positive items, and the gradient weight $w_\theta$ will be higher. Therefore, optimizing $\mathcal{L}_{\text{BPR-Diff}}$ is capable of handling hard negatives, which has become increasingly important in recent research Chen et al. (2022); Fan et al. (2023); Zhang et al. (2023).

**Connection with Direct Preference Optimization.** After determining how to minimize $\mathcal{L}_{\text{BPR-Diff}}$ using the aforementioned upper bound and analyzing the gradient, we proceed to validate the rationality of $\mathcal{L}_{\text{BPR-Diff}}$. Here, we establish a connection with the recently prominent Direct Preference Optimization (DPO) (Rafailov et al., 2023; Wallace et al., 2024; Meng et al., 2024), which has been shown to effectively align human feedback with large language models. For further details on DPO, we refer readers to (Rafailov et al., 2023). The equation of DPO is expressed as follows:

$$\mathcal{L}_{\text{DPO}}(\theta) = -\mathbb{E}_{(x_0^w, x_0^l, \mathbf{c})} \left[ \log \sigma \left( \beta \log \frac{p_\theta(x_0^w \mid \mathbf{c})}{p_{\text{ref}}(x_0^w \mid \mathbf{c})} - \beta \log \frac{p_\theta(x_0^l \mid \mathbf{c})}{p_{\text{ref}}(x_0^l \mid \mathbf{c})} \right) \right]. \tag{9}$$

By comparing equation 3 with equation 9, we observe that $\mathcal{L}_{\text{BPR-Diff}}$ can be viewed as a special case of DPO, where $\beta = 1$ and $p_{\text{ref}}$ is a constant distribution (e.g., uniform distribution). This validates that optimizing the proposed $\mathcal{L}_{\text{BPR-Diff}}$ has the potential to align user preferences in DMs. Notably, we give more details about the connection of DPO and PreferDiff in Appendix F.6.

### 3.3 EXTEND TO MULTIPLE NEGATIVES

As previous works have demonstrated that incorporating multiple negatives during the training phase can better capture user preferences, we extend $\mathcal{L}_{\text{BPR-Diff}}$ to support multiple negatives for instilling more fruitful rank information. Suppose that for each sequence, we have negative items' set $\mathcal{H}$ introduced in Section 2, according to equation 7, we can directly derive that:

$$\mathcal{L}_{\text{BPR-Diff-V}} = -\log \sigma(-|\mathcal{H}| \cdot (S(\hat{\mathbf{e}}_0^+, \mathbf{e}_0^+) - \frac{1}{|\mathcal{H}|} \sum_{v=1}^{|\mathcal{H}|} S(\hat{\mathbf{e}}_0^{-v}, \mathbf{e}_0^{-v})). \tag{10}$$

For brevity, we omit the expectation term. However, the above equation applies the noising and denoising process to all negative samples, which significantly reduces the model's training speed and increases susceptibility to false negatives. Therefore, we propose to replace the $|\mathcal{H}|$ negative samples with their centroid $\bar{\mathbf{e}}_0^- = \frac{1}{|\mathcal{H}|} \sum_{v=1}^{|\mathcal{H}|} \mathbf{e}_0^{-v}$ as the diffusion target and derive the following:

$$\mathcal{L}_{\text{BPR-Diff-C}} = -\log \sigma(-|\mathcal{H}| \cdot [S(\hat{\mathbf{e}}_0^+, \mathbf{e}_0^+) - S(\mathcal{F}_\theta(\bar{\mathbf{e}}_t^-, t, \mathcal{M}(\mathbf{c})), \bar{\mathbf{e}}_0^-)]). \tag{11}$$

Assuming that $\mathcal{F}(\cdot)$ is a convex function, we can apply Jensen's inequality and derive that $\mathcal{L}_{\text{BPR-Diff-V}} \leq \mathcal{L}_{\text{BPR-Diff-C}}$. Therefore, minimizing $\mathcal{L}_{\text{BPR-Diff-C}}$ can efficiently increase the likelihood of the positive items while simultaneously distancing them from the centroid of the negative items. Intuitively, this aligns with the phenomenon that users may not explicitly indicate dislike for specific items, but rather for a certain category of items. A detailed derivation can be found in Appendix C.4.

**Training and Inference of PreferDiff.** Here, we introduce the training and inference details of PreferDiff, as demonstrated in Algorithm 1 and Algorithm 2 in the Appendix. Empirically, we find that solely using the proposed $\mathcal{L}_{\text{BPR-Diff-C}}$ leads to instability during training. This may be due to an overemphasis on ranking information, which can neglect the more accurate generation of the next item. Therefore, we balance the trade-off between learning generation and learning preference with hyperparameter $\lambda$, with the following:

$$\mathcal{L}_{\text{PerferDiff}} = \underbrace{\lambda \mathcal{L}_{\text{Simple}}}_{\text{Learning Generation}} + \underbrace{(1-\lambda)\mathcal{L}_{\text{BPR-Diff-C}}}_{\text{Learning Preference}}. \tag{12}$$

We conduct experiments about different $\lambda$ to show the instable training issue in Section 4.3.

## 4 EXPERIMENTS

In this section, we aim to answer the following research questions:

• **RQ1**: How does PreferDiff perform compared with other sequential recommenders?

• **RQ2**: Can PreferDiff leverage pretraining to achieve commendable zero-shot performance on unseen datasets or datasets from other platforms just like DMs in other fields?

• **RQ3**: What is the impact of factors (e.g., $\lambda$) on PreferDiff's performance?

### 4.1 PERFORMANCE OF SEQUENTIAL RECOMMENDATION

**Baselines.** We comprehensively compare PreferDiff with five categories of sequential recommenders: traditional sequential recommenders, including GRU4Rec (Hidasi et al., 2016), SASRec (Kang &

Table 1: Comparison of the performance with sequential recommenders. The improvement achieved by PreferDiff is significant ($p$-value $\ll 0.05$). Results of three additional datasets are in Appendix F.1.

| Model | Sports and Outdoors | | | | Beauty | | | | Toys and Games | | | |
|---|---|---|---|---|---|---|---|---|---|---|---|---|
| | R@5 | N@5 | R@10 | N@10 | R@5 | N@5 | R@10 | N@10 | R@5 | N@5 | R@10 | N@10 |
| GRU4Rec | 0.0022 | 0.0020 | 0.0030 | 0.0023 | 0.0093 | 0.0078 | 0.0102 | 0.0081 | 0.0097 | 0.0087 | 0.0100 | 0.0090 |
| SASRec | 0.0047 | 0.0036 | 0.0067 | 0.0042 | 0.0138 | 0.0090 | 0.0219 | 0.0116 | 0.0133 | 0.0097 | 0.0170 | 0.0109 |
| BERT4Rec | 0.0101 | 0.0060 | 0.0157 | 0.0078 | 0.0174 | 0.0112 | 0.0286 | 0.0148 | 0.0226 | 0.0139 | 0.0304 | 0.0163 |
| CL4SRec | 0.0105 | 0.0070 | 0.0159 | 0.0085 | 0.0221 | 0.0123 | 0.0345 | 0.0178 | 0.0224 | 0.0142 | 0.0321 | 0.0169 |
| TIGER | 0.0093 | 0.0073 | 0.0166 | 0.0089 | 0.0236 | 0.0151 | 0.0366 | 0.0193 | 0.0185 | 0.0135 | 0.0252 | 0.0156 |
| DiffRec | 0.0125 | 0.0068 | 0.0200 | 0.0101 | 0.0195 | 0.0121 | 0.0409 | 0.0188 | 0.0268 | 0.0142 | 0.0426 | 0.0193 |
| DreamRec | 0.0155 | 0.0130 | 0.0211 | 0.0140 | 0.0406 | 0.0299 | 0.0483 | 0.0326 | 0.0440 | 0.0323 | 0.0490 | 0.0353 |
| DiffuRec | 0.0093 | 0.0078 | 0.0121 | 0.0087 | 0.0286 | 0.0215 | 0.0335 | 0.0230 | 0.0330 | 0.0262 | 0.0355 | 0.0271 |
| MoRec | 0.0056 | 0.0045 | 0.0076 | 0.0051 | 0.0259 | 0.0189 | 0.0353 | 0.0219 | 0.0154 | 0.0115 | 0.0191 | 0.0127 |
| LLM2BERT4Rec | 0.0118 | 0.0076 | 0.0183 | 0.0097 | 0.0379 | 0.0262 | 0.0474 | 0.0265 | 0.0339 | 0.0246 | 0.0443 | 0.0263 |
| PreferDiff | **0.0185** | **0.0147** | **0.0247** | **0.0167** | **0.0429** | 0.0323 | 0.0514 | 0.0350 | **0.0473** | **0.0367** | **0.0535** | **0.0387** |
| PreferDiff-T | 0.0182 | 0.0145 | 0.0222 | 0.0158 | 0.0429 | **0.0327** | **0.0532** | **0.0360** | 0.0460 | 0.0351 | 0.0525 | 0.0380 |
| Improve | 19.35% | 16.94% | 17.06% | 19.28% | 5.66% | 9.36% | 10.43% | 7.36% | 7.50% | 13.62% | 9.18% | 9.63% |

McAuley, 2018), and BERT4Rec (Sun et al., 2019); contrastive learning-based recommenders, such as CL4SRec (Xie et al., 2022); generative sequential recommenders like TIGER (Rajput et al., 2023); DM-based recommenders, including DiffRec (Wang et al., 2023b), DreamRec (Yang et al., 2023b) and DiffuRec (Li et al., 2024); and text-based recommenders like MoRec (Yuan et al., 2023) and LLM2Bert4Rec (Harte et al., 2023). See Appendix D.3 for details on the introduction, selection and hyperparameter of the baselines.

**Datasets.** We evaluate the proposed PreferDiff on six public real-world benchmarks (i.e., Sports, Beauty, and Toys from Amazon Reviews 2014 (He & McAuley, 2016), Steam, ML-1M, and Yahoo!R1). Detailed statistics of three benchmarks can be found in Table 5. Here, we utilize the common five-core datasets, filtering out users and items with fewer than five interactions. More Details about data prepossessing can be found in Appendix D.1. Following prior work (Yang et al., 2023b), in Table 1 and Table 14, we employ user-split which first sorts all sequences chronologically for each dataset, then split the data into training, validation, and test sets with an 8:1:1 ratio, while preserving the last 10 interactions as the historical sequence. We reproduce all baselines for a fair comparison. Notably, in Table 8 and Table 9 of Appendix D.4, we also give comparison under another setting (i.e., leave-one-out) to provide more insights where the baselines' results are copied from TIGIR. Moreover, we conduct experiments on varied user history lengths in Appendix F.2.

**Implementation Details.** For PerferDiff, for each user sequence, we treat the other next-items (a.k.a., labels) in the same batch as negative samples. We set the default diffusion timestep to 2000, DDIM step as 20, $p_u = 0.1$, and the $\beta$ linearly increase in the range of $[1e^{-4}, 0.02]$ for all DM-based sequential recommenders (e.g., DreamRec). For all text-based recommenders, we utilize OpenAI-3-Large (Neelakantan et al., 2022) to obtain the text embeddings. We fix the embedding dimension to 64 for all models except DM-based recommenders, as the latter only demonstrates strong performance with higher embedding dimensions. The former does not gain much from high embedding dimensions, which will be discussed in Section 4.3. Refer to Appendix D.2 for more implementation details about baselines. Notably, PreferDiff can be applied to any sequence encoder, $\mathcal{M}(\cdot)$. We provide the results of PreferDiff with other backbones in Appendix D.3.

**Evaluation Metrics.** We evaluate the recommendation performance in a full-ranking manner (Yang et al., 2023b) using Recall (Recall@K) and Normalized Discounted Cumulative Gain (NDCG@K) with K = 5, 10, following the widely adopted top-K protocol as the primary metrics for sequential recommendation (Kang & McAuley, 2018; Rajput et al., 2023).

**Results.** Table 1 presents the performance of PreferDiff compared with five categories sequential recommenders. For brevity, R stands for Recall, and N stands for NDCG. The top-performing and runner-up results are shown in bold and underlined, respectively. "Improv" represents the relative improvement percentage of PreferDiff over the best baseline. "*" indicates that the improvements are statistically significant at 0.05, according to the t-test. We can have the following observations:

• **DM-based recommenders have exhibited substantial performance gains over other sequential recommenders across most metrics.** This is consistent with prior research, which demonstrates that the powerful generation and generalization capabilities (Yang et al., 2023b) or noise robustness (Wang et al., 2023b; Li et al., 2024) of DM can better capture user behavior distributions

Table 2: Ablation Study of PreferDiff. Details are the same as Table 1.

| Model | Sports and Outdoors | | | | Beauty | | | | Toys and Games | | | |
|---|---|---|---|---|---|---|---|---|---|---|---|---|
| | R@5 | N@5 | R@10 | N@10 | R@5 | N@5 | R@10 | N@10 | R@5 | N@5 | R@10 | N@10 |
| **PreferDiff** | **0.0185** | **0.0147** | **0.0247** | **0.0167** | **0.0429** | **0.0323** | **0.0514** | **0.0350** | **0.0473** | **0.0367** | **0.0535** | **0.0387** |
| w/o-N | 0.0165 | 0.0139 | 0.0214 | 0.0149 | 0.0415 | 0.0304 | 0.0492 | 0.0333 | 0.0445 | 0.0349 | 0.0495 | 0.0367 |
| w/o-C | 0.0180 | 0.0139 | 0.0230 | 0.0159 | 0.0393 | 0.0282 | 0.0496 | 0.0322 | 0.0458 | 0.0356 | 0.0521 | 0.0374 |
| w/o-C&N | 0.0155 | 0.0130 | 0.0211 | 0.0140 | 0.0406 | 0.0299 | 0.0483 | 0.0326 | 0.0440 | 0.0323 | 0.0490 | 0.0353 |

compared to other sequential recommenders and alleviate the false negative or false positive issue in recommendation (Sato et al., 2020; Chen et al., 2023b).

• **PreferDiff significantly outperforms other DM-based recommenders across all metrics on three public benchmarks.** PreferDiff demonstrates an improvement ranging from 6.41% to 19.35% over the second-best baseline. Our results indicate that modeling the user's next-item distribution is more effective than modeling the user's interaction probability distribution (e.g., DiffRec) in sequential recommendation. Additionally, directly applying classic recommendation objectives (e.g., DiffuRec) or using objectives that deviate significantly from the original (e.g., MSE) may impede diffusion models from effectively learning user preference distributions and fully harnessing their generative and generalization capabilities. Moreover, the performance gap between DreamRec and PreferDiff further validates that our tailored optimization objective for DM-based recommenders successfully incorporates personalized ranking information into DMs, enabling them to better unleash their generative potential while more effectively capturing user preference distributions.

• **PreferDiff can benefit from advanced text-embeddings.** We observe that PreferDiff, when incorporating the identical text embeddings (referred to as PreferDiff-T), outperforms MoRec and LLM2Bert4Rec by replacing traditional ID embeddings with semantic text embeddings or using them as initialization parameters of ID-embeddings. This demonstrates that incorporating text embeddings, which provide a more semantic and stable feature space, into PreferDiff can obtain commendable recommendation performance. This finding aligns with current trends in the text-diffusion field (Lovelace et al., 2023; Liu et al., 2023). Building on this, due to the unified nature of the language space, PreferDiff possesses the potential to generalize sequential recommendations to other unseen domains, which we will elaborate on in the following subsection.

**Ablation Study.** As shown in Table 2, we scrutinize and evaluate each key individual component of PreferDiff to comprehend their respective impacts and significance. The ablation analysis is conducted using the following three versions. (1) PreferDiff-w/o-N employs cosine error as the measure function and drops the learning preference term in $\mathcal{L}_{\text{PreferDiff}}$. (2) PreferDiff-w/o-C employs MSE as a measure function. (3) PreferDiff-w/o-C&N employs MSE as the measure function and drops the learning preference term in $\mathcal{L}_{\text{PreferDiff}}$. We can observe that each component in PreferDiff contributes positively. Specifically, the performance degradation due to the omission of negative samples highlights the importance of incorporating preference information into DMs to better capture the underlying user preference distributions. Furthermore, replacing MSE with cosine error results in performance improvements, as the recommendation phase is conducted through maximum inner product search, which better aligns with the objective of capturing similarity in the embedding space.

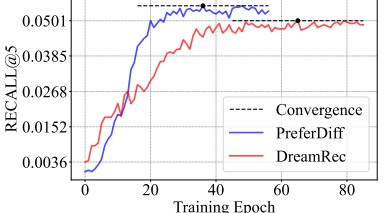 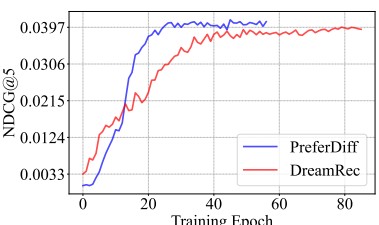

Figure 2: Training Comparison with DreamRec on Amazon Beauty.

**Faster Convergence than DreamRec.** As analyzed in Section 3.2, PreferDiff handles hard negatives with higher gradient weight, as shown in Figure 4.1. Empirically, we find that PreferDiff converges faster (approximately 35 epochs, 8 minutes) than other DM-based sequential recommenders, such as

Table 3: Performance comparison of General Sequential Recommendation on Different Target Datasets. Details are the same as Table 1.

| Supervision | Models | Metrics | In Domains | | Out Domains | | Other Platform |
| | | | Instruments | Tools | CDs | Movies | Steam |
|---|---|---|---|---|---|---|---|
| Full-Supervised | SASRec | R@5 | 0.1060 | 0.0673 | 0.0608 | 0.1392 | 0.0874 |
| | | N@5 | 0.0951 | 0.0642 | 0.0542 | 0.1210 | 0.0720 |
| Zero-Shot | UniSRec | R@5 | 0.1067 | 0.0627 | 0.0253 | 0.0286 | 0.0397 |
| | | N@5 | 0.1009 | 0.0605 | 0.0239 | 0.0271 | 0.0329 |
| | MoRec | R@5 | **0.1220** | 0.0699 | 0.0268 | 0.0306 | 0.0585 |
| | | N@5 | 0.1094 | 0.0655 | 0.0274 | 0.0293 | 0.0556 |
| | PreferDiff-T | R@5 | 0.1213 | **0.0723** | **0.0295** | **0.0312** | **0.0621** |
| | | N@5 | **0.1135** | **0.0691** | **0.0293** | **0.0299** | **0.0583** |

DreamRec (approximately 65 epochs, 15 minutes) with better performance on validation sets. Notably, we compare the training time and inference time with a 2-D scatter plot and table in Appendix F.4. By adjusting the denoising steps, we can achieve a trade-off between inference time and recommendation performance for real-time recommendation scenarios, as detailed in Appendix F.5.

## 4.2 General Sequential Recommendation (RQ2)

Given that DMs have exhibited exceptional zero-shot inference capabilities after pretraining on large, high-quality datasets in other fields (Khachatryan et al., 2023; Clark & Jaini, 2023), we aim to explore how PreferDiff can effectively zero-shot recommendation on unseen datasets, either within the same platform (e.g., Amazon) or across different platforms (e.g., Steam), without any overlap of users or items (Ding et al., 2021; Hou et al., 2023; Li et al., 2023a; Sheng et al., 2024), which distinguishes it from traditional ID-based cross-domain recommendation (Zhu et al., 2021; Ma et al., 2024b).

**Baselines.** Here, we compare PreferDiff with two baselines that are towards general sequential recommendations, namely UniSRec (Hou et al., 2022a) and MoRec (Yuan et al., 2023). See Appendix D.5 for details on the introduction, selection, and hyperparameter search range of the baselines. For a fair comparison, we employ the `text-embedding-3-large` model from OpenAI (Neelakantan et al., 2022) as the text encoder to convert identical item descriptions (e.g., title, category, brand) into representations, as it has been proven to deliver commendable performance in recommendation (Harte et al., 2023). More additional experiments about different text encoders can be found in Appendix E.3.

**Datasets and Evaluation Metrics.** Following the previous work (Hou et al., 2022a; Li et al., 2023a), we select five different product reviews from Amazon 2018 (Ni et al., 2019), namely, "Automotive", "Cell Phones and Accessories", "Grocery and Gourmet Food", "Musical Instruments" and "Tools and Home Improvement", as pretraining datasets. "Office Products" is selected as the validation dataset for early stopping when Recall@5 (i.e., **R@5**) shows no improvement for 20 consecutive epochs. Here, we consider three scenarios for the incoming evaluated target datasets. (1) "In Domains" refers to target datasets that are part of the pretraining dataset. (2) "Out Domains" refers to target datasets that are not in the pretraining dataset but belong to the same platform (i.e., Amazon). Here, we select "CDs and Vinyl" and "Movies and TV". (3) "Other Platform" refers to target datasets that are neither in the pretraining dataset nor from the same platform. Here, we select a commonly used game dataset collected from Steam (Kang & McAuley, 2018). Detailed dataset statistics can be found in Table 5.

**Results.** Tables 3 present the performance of PreferDiff compared with the chosen two general sequential recommenders. We can observe that:

• **Without any additional components, PreferDiff-T outperforms other general sequential recommenders.** Unlike UniSRec, which employs a mixture of experts technique for whitening, and MoRec, which uses dimension transformation, PreferDiff-T directly utilizes raw semantic text embeddings. This results in improvements of 2% to 8% in in-domain scenarios, 2% to 10% in out-domain scenarios, and 3% to 6% on other platforms, validating PreferDiff's strong capability in general sequential recommendation tasks without harming the performance on pretraining datasets.

• **The general sequential recommendation capacity of PreferDiff-T increases significantly as the amount of training data grows.** As shown in Figure 4, we empirically find that as we continuously expand the scale of the training data (by adding more diverse datasets), NDCG@5 and HR@5

have nearly improved **500%** as the scale of the training data increased five times, approaching the performance of full-supervised SASRec. This suggests that PreferDiff-T can effectively learn general knowledge to model user preference distributions by pretraining on even diverse datasets and transferring this knowledge to unseen datasets via advanced textual representations.

## 4.3 STUDY OF PREFERDIFF (RQ3)

In this subsection, we study the important factors (e.g., $\lambda$, embedding size, and $S(\cdot)$) that may impact the recommendation performance of PreferDiff. Others can be found in Appendix E.1 and Appendix E.2. We also provide visualization of learned item embeddings via t-SNE in Appendix E.4.

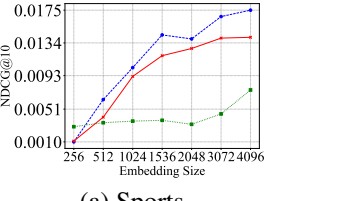 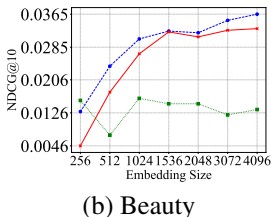 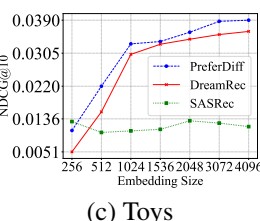

(a) Sports     (b) Beauty     (c) Toys

Figure 3: Effect of the Embedding Size for PreferDiff.

**Dimension of Embedding for PreferDiff.** As shown in Figure 3, we empirically observe that the recommendation performance of both PreferDiff and DreamRec improves significantly as the embedding size increases. This finding contrasts with previous observations in some non-DM-based recommenders (Liu et al., 2020; Qu et al., 2023; Guo et al., 2024). We attribute this phenomenon to the dynamic feature space of ID embeddings, in which DMs require higher dimensions to capture the user preference and ensure the stability of embedding space. Notably, in the Appendix F.3, we provide a simple theoretical analysis and experimental validation to explain this phenomenon.

**Importance of $\lambda$ for PreferDiff** $\lambda$ controls the balance between learning generation and learning preference in PreferDiff. As shown in Figure 5 of Appendix E, PreferDiff performs best when $\lambda = 0.4$ or $\lambda = 0.6$, highlighting the importance of enabling DMs to understand negatives in the recommendation task.

**Measure Function for PreferDiff.** As the final recommendation is ranked by maximal inner product search, we replace MSE with cosine error, as introduced in equation 7. The results presented in Table 4 demonstrate the superiority of using set cosine error as the default measurement function over MSE in PreferDiff.

Table 4: Effect of Measure Function for Prefer-Diff.

| Datasets | Sports | | Beauty | | Toys | |
|---|---|---|---|---|---|---|
| Measure | R@5 | N@5 | R@5 | N@5 | R@5 | N@5 |
| L1 | 0.0152 | 0.0121 | 0.0362 | 0.0281 | 0.0448 | 0.0345 |
| Huber | 0.0154 | 0.0123 | 0.0364 | 0.0279 | 0.0371 | 0.0286 |
| L2 | 0.0180 | 0.0139 | 0.0393 | 0.0282 | 0.0458 | 0.0356 |
| Cosine | 0.0185* | 0.0147* | 0.0429* | 0.0323* | 0.0473 | 0.0367* |

## 5 CONCLUSIONS AND LIMITATIONS

We propose PreferDiff, an optimization objective specifically designed for DM-based recommenders which can integrate multiple negative samples into DMs via generative modeling paradigm. Optimization is achieved through variational inference, deriving a variational upper bound as a surrogate objective. However, PreferDiff has limitations: (1) Dimension Sensitivity: The recommendation performance of PreferDiff is highly dependent on the embedding dimension. Empirical results show a sharp decline in performance when the embedding size is reduced to 64, a common dimension in existing studies. This dependency may lead to increased computational resources and slower training times when larger embedding sizes are required. (2) Hyperparameter $\lambda$ Dependence: PreferDiff heavily relies on the hyperparameter $\lambda$ to balance the generation and preference learning in DMs.

**Ethic Statement.** This paper aims to develop a specially tailored objective for DM-based recommenders through generative modeling. We do not anticipate any negative social impacts or violations of the ICLR code of ethics.

**Reproducibility Statement.** All results in this work are fully reproducible. The hyperparameter search space is discussed in Table 11, and further details about the hardware and software environment are provided in Appendix D.2. We provide the code and the best hyperparameters for our method at `https://github.com/lswhim/PreferDiff` and Table 12.

ACKNOWLEDGMENTS

This research is supported by the NExT Research Centre and the Spring B-Class Visiting Program of East China Normal University. We sincerely appreciate all the reviewers and the AC for their valuable suggestions during the review process. We would also like to thank Xinyue Ma for her support throughout the completion of this paper.

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

## A  RELATED WORK

We highlight key related works to contextualize how PreferDiff fits within and contributes to the broader literature. Specifically, our work aligns with research on sequential recommendation and DMs based recommenders.

**Sequential Recommendation** have gained significant attention in both academia (Rendle, 2022; Liu et al., 2024b) and industry (Wang et al., 2019; Fang et al., 2020) due to their ability to capture user preferences from historical interactions and recommend the next item. One common research line has focused on developing more efficient network architectures, such as GRU (Hidasi et al., 2016), convolutional neural networks (Tang & Wang, 2018), Transformer (Kang & McAuley, 2018; Fan et al., 2021), Bert4Rec (Devlin et al., 2019), and HSTU (Zhai et al., 2024). Another research line focuses on leveraging additional unsupervised signals (Xie et al., 2022; Wang et al., 2023a; Ren et al., 2024a) or reshaping sequential recommendation into other tasks such as retrieval (Rajput et al., 2023; Wang et al., 2024a) and language generation (Bao et al., 2023; Li et al., 2023b; Liao et al., 2024).

**DM-based Recommenders** have been explored in recent studies due to the powerful generative and generalization capabilities of DMs (DMs) (Lin et al., 2024). These recommenders either focus on modeling the distribution of the next item (e.g., (Yang et al., 2023b; Wang et al., 2024b; Li et al., 2024)), capture the probability distribution of user interactions (e.g., (Wang et al., 2023b; Zhao et al., 2024)), or focus on the distribution of time intervals between user behaviors (e.g., (Ma et al., 2024a)). However, existing approaches often rely on conventional objectives, such as mean squared error (MSE), or standard recommendation-specific objectives like Bayesian Personalized Ranking (BPR) (Rendle et al., 2009) and Cross Entropy (CE) (Klenitskiy & Vasilev, 2023). We argue that the former may diverge from the core objective of accurately modeling user preference distributions in recommendation tasks (Rendle, 2022), as DMs often lack an adequate understanding of negative items. While the latter leverages DMs' noise resistance to mitigate noisy interactions in recommendations which might fall short of fully exploiting the generative and generalization capabilities of DMs.

## B  SAMPLING ALGORITHM IN PREFERDIFF

We utilize DDIM (Song et al., 2021a) as the default sampler in PreferDiff, replacing the DDPM used in DreamRec, as we empirically find that DDIM is faster and performs better, requiring only a few denoising steps. Here, we briefly introduce how DDIM is employed in PreferDiff; Detailed derivations can be found in (Song et al., 2021a), and the code implementation is available at `https://github.com/lswhim/PreferDiff`.

**Details.** Specifically, in PreferDiff, the training is to predict the original data $\mathbf{e}_0$. The sampling process should be reparameterized to predict $\mathbf{e}_0$ directly instead of the noise $\epsilon$. Starting from the original DDIM update equation (Song et al., 2021a):

$$\mathbf{e}_{t-1} = \sqrt{\alpha_{t-1}} \left( \frac{\mathbf{e}_t - \sqrt{1-\alpha_t}\, \boldsymbol{\epsilon}_\theta(\mathbf{e}_t, t)}{\sqrt{\alpha_t}} \right) + \sqrt{1 - \alpha_{t-1} - \sigma_t^2}\, \boldsymbol{\epsilon}_\theta(\mathbf{e}_t, t) + \sigma_t \mathbf{z}, \qquad (13)$$

where $\mathbf{z} \sim \mathcal{N}(\mathbf{0}, \mathbf{I})$, $\sigma_t$ controls the stochasticity of the process, and $\boldsymbol{\epsilon}_\theta(\mathbf{e}_t, t)$ is the predicted noise at time step $t$.

In **PreferDiff**, since our model is trained to predict the original data $\mathbf{e}_0$ directly, we use the relationship between $\mathbf{e}_t$, $\mathbf{e}_0$, and the noise $\boldsymbol{\epsilon}$:

$$\mathbf{e}_t = \sqrt{\alpha_t}\, \mathbf{e}_0 + \sqrt{1 - \alpha_t}\, \boldsymbol{\epsilon}. \qquad (14)$$

Solving for $\boldsymbol{\epsilon}$, we obtain:

$$\boldsymbol{\epsilon} = \frac{\mathbf{e}_t - \sqrt{\alpha_t}\, \mathbf{e}_0}{\sqrt{1 - \alpha_t}}. \qquad (15)$$

Since $\mathbf{e}_0$ is predicted by our model as $\hat{\mathbf{e}}_0 = \mathcal{F}_\theta(\mathbf{e}_t, c, t)$, we can estimate the noise as:

$$\hat{\boldsymbol{\epsilon}}_\theta = \frac{\mathbf{e}_t - \sqrt{\alpha_t}\,\hat{\mathbf{e}}_0}{\sqrt{1-\alpha_t}}. \tag{16}$$

Substituting $\hat{\boldsymbol{\epsilon}}_\theta$ back into the DDIM update equation and setting $\sigma_t = 0$ for deterministic sampling, we get:

$$\mathbf{e}_{t-1} = \sqrt{\alpha_{t-1}}\left(\frac{\mathbf{e}_t - \sqrt{1-\alpha_t}\,\hat{\boldsymbol{\epsilon}}_\theta}{\sqrt{\alpha_t}}\right) + \sqrt{1-\alpha_{t-1}}\,\hat{\boldsymbol{\epsilon}}_\theta \tag{17}$$

$$= \sqrt{\alpha_{t-1}}\,\hat{\mathbf{e}}_0 + \sqrt{1-\alpha_{t-1}}\,\hat{\boldsymbol{\epsilon}}_\theta. \tag{18}$$

This simplification allows us to update $\mathbf{e}_{t-1}$ directly using the predicted $\hat{\mathbf{e}}_0$ and $\hat{\boldsymbol{\epsilon}}_\theta$ without introducing additional randomness, thus making the sampling process deterministic and more efficient.

**Summary.** Therefore, the deterministic DDIM sampling steps in our inference algorithm are:

1. **Predict** $\hat{\mathbf{e}}_0 = \mathcal{F}_\theta(\mathbf{e}_t, c, t)$.

2. **Compute** $\hat{\boldsymbol{\epsilon}}_\theta = \dfrac{\mathbf{e}_t - \sqrt{\alpha_t}\,\hat{\mathbf{e}}_0}{\sqrt{1-\alpha_t}}$.

3. **Update** $\mathbf{e}_{t-1} = \sqrt{\alpha_{t-1}}\,\hat{\mathbf{e}}_0 + \sqrt{1-\alpha_{t-1}}\,\hat{\boldsymbol{\epsilon}}_\theta$.

By iteratively applying these steps, we can efficiently generate the predicted original data $\hat{\mathbf{e}}_0$. During inference, by setting $\sigma_t = 0$, we eliminate the noise term $\sigma_t \mathbf{z}$ and focus solely on the deterministic components of the update rule. This results in faster convergence with fewer denoising steps while maintaining high-quality predictions. Detailed derivations and explanations of this reparameterization and the DDIM sampling process can be found in (Song et al., 2021a).

## C   DETAILS ABOUT PREFERDIFF

### C.1   FROM RATINGS TO PROBABILITY DISTRIBUTION

$$\mathcal{L}_{\text{BPR}} = -\mathbb{E}_{(\mathbf{e}_0^+, \mathbf{e}_0^-, \mathbf{c})}\left[\log \sigma\left(f_\theta(\mathbf{e}_0^+ \mid \mathbf{c}) - f_\theta(\mathbf{e}_0^- \mid \mathbf{c})\right)\right], \tag{19}$$

The primary objective of equation 19 is to maximize the rating margin between positive items and sampled negative items. Here, we employ softmax normalization to transform the rating ranking into a log-likelihood ranking.

We begin by expressing the rating $f_\theta(\mathbf{e}_0 \mid \mathbf{c})$ in terms of the probability distribution $p_\theta(\mathbf{e}_0 \mid \mathbf{c})$. This relationship is established through the following set of equations:

$$p_\theta(\mathbf{e}_0 \mid \mathbf{c}) = \frac{\exp(f_\theta(\mathbf{e}_0 \mid \mathbf{c}))}{Z_\theta},$$
$$\log p_\theta(\mathbf{e}_0 \mid \mathbf{c}) = f_\theta(\mathbf{e}_0 \mid \mathbf{c}) - \log Z_\theta,$$
$$f_\theta(\mathbf{e}_0 \mid \mathbf{c}) = \log p_\theta(\mathbf{e}_0 \mid \mathbf{c}) + \log Z_\theta. \tag{20}$$

Substituting equation 20 into equation 19 yields the BPR loss expressed solely in terms of the probability distributions of positive and negative items.

$$\mathcal{L}_{\text{BPR-Diff}} = -\mathbb{E}_{(\mathbf{e}_0^+, \mathbf{e}_0^-, \mathbf{c})} \left[ \log \sigma \left( \underbrace{f_\theta(\mathbf{e}_0^+ \mid \mathbf{c})}_{\text{rating of Positive Item}} - \underbrace{f_\theta(\mathbf{e}_0^- \mid \mathbf{c})}_{\text{rating of Negative Item}} \right) \right]$$

$$= -\mathbb{E}_{(\mathbf{e}_0^+, \mathbf{e}_0^-, \mathbf{c})} \left[ \log \sigma \left( \underbrace{\log p_\theta(\mathbf{e}_0^+ \mid \mathbf{c}) + \log Z_\theta}_{\text{From equation 20}} - \underbrace{\log p_\theta(\mathbf{e}_0^- \mid \mathbf{c}) - \log Z_\theta}_{\text{From equation 20}} \right) \right] \qquad (21)$$

$$= -\mathbb{E}_{(\mathbf{e}_0^+, \mathbf{e}_0^-, \mathbf{c})} \left[ \log \sigma \left( \log p_\theta(\mathbf{e}_0^+ \mid \mathbf{c}) - \log p_\theta(\mathbf{e}_0^- \mid \mathbf{c}) + \underbrace{\log Z_\theta - \log Z_\theta}_{=0} \right) \right]$$

$$= -\mathbb{E}_{(\mathbf{e}_0^+, \mathbf{e}_0^-, \mathbf{c})} \left[ \log \sigma \left( \log \frac{p_\theta(\mathbf{e}_0^+ \mid \mathbf{c})}{p_\theta(\mathbf{e}_0^- \mid \mathbf{c})} \right) \right].$$

### C.2 CONNECTING THE RATING FUNCTION TO THE SCORE FUNCTION

In this subsection, we establish the relationship between the rating function $f_\theta(\mathbf{e}_0 \mid \mathbf{c})$ and the score function in the context of score-based DMs. Specifically, we demonstrate that the gradient of the rating function with respect to the item embedding $\mathbf{e}_0$ is equivalent to the score function $\nabla_{\mathbf{e}_0} \log p_\theta(\mathbf{e}_0 \mid \mathbf{c})$.

Starting from Equation equation 20:

$$f_\theta(\mathbf{e}_0 \mid \mathbf{c}) = \log p_\theta(\mathbf{e}_0 \mid \mathbf{c}) + \log Z_\theta, \qquad (22)$$

where $Z_\theta$ is the partition function:

$$Z_\theta = \int \exp(f_\theta(\mathbf{e} \mid \mathbf{c})) \, d\mathbf{e}. \qquad (23)$$

DERIVATIVE OF THE RATING FUNCTION WITH RESPECT TO $\mathbf{e}_0$

Taking the gradient of Equation equation 22 with respect to $\mathbf{e}_0$, we have:

$$\nabla_{\mathbf{e}_0} f_\theta(\mathbf{e}_0 \mid \mathbf{c}) = \nabla_{\mathbf{e}_0} \log p_\theta(\mathbf{e}_0 \mid \mathbf{c}) + \nabla_{\mathbf{e}_0} \log Z_\theta. \qquad (24)$$

Since the partition function $Z_\theta$ is obtained by integrating over all possible item embeddings $\mathbf{e}$, and does not depend on the specific $\mathbf{e}_0$, its gradient with respect to $\mathbf{e}_0$ is zero:

$$\nabla_{\mathbf{e}_0} \log Z_\theta = 0. \qquad (25)$$

Therefore, Equation equation 24 simplifies to:

$$\nabla_{\mathbf{e}_0} f_\theta(\mathbf{e}_0 \mid \mathbf{c}) = \nabla_{\mathbf{e}_0} \log p_\theta(\mathbf{e}_0 \mid \mathbf{c}). \qquad (26)$$

**Definition of the Score Function** In score-based DMs, the **score function** is defined as the gradient of the log-probability density with respect to the data point $\mathbf{e}_0$:

$$\mathbf{s}_\theta(\mathbf{e}_0, \mathbf{c}) \triangleq \nabla_{\mathbf{e}_0} \log p_\theta(\mathbf{e}_0 \mid \mathbf{c}). \qquad (27)$$

Comparing Equations equation 26 and equation 27, we find that:

$$\nabla_{\mathbf{e}_0} f_\theta(\mathbf{e}_0 \mid \mathbf{c}) = \mathbf{s}_\theta(\mathbf{e}_0, \mathbf{c}). \qquad (28)$$

This reveals that the gradient of the rating function with respect to the item embedding $\mathbf{e}_0$ is exactly the score function of the probability distribution $p_\theta(\mathbf{e}_0 \mid \mathbf{c})$. Score-based DMs Song et al. (2021b) utilize the score function $\mathbf{s}_\theta(\mathbf{e}_0, \mathbf{c})$ to define the reverse diffusion process. In these models, the data generation process involves integrating the score function over time to recover the data distribution from noise. Intuitively, we can utilize $\nabla_{\mathbf{e}_0} f_\theta(\mathbf{e}_0 \mid \mathbf{c})$ to sample item embeddings with high ratings

through Langevin dynamics (Song & Ermon, 2020) given certain user historical conditions. Therefore, it bridges the objective of recommendation with generative modeling in DMs.

**Connection to Our Loss Function.** Our BPR-Diff loss function, as expressed in Equation equation 21, involves the log-ratio of the probabilities of positive and negative items:

$$\mathcal{L}_{\text{BPR-Diff}} = -\mathbb{E}_{(\mathbf{e}_0^+, \mathbf{e}_0^-, \mathbf{c})} \left[ \log \sigma \left( \log \frac{p_\theta(\mathbf{e}_0^+ \mid \mathbf{c})}{p_\theta(\mathbf{e}_0^- \mid \mathbf{c})} \right) \right] . \tag{29}$$

Using the equivalence between the rating function and the log-probability (from Equation equation 22), the loss function can also be seen as a function of the rating differences:

$$\mathcal{L}_{\text{BPR-Diff}} = -\mathbb{E} \left[ \log \sigma \left( f_\theta(\mathbf{e}_0^+ \mid \mathbf{c}) - f_\theta(\mathbf{e}_0^- \mid \mathbf{c}) \right) \right] . \tag{30}$$

**Gradient of the Loss with Respect to $\mathbf{e}_0$.** Taking the gradient of the loss function with respect to the positive item embedding $\mathbf{e}_0^+$, we get:

$$\nabla_{\mathbf{e}_0^+} \mathcal{L}_{\text{BPR-Diff}} = -\mathbb{E} \left[ \sigma(-s) \cdot \nabla_{\mathbf{e}_0^+} f_\theta(\mathbf{e}_0^+ \mid \mathbf{c}) \right] , \tag{31}$$

where $s = f_\theta(\mathbf{e}_0^+ \mid \mathbf{c}) - f_\theta(\mathbf{e}_0^- \mid \mathbf{c})$.

Similarly, for the negative item embedding $\mathbf{e}_0^-$:

$$\nabla_{\mathbf{e}_0^-} \mathcal{L}_{\text{BPR-Diff}} = \mathbb{E} \left[ \sigma(-s) \cdot \nabla_{\mathbf{e}_0^-} f_\theta(\mathbf{e}_0^- \mid \mathbf{c}) \right] . \tag{32}$$

These gradients indicate that the loss function encourages:

- Increasing the rating $f_\theta(\mathbf{e}_0^+ \mid \mathbf{c})$ of the positive item by moving $\mathbf{e}_0^+$ in the direction of $\nabla_{\mathbf{e}_0^+} f_\theta$.

- Decreasing the rating $f_\theta(\mathbf{e}_0^- \mid \mathbf{c})$ of the negative item by moving $\mathbf{e}_0^-$ opposite to $\nabla_{\mathbf{e}_0^-} f_\theta$.

### C.3    Derivation the Variational Upper Bound

In this section, we provide a comprehensive derivation of the upper bound for the proposed $\mathcal{L}_{\text{BPR-Diff}}$. We focus particularly on the steps involving the Kullback-Leibler divergence, leading to the final loss function used for training.

**Assumptions and Definitions:**

- $\mathbf{e}_0^+$ and $\mathbf{e}_0^-$ represent the embeddings of the positive and negative items, respectively.

- $\mathbf{e}_t^+$ and $\mathbf{e}_t^-$ are the noisy embeddings at timestep $t$ for the positive and negative items, obtained via the forward diffusion process.

- $\mathbf{c}$ denotes the historical item sequence for a user.

- $q(\mathbf{e}_{t-1} \mid \mathbf{e}_t, \mathbf{e}_0)$ is the posterior distribution in the forward diffusion process.

- $p_\theta(\mathbf{e}_{t-1} \mid \mathbf{e}_t, \mathbf{c})$ is the reverse diffusion process modeled by our neural network $\mathcal{F}_\theta$.

- $\mathcal{M}(\mathbf{c})$ is a mapping function that encodes the historical context $\mathbf{c}$ into a suitable representation for conditioning.

- $\sigma(\cdot)$ is the sigmoid function.

- $\beta_t$, $\alpha_t$, and $\bar{\alpha}_t$ are predefined constants in the diffusion schedule.

Starting from equation 4 in the main text, we have:

$$\mathcal{L}_{\text{BPR-Diff}}(\theta) = -\mathbb{E}_{(\mathbf{e}_0^+, \mathbf{e}_0^-, \mathbf{c})} \left[ \log \sigma \left( \log \mathbb{E}_{q(\mathbf{e}_{1:T}^+ \mid \mathbf{e}_0^+)} \left[ \frac{p_\theta(\mathbf{e}_{0:T}^+ \mid \mathbf{c})}{q(\mathbf{e}_{1:T}^+ \mid \mathbf{e}_0^+)} \right] - \log \mathbb{E}_{q(\mathbf{e}_{1:T}^- \mid \mathbf{e}_0^-)} \left[ \frac{p_\theta(\mathbf{e}_{0:T}^- \mid \mathbf{c})}{q(\mathbf{e}_{1:T}^- \mid \mathbf{e}_0^-)} \right] \right) \right] . \tag{33}$$

To address the intractability of directly computing the expectations inside the logarithms, we apply Jensen's inequality, which states that for a convex function $f$, we have $f(\mathbb{E}[X]) \leq \mathbb{E}[f(X)]$. Recognizing that $-\log \sigma(x)$ is convex in $x$, we obtain an upper bound:

$$\mathcal{L}_{\text{BPR-Diff}}(\theta) \leq -\mathbb{E}_{(\mathbf{e}_0^+, \mathbf{e}_0^-, \mathbf{c})} \mathbb{E}_{\substack{q(\mathbf{e}_{1:T}^+ | \mathbf{e}_0^+), \\ q(\mathbf{e}_{1:T}^- | \mathbf{e}_0^-)}} \left[ \log \sigma \left( \underbrace{\log \left[ \frac{p_\theta(\mathbf{e}_{0:T}^+ | \mathbf{c})}{q(\mathbf{e}_{1:T}^+ | \mathbf{e}_0^+)} \right]}_{(a)} - \underbrace{\log \left[ \frac{p_\theta(\mathbf{e}_{0:T}^- | \mathbf{c})}{q(\mathbf{e}_{1:T}^- | \mathbf{e}_0^-)} \right]}_{(b)} \right) \right] . \tag{34}$$

The terms (a) and (b) represent the variational lower bounds of the log-likelihoods for the positive and negative items, respectively. According to the properties of DMs (Ho et al., 2020), these terms can be related to the evidence lower bound (ELBO). Specifically, for any item $\mathbf{e}_0$, we have:

$$\log p_\theta(\mathbf{e}_0 | \mathbf{c}) \geq \mathbb{E}_{q(\mathbf{e}_{1:T} | \mathbf{e}_0)} \left[ \log \left( \frac{p_\theta(\mathbf{e}_{0:T} | \mathbf{c})}{q(\mathbf{e}_{1:T} | \mathbf{e}_0)} \right) \right] = -\mathcal{L}_{\text{ELBO}}(\theta; \mathbf{e}_0, \mathbf{c}) . \tag{35}$$

Substituting equation 35 into equation 34, we get:

$$\mathcal{L}_{\text{BPR-Diff}}(\theta) \leq -\mathbb{E}_{(\mathbf{e}_0^+, \mathbf{e}_0^-, \mathbf{c})} \left[ \log \sigma \left( -\mathcal{L}_{\text{ELBO}}(\theta; \mathbf{e}_0^+, \mathbf{c}) + \mathcal{L}_{\text{ELBO}}(\theta; \mathbf{e}_0^-, \mathbf{c}) \right) \right] . \tag{36}$$

The ELBO for each item can be decomposed into a sum over timesteps $t$:

$$\mathcal{L}_{\text{ELBO}}(\theta; \mathbf{e}_0, \mathbf{c}) = \sum_{t=1}^{T} \mathbb{E}_{q(\mathbf{e}_t | \mathbf{e}_0)} \left[ D_{\text{KL}} \left( q(\mathbf{e}_{t-1} | \mathbf{e}_t, \mathbf{e}_0) \| p_\theta(\mathbf{e}_{t-1} | \mathbf{e}_t, \mathbf{c}) \right) \right] + C , \tag{37}$$

where $C$ is a constant independent of $\theta$.

Substituting equation 37 back into equation 36, we obtain:

$$\mathcal{L}_{\text{BPR-Diff}}(\theta) \leq -\mathbb{E}_{(\mathbf{e}_0^+, \mathbf{e}_0^-, \mathbf{c})} \left[ \log \sigma \left( - \left( \sum_{t=1}^{T} \mathbb{E}_{q(\mathbf{e}_t^+ | \mathbf{e}_0^+)} \left[ D_{\text{KL}} \left( q(\mathbf{e}_{t-1}^+ | \mathbf{e}_t^+, \mathbf{e}_0^+) \| p_\theta(\mathbf{e}_{t-1}^+ | \mathbf{e}_t^+) \right) \right] \right. \right. \right.$$
$$\left. \left. \left. - \sum_{t=1}^{T} \mathbb{E}_{q(\mathbf{e}_t^- | \mathbf{e}_0^-)} \left[ D_{\text{KL}} \left( q(\mathbf{e}_{t-1}^- | \mathbf{e}_t^-, \mathbf{e}_0^-) \| p_\theta(\mathbf{e}_{t-1}^- | \mathbf{e}_t^-) \right) \right] + C_1 \right) \right) \right] , \tag{38}$$

where $C_1$ aggregates constants and is independent of $\theta$.

Now, we focus on the KL divergence terms. In DMs, both $q(\mathbf{e}_{t-1} | \mathbf{e}_t, \mathbf{e}_0)$ and $p_\theta(\mathbf{e}_{t-1} | \mathbf{e}_t, \mathbf{c})$ are Gaussian distributions (Ho et al., 2020). Specifically, for the forward process $q$ and the reverse process $p_\theta$, we have:

$$q(\mathbf{e}_{t-1} | \mathbf{e}_t, \mathbf{e}_0) = \mathcal{N} \left( \mathbf{e}_{t-1}; \tilde{\boldsymbol{\mu}}_t(\mathbf{e}_t, \mathbf{e}_0), \tilde{\beta}_t \mathbf{I} \right) , \tag{39}$$

$$p_\theta(\mathbf{e}_{t-1} | \mathbf{e}_t, \mathbf{c}) = \mathcal{N} \left( \mathbf{e}_{t-1}; \boldsymbol{\mu}_\theta(\mathbf{e}_t, t, \mathbf{c}), \beta_t \mathbf{I} \right) , \tag{40}$$

where $\tilde{\boldsymbol{\mu}}_t(\mathbf{e}_t, \mathbf{e}_0)$ is the mean of the posterior $q(\mathbf{e}_{t-1} | \mathbf{e}_t, \mathbf{e}_0)$, $\tilde{\beta}_t$ is the variance, and $\beta_t$ is the variance schedule for the reverse process.

The KL divergence between two Gaussian distributions can be computed as:

$$D_{\text{KL}} \left( q \| p_\theta \right) = \frac{1}{2} \left( \text{tr} \left( \beta_t^{-1} \tilde{\beta}_t \mathbf{I} \right) + (\boldsymbol{\mu}_\theta - \tilde{\boldsymbol{\mu}}_t)^\top \beta_t^{-1} \mathbf{I} (\boldsymbol{\mu}_\theta - \tilde{\boldsymbol{\mu}}_t) - k + \ln \left( \frac{\det(\beta_t \mathbf{I})}{\det(\tilde{\beta}_t \mathbf{I})} \right) \right) , \tag{41}$$

where $k$ is the dimensionality of the Gaussian distributions (i.e., the embedding dimension).

Assuming that $\tilde{\beta}_t = \beta_t$ (Ho et al., 2020), the trace term simplifies to $k$, and the determinant term becomes $\ln(1) = 0$. Therefore, the KL divergence simplifies to:

$$D_{\text{KL}} \left( q \| p_\theta \right) = \frac{1}{2\beta_t} \| \boldsymbol{\mu}_\theta - \tilde{\boldsymbol{\mu}}_t \|_2^2 . \tag{42}$$

Next, we define the network prediction $\boldsymbol{\mu}_\theta$ and relate it to the mean $\tilde{\boldsymbol{\mu}}_t$ from the forward process.

**Relationship between $\tilde{\boldsymbol{\mu}}_t$ and $\mathbf{e}_0$:**

The mean $\tilde{\boldsymbol{\mu}}_t$ is given by:

$$\tilde{\boldsymbol{\mu}}_t(\mathbf{e}_t, \mathbf{e}_0) = \frac{\sqrt{\bar{\alpha}_{t-1}}\beta_t}{1 - \bar{\alpha}_t}\mathbf{e}_0 + \frac{\sqrt{\alpha_t}(1 - \bar{\alpha}_{t-1})}{1 - \bar{\alpha}_t}\mathbf{e}_t , \tag{43}$$

where $\alpha_t = 1 - \beta_t$, and $\bar{\alpha}_t = \prod_{s=1}^t \alpha_s$. In practice, it is common to predict $\mathbf{e}_0$ directly using the neural network $\mathcal{F}_\theta$:

$$\hat{\mathbf{e}}_0 = \mathcal{F}_\theta(\mathbf{e}_t, t, \mathcal{M}(\mathbf{c})) . \tag{44}$$

Given $\hat{\mathbf{e}}_0$, we can compute $\boldsymbol{\mu}_\theta$ as:

$$\boldsymbol{\mu}_\theta(\mathbf{e}_t, t, \mathbf{c}) = \frac{\sqrt{\bar{\alpha}_{t-1}}\beta_t}{1 - \bar{\alpha}_t}\hat{\mathbf{e}}_0 + \frac{\sqrt{\alpha_t}(1 - \bar{\alpha}_{t-1})}{1 - \bar{\alpha}_t}\mathbf{e}_t . \tag{45}$$

Substituting equations equation 43 and equation 45 into equation 42, we have:

$$D_{\mathrm{KL}}\left(q \,\|\, p_\theta\right) = \frac{1}{2\beta_t}\|\boldsymbol{\mu}_\theta - \tilde{\boldsymbol{\mu}}_t\|_2^2 = \frac{1}{2\beta_t}\left\|\left(\frac{\sqrt{\bar{\alpha}_{t-1}}\beta_t}{1 - \bar{\alpha}_t}(\hat{\mathbf{e}}_0 - \mathbf{e}_0)\right)\right\|_2^2 = \frac{(\sqrt{\bar{\alpha}_{t-1}}\beta_t)^2}{2\beta_t^2(1 - \bar{\alpha}_t)^2}\|\hat{\mathbf{e}}_0 - \mathbf{e}_0\|_2^2 . \tag{46}$$

Simplifying the constants, we observe that the coefficient reduces to a constant factor dependent on $t$, which we can denote as $\lambda_t$:

$$\lambda_t = \frac{(\sqrt{\bar{\alpha}_{t-1}}\beta_t)^2}{2\beta_t^2(1 - \bar{\alpha}_t)^2} = \frac{\bar{\alpha}_{t-1}}{2(1 - \bar{\alpha}_t)^2} . \tag{47}$$

Therefore, the KL divergence becomes:

$$D_{\mathrm{KL}}\left(q \,\|\, p_\theta\right) = \lambda_t \|\hat{\mathbf{e}}_0 - \mathbf{e}_0\|_2^2 . \tag{48}$$

Since $\lambda_t$ is independent of $\theta$ and depends only on $t$, when we sum over all timesteps and average over $t$, this term becomes proportional to the mean squared error between $\hat{\mathbf{e}}_0$ and $\mathbf{e}_0$.

**Equivalence of MSE and Cosine Error for Unit Norm Vectors:**

Alternatively, to mitigate sensitivity to vector norms and dimensionality (Friedman, 1997; Hou et al., 2022b) (the recommendation performance of PreferDiff is competitive when embedding size is higher), we can use the cosine error as the distance measure. The cosine similarity between $\hat{\mathbf{e}}_0$ and $\mathbf{e}_0$ is given by:

$$\cos\left(\hat{\mathbf{e}}_0, \mathbf{e}_0\right) = \frac{\hat{\mathbf{e}}_0^\top \mathbf{e}_0}{\|\hat{\mathbf{e}}_0\|_2 \|\mathbf{e}_0\|_2} . \tag{49}$$

The cosine error is then:

$$S\left(\hat{\mathbf{e}}_0, \mathbf{e}_0\right) = 1 - \cos\left(\hat{\mathbf{e}}_0, \mathbf{e}_0\right) . \tag{50}$$

Actually, when both $\hat{\mathbf{e}}_0$ and $\mathbf{e}_0$ are normalized to have unit norm (i.e., $\|\hat{\mathbf{e}}_0\|_2 = \|\mathbf{e}_0\|_2 = 1$), the mean squared error and the cosine error are directly related. Specifically, the squared Euclidean distance between two unit vectors is:

$$\|\hat{\mathbf{e}}_0 - \mathbf{e}_0\|_2^2 = (\hat{\mathbf{e}}_0 - \mathbf{e}_0)^\top (\hat{\mathbf{e}}_0 - \mathbf{e}_0) = \|\hat{\mathbf{e}}_0\|_2^2 + \|\mathbf{e}_0\|_2^2 - 2\hat{\mathbf{e}}_0^\top \mathbf{e}_0 = 2(1 - \cos\left(\hat{\mathbf{e}}_0, \mathbf{e}_0\right)) . \tag{51}$$

Thus, under the unit norm constraint, minimizing the MSE is equivalent to minimizing the cosine error up to a constant factor of 2. This shows that both distance measures capture the same notion of similarity in this case. Substituting the KL divergence approximation back into equation 38, and considering both positive and negative items, we simplify the expression:

$$\mathcal{L}_{\text{BPR-Diff}}(\theta) \leq -\mathbb{E}_{(\mathbf{e}_0^+, \mathbf{e}_0^-, \mathbf{c}),\, t\sim U(1,T)}\left[\log \sigma\left(-\left(\underbrace{S\left(\hat{\mathbf{e}}_0^+, \mathbf{e}_0^+\right)}_{\text{Positive item error}} - \underbrace{S\left(\hat{\mathbf{e}}_0^-, \mathbf{e}_0^-\right)}_{\text{Negative item error}}\right)\right)\right] , \tag{52}$$

where $\hat{\mathbf{e}}_0^+ = \mathcal{F}_\theta(\mathbf{e}_t^+, t, \mathcal{M}(\mathbf{c}))$ and $\hat{\mathbf{e}}_0^- = \mathcal{F}_\theta(\mathbf{e}_t^-, t, \mathcal{M}(\mathbf{c}))$.

Equation equation 52 represents our final trainable objective:

$$\mathcal{L}_{\text{Upper}}(\theta) = -\mathbb{E}_{(\mathbf{e}_0^+, \mathbf{e}_0^-, \mathbf{c}), \, t \sim U(1,T)} \left[ \log \sigma \left( - \left( S \left( \mathcal{F}_\theta(\mathbf{e}_t^+, t, \mathcal{M}(\mathbf{c})), \mathbf{e}_0^+ \right) - S \left( \mathcal{F}_\theta(\mathbf{e}_t^-, t, \mathcal{M}(\mathbf{c})), \mathbf{e}_0^- \right) \right) \right) \right] . \tag{53}$$

**Explanation**. This objective encourages the model to minimize the distance between the predicted embedding and the true embedding for the positive item while maximizing the distance for the negative item, effectively widening the gap between them in the latent space. By doing so, we enhance the personalized ranking capability of the model.

**Summary**. By minimizing $\mathcal{L}_{\text{Upper}}(\theta)$, we implicitly minimize the original $\mathcal{L}_{\text{BPR-Diff}}(\theta)$ due to the application of Jensen's inequality. This aligns the training objective with the goal of improving personalized ranking by leveraging DMs within the BPR.

## C.4 Extend into Multiple Negative Samples

In this section, we provide a detailed derivation of the inequality $\mathcal{L}_{\text{BPR-Diff-V}} \leq \mathcal{L}_{\text{BPR-Diff-C}}$, under the assumption that $\mathcal{F}_\theta$ and $S$ are convex functions.

**Definitions and Assumptions**

We define:

- $\mathcal{F}_\theta(\mathbf{e}_t, t, \mathcal{M}(\mathbf{c}))$: the denoising function at time step $t$, parameterized by $\theta$, conditioned on context $\mathcal{M}(\mathbf{c})$.
- $S(\mathbf{a}, \mathbf{b})$: a measure function quantifying the discrepancy between vectors $\mathbf{a}$ and $\mathbf{b}$, such as Mean Squared Error (MSE).
- $\sigma(\cdot)$: the sigmoid function.

Assume that:

- $\mathcal{F}_\theta$ is convex with respect to its input $\mathbf{e}_t$.
- $S$ is convex with respect to both of its inputs.

Starting with the definition of $\mathcal{L}_{\text{BPR-Diff-V}}$:

$$\mathcal{L}_{\text{BPR-Diff-V}} = -\log \sigma \left( -V \left( S \left( \mathcal{F}_\theta \left( \mathbf{e}_t^+, t, \mathcal{M}(\mathbf{c}) \right), \mathbf{e}_0^+ \right) - \frac{1}{V} \sum_{v=1}^V S \left( \mathcal{F}_\theta \left( \mathbf{e}_t^{-v}, t, \mathcal{M}(\mathbf{c}) \right), \mathbf{e}_0^{-v} \right) \right) \right) . \tag{54}$$

Similarly, for $\mathcal{L}_{\text{BPR-Diff-C}}$:

$$\mathcal{L}_{\text{BPR-Diff-C}} = -\log \sigma \left( -V \left( S \left( \mathcal{F}_\theta \left( \mathbf{e}_t^+, t, \mathcal{M}(\mathbf{c}) \right), \mathbf{e}_0^+ \right) - S \left( \mathcal{F}_\theta \left( \tilde{\mathbf{e}}_t^-, t, \mathcal{M}(\mathbf{c}) \right), \tilde{\mathbf{e}}_0^- \right) \right) \right), \tag{55}$$

where we have defined the centroids:

$$\tilde{\mathbf{e}}_t^- = \frac{1}{V} \sum_{v=1}^V \mathbf{e}_t^{-v}, \quad \tilde{\mathbf{e}}_0^- = \frac{1}{V} \sum_{v=1}^V \mathbf{e}_0^{-v}. \tag{56}$$

Our aim is to show that $\mathcal{L}_{\text{BPR-Diff-V}} \leq \mathcal{L}_{\text{BPR-Diff-C}}$.

First, consider the term:

$$D_V = S \left( \mathcal{F}_\theta \left( \mathbf{e}_t^+, t, \mathcal{M}(\mathbf{c}) \right), \mathbf{e}_0^+ \right) - \frac{1}{V} \sum_{v=1}^V S \left( \mathcal{F}_\theta \left( \mathbf{e}_t^{-v}, t, \mathcal{M}(\mathbf{c}) \right), \mathbf{e}_0^{-v} \right) . \tag{57}$$

By the convexity of $S$, we have:

$$\frac{1}{V}\sum_{v=1}^{V} S\left(\mathcal{F}_\theta\left(\mathbf{e}_t^{-v}, t, \mathcal{M}(\mathbf{c})\right), \mathbf{e}_0^{-v}\right) \leq S\left(\underbrace{\frac{1}{V}\sum_{v=1}^{V}\mathcal{F}_\theta\left(\mathbf{e}_t^{-v}, t, \mathcal{M}(\mathbf{c})\right)}_{\text{Convex combination of }\mathcal{F}_\theta(\mathbf{e}_t^{-v})}, \underbrace{\frac{1}{V}\sum_{v=1}^{V}\mathbf{e}_0^{-v}}_{\tilde{\mathbf{e}}_0^-}\right). \tag{58}$$

Next, using the convexity of $\mathcal{F}_\theta$, we have:

$$\mathcal{F}_\theta\left(\tilde{\mathbf{e}}_t^-, t, \mathcal{M}(\mathbf{c})\right) \leq \underbrace{\frac{1}{V}\sum_{v=1}^{V}\mathcal{F}_\theta\left(\mathbf{e}_t^{-v}, t, \mathcal{M}(\mathbf{c})\right)}_{\text{Convex combination}}. \tag{59}$$

Combining equation 58 and equation 59, and recognizing that $S$ is non-decreasing with respect to its first argument, we get:

$$\frac{1}{V}\sum_{v=1}^{V} S\left(\mathcal{F}_\theta\left(\mathbf{e}_t^{-v}, t, \mathcal{M}(\mathbf{c})\right), \mathbf{e}_0^{-v}\right) \leq S\left(\mathcal{F}_\theta\left(\tilde{\mathbf{e}}_t^-, t, \mathcal{M}(\mathbf{c})\right), \tilde{\mathbf{e}}_0^-\right). \tag{60}$$

Therefore, we have:

$$D_V = S\left(\mathcal{F}_\theta\left(\mathbf{e}_t^+, t, \mathcal{M}(\mathbf{c})\right), \mathbf{e}_0^+\right) - \frac{1}{V}\sum_{v=1}^{V} S\left(\mathcal{F}_\theta\left(\mathbf{e}_t^{-v}, t, \mathcal{M}(\mathbf{c})\right), \mathbf{e}_0^{-v}\right) \tag{61}$$

$$\geq S\left(\mathcal{F}_\theta\left(\mathbf{e}_t^+, t, \mathcal{M}(\mathbf{c})\right), \mathbf{e}_0^+\right) - S\left(\mathcal{F}_\theta\left(\tilde{\mathbf{e}}_t^-, t, \mathcal{M}(\mathbf{c})\right), \tilde{\mathbf{e}}_0^-\right) = D_C. \tag{62}$$

Since $D_V \geq D_C$, it follows that:

$$-V D_V \leq -V D_C. \tag{63}$$

Applying the monotonicity of the $\log \sigma(\cdot)$ function (since $\sigma$ is an increasing function and $\log$ is monotonic), we have:

$$\mathcal{L}_{\text{BPR-Diff-V}} = -\log\sigma(-V D_V) \leq -\log\sigma(-V D_C) = \mathcal{L}_{\text{BPR-Diff-C}}. \tag{64}$$

Therefore, we have shown that:

$$\mathcal{L}_{\text{BPR-Diff-V}} \leq \mathcal{L}_{\text{BPR-Diff-C}}. \tag{65}$$

**Explanation.** This inequality implies that minimizing $\mathcal{L}_{\text{BPR-Diff-C}}$ effectively minimizes an upper bound of $\mathcal{L}_{\text{BPR-Diff-V}}$, leading to an efficient increase in the likelihood of positive items while distancing them from the centroid of negative items. Notably, although the assumption of convexity is difficult to satisfy in practice, the aforementioned method still empirically achieves strong results than one negative item.

## D   EXPERIMENTS

### D.1   DATASETS PREPOSSESSING IN USER SPLITTING SETTING

Following prior works (Yang et al., 2023a;b), we adopt the user-splitting setting, which has been shown to effectively prevent information leakage in test sets (Ji et al., 2023). Specifically, we first

---

**Algorithm 1** Training Phase of PreferDiff

---

1: **Input:** Trainable parameters $\theta$, training dataset $\mathcal{D}_{\text{train}} = \{(\mathbf{e}_0^+, \mathbf{c}, \mathcal{H})\}_{n=1}^{|\mathcal{D}_{\text{train}}|}$, total steps $T$, unconditional probability $p_u$, learning rate $\eta$, variance schedules $\{\alpha_t\}_{t=1}^{T}$
2: **Output:** Updated parameters $\theta$
3: **repeat**
4:     $(\mathbf{e}_0^+, \mathbf{c}, \mathcal{H}) \sim \mathcal{D}_{\text{train}}$                  $\triangleright$ Sample data from training dataset.
5:     **With probability** $p_u$: $c = \Phi$         $\triangleright$ Set unconditional condition with probability $p_u$.
6:     $t \sim \text{Uniform}(1, T)$, $\epsilon^+, \epsilon^- \sim \mathcal{N}(\mathbf{0}, \mathbf{I})$        $\triangleright$ Sample diffusion step and noise.
7:     $\mathbf{e}_t^+ = \sqrt{\bar{\alpha}_t}\mathbf{e}_0^+ + \sqrt{1 - \bar{\alpha}_t}\epsilon^+$          $\triangleright$ Add noise to positive item embedding.
8:     $\mathbf{e}_t^- = \frac{\sqrt{\bar{\alpha}_t}}{V}\sum_{v=1}^{V}\mathbf{e}_0^{-v} + \sqrt{1 - \bar{\alpha}_t}\epsilon^-$     $\triangleright$ Add noise to negative item embeddings' centroid.
9:     $\theta \leftarrow \theta - \eta\nabla_\theta\mathcal{L}_{\text{PreferDiff}}(\mathbf{e}_t^+, \mathbf{e}_t^-, t, \mathbf{c}, \Phi; \theta)$        $\triangleright$ Gradient descent update.
10: **until** convergence
11: **return** $\theta$

---

**Algorithm 2** Inference Phase of PreferDiff

---

1: **Input:** Trained parameters $\theta$, Sequence encoder $\mathcal{M}(\cdot)$, test dataset $\mathcal{D}_{\text{test}} = \{(\mathbf{e}_0, \mathbf{c})\}_{n=1}^{|\mathcal{D}_{\text{test}}|}$, total steps $T$, DDIM steps $S$, guidance weight $w$, variance schedules $\{\alpha_t\}_{t=1}^{T}$
2: **Output:** Predicted next item $\hat{\mathbf{e}}_0$
3: $\mathbf{c} \sim \mathcal{D}_{\text{test}}$                $\triangleright$ Sample user historical sequence from testing dataaset.
4: $\mathbf{e}_T \sim \mathcal{N}(\mathbf{0}, \mathbf{I})$                 $\triangleright$ Sample standard Gaussian noise.
5: **for** $s = S, \ldots, 1$ **do**                 $\triangleright$ Denoise over $S$ DDIM steps.
6:     $t = \lfloor s \times (T/S) \rfloor$             $\triangleright$ Map DDIM step $s$ to original step $t$.
7:     **With probability** $p_u$: $\mathcal{M}(\mathbf{c}) = \Phi$      $\triangleright$ Set unconditional condition with probability $p_u$.
8:     $\mathbf{z} \sim \mathcal{N}(\mathbf{0}, \mathbf{I})$ **if** $s > 1$ **else** $\mathbf{z} = 0$       $\triangleright$ Sample noise if not final step.
9:     $\hat{\mathbf{e}}_0 = (1 + w)\mathcal{F}_\theta(\hat{\mathbf{e}}_t, \mathcal{M}(\mathbf{c}), t) - w\mathcal{F}_\theta(\hat{\mathbf{e}}_t, \Phi, t)$    $\triangleright$ Apply classifier-free guidance.
10:     $\hat{\epsilon}_\theta = \frac{\hat{\mathbf{e}}_t - \sqrt{\bar{\alpha}_t}\hat{\mathbf{e}}_0}{\sqrt{1 - \bar{\alpha}_t}}$               $\triangleright$ Compute predicted noise.
11:     $\hat{\mathbf{e}}_{t-1} = \sqrt{\bar{\alpha}_{t-1}}\hat{\mathbf{e}}_0 + \sqrt{1 - \bar{\alpha}_{t-1}}\hat{\epsilon}_\theta$        $\triangleright$ DDIM update step when $\sigma_t = 0$.
12: **end for**
13: **return** $\hat{\mathbf{e}}_0$

---

Table 5: Detailed Statistics of Datasets after Preprocessing.

| Datasets | Fully Trained Recommendation | | | | | | General Sequential Recommendation | | | | |
|---|---|---|---|---|---|---|---|---|---|---|---|
| | Sports | Beauty | Toys | Steam | ML-1M | Yahoo!R1 | Pretraining | Validation | CDs | Movies | Steam |
| #Sequences | 35,598 | 22,363 | 19,412 | 39,795 | 6,040 | 50,000 | 746,688 | 101,501 | 112,379 | 297,529 | 39,795 |
| #Items | 18,357 | 12,101 | 11,924 | 9,265 | 3,706 | 23,589 | 68,668 | 8,623 | 15,520 | 25,925 | 9,265 |
| #Interactions | 256,598 | 162,150 | 138,444 | 437,733 | 60,400 | 500,000 | 3,258,523 | 452,415 | 457,589 | 2,053,497 | 437,733 |

sort all sequences chronologically for each dataset, then split the data into training, validation, and test sets with an 8:1:1 ratio, while preserving the last 10 interactions as the historical sequence.

**Amazon 2014** [1]. Here, we choose three public real-world benchmarks (i.e., Sports, Beauty and Toys) which has been widely utilized in recent studies (Rajput et al., 2023). Here, we utilize the common five-core datasets (Hou et al., 2022a), filtering out users and items with fewer than five interactions across all datasets. Following previous work (Yang et al., 2023b), we set the maximized length user interaction sequence as 10.

**Amazon 2018** [2]. Following prior works (Hou et al., 2022a; Li et al., 2023a), we select five distinct product review categories—namely, "Automotive," "Electronics," "Grocery and Gourmet Food," "Musical Instruments," and "Tools and Home Improvement"—as pretraining datasets. "Cell Phones and Accessories" is used as the validation set for early stopping. In line with previous research (Yang et al., 2023b), we filter out items with fewer than 20 interactions and user interaction sequences shorter than 5, capping the maximum length of each user's interaction sequence at 10.

**Steam** is a game review dataset collected from Steam [3]. Due to the large number of game reviews, we filter out users and items with fewer than 20 interactions.

**ML-1M** is a movie rating dataset collected by GroupLens [4]. We filter out users and items with fewer than 20 interactions.

**Yahoo!R1** is a music rating dataset collected by Yahoo [5]. We filter out users and items with fewer than 20 interactions.

## D.2 IMPLEMENTATION DETAILS

For a fair comparison, all experiments are conducted in PyTorch using a single Tesla V100-SXM3-32GB GPU and an Intel(R) Xeon(R) Gold 6248R CPU. We optimize all methods using the AdamW optimizer and all models' parameters are initialized with Standard Normal initialization. We fix the embedding dimension to 64 for all models except DM-based recommenders, as the latter only demonstrate strong performance with higher embedding dimensions, as discussed in Section 4.3. Since our focus is not on network architecture and for fair comparison, we adopt a lightweight configuration for baseline models that employ a Transformer backbone [6], using a single layer with two attention heads. **Notably, all baselines, unless otherwise specified, use cross-entropy as the loss function**, as recent studies (Zhang et al., 2024; Klenitskiy & Vasilev, 2023; Zhai et al., 2023) have demonstrated its effectiveness.

For PerferDiff, for each user sequence, we treat the other next-items (a.k.a., labels) in the same batch as negative samples. We set the default diffusion timestep to 2000, DDIM step as 20, $p_u = 0.1$, and the $\beta$ linearly increase in the range of $[1e^{-4}, 0.02]$ for all DM-basd sequential recommenders (e.g., DreamRec). We empirically find that tuning these parameters may lead to better recommendation performance. However, as this is not the focus of the paper, we do not elaborate on it.

The other hyperparameter (e.g., learning rate) search space for PreferDiff and the baseline models is provided in Table 11, while the best hyperparameters for PreferDiff are listed in Table 12.

---

[1] https://cseweb.ucsd.edu/~jmcauley/datasets/amazon/links.html

[2] https://cseweb.ucsd.edu/~jmcauley/datasets/amazon_v2/

[3] https://github.com/kang205/SASRec

[4] https://grouplens.org/datasets/movielens/1m/

[5] https://webscope.sandbox.yahoo.com/

[6] https://github.com/YangZhengyi98/DreamRec/

### D.3 BASELINES OF SEQUENTIAL RECOMMENDATION

Traditional sequential recommenders:

• **GRU4Rec** (Hidasi et al., 2016) adopts RNNs to model user behavior sequences for session-based recommendations. Here, following the previopus work (Kang & McAuley, 2018; Yang et al., 2023b), we treat each user's interaction sequence as a session.

• **SASRec** (Kang & McAuley, 2018) adopts a directional self-attention network to model the user user behavior sequences.

• **Bert4Rec** (Sun et al., 2019) adapts the original text-based BERT model with the cloze objective for modeling user behavior sequences. We adopt the implementation of mask from (Ren et al., 2024b)

Contrastive learning based sequential recommenders:

• **CL4SRec** (Xie et al., 2022) incorporates the contrastive learning with the transformer-based sequential recommendation model to obtain more robust results. We adopt the implementation [7] from (Ren et al., 2024b).

Generative sequential recommenders:

• **TIGER**(Rajput et al., 2023) introduces codebook-based identifiers through RQ-VAE, which quantizes semantic information into code sequences for generative recommendation. Since the source code is unavailable, we implement it using the HuggingFace and Transformers APIs, following the original paper by utilizing T5 (Ni et al., 2022) as the backbone. For quantization, we employ FAISS (Johnson et al., 2019), which is widely used [8] in recent studies of recommendation (Hou et al., 2023).

DM-based sequential recommenders:

• **DiffRec** (Wang et al., 2023b) introduces the application of diffusion on user interaction vectors (i.e., multi-hot vectors) for collaborative recommendation, where "1" denotes a positive interaction and "0" indicates a potential negative interaction. We adopt the author's public implementation [9].

• **DreamRec** (Yang et al., 2023b) uses the historical interaction sequence as conditional guiding information for the diffusion model to enable personalized recommendations and utilize MSE as the training objective. We adopt the author's public implementation [10].

• **DiffuRec** (Li et al., 2024) introduces the DM to reconstruct target item embedding from a Transformer backbone with the user's historical interaction behaviors and utilize CE as the training objective. We adopt the author's public implementation [11].

Text-based sequential recommenders:

• **MoRec** (Yuan et al., 2023) utilizes item features from text descriptions or images, encoded using a text encoder or vision encoder, and applies dimensional transformation to match the appropriate dimension for recommendation. Here, we utilize the OpenAI-3-large embeddings, SASRec as backbone and transform the dimension to 64.

• **LLM2Bert4Rec** (Harte et al., 2023) proposes initializing item embeddings with textual embeddings. In our implementation, we use OpenAI-3-large embeddings, Bert4Rec as backbone and apply PCA to reduce the dimensionality to 64, as mentioned in the original paper.

Noablely, the inconsistent performance of Tiger and LLM2BERT4Rec with their origin paper is actually caused by the differences in evaluation settings. Both of these papers use the Leave-one-out evaluation setting, which differs from the User-split used in our work.

**Results of Other Backbone.** Here, we present a comparison of PreferDiff with other recommenders using a different backbone, namely GRU. As shown in Table 6, PreferDiff still outperforms DreamRec across all datasets, further validating its versatility. Empirically, we find that, unlike SASRec, which

---

[7]https://github.com/HKUDS/SSLRec/
[8]https://github.com/facebookresearch/faiss
[9]https://github.com/YiyanXu/DiffRec/
[10]https://github.com/YangZhengyi98/DreamRec/
[11]https://github.com/WHUIR/DiffuRec/

performs better with a Transformer than with GRU4Rec, PreferDiff performs better with GRU as the backbone on the Sports and Toys datasets compared to using a Transformer. This could be due to the relatively shallow Transformer used, making GRU easier to fit. More suitable network architectures for DM-based recommenders will be explored in future work.

Table 6: Comparison of the performance with sequential recommenders with GRU as backbone. The improvement achieved by PreferDiff is significant ($p$-value $\ll 0.05$).

| Model | Sports and Outdoors | | | | Beauty | | | | Toys and Games | | | |
|---|---|---|---|---|---|---|---|---|---|---|---|---|
| | R@5 | N@5 | R@10 | N@10 | R@5 | N@5 | R@10 | N@10 | R@5 | N@5 | R@10 | N@10 |
| GRU4Rec | 0.0022 | 0.0020 | 0.0030 | 0.0023 | 0.0093 | 0.0078 | 0.0102 | 0.0081 | 0.0097 | 0.0087 | 0.0100 | 0.0090 |
| SASRec | 0.0047 | 0.0036 | 0.0067 | 0.0042 | 0.0138 | 0.0090 | 0.0219 | 0.0116 | 0.0133 | 0.0097 | 0.0170 | 0.0109 |
| DreamRec | 0.0201 | 0.0147 | 0.0230 | 0.0165 | 0.0431 | 0.0290 | 0.0543 | 0.0321 | 0.0484 | 0.0343 | 0.0591 | 0.0382 |
| PreferDiff | 0.0216 | 0.0165 | 0.0250 | 0.0176 | 0.0451 | 0.0313 | 0.0590 | 0.0358 | 0.0530 | 0.0385 | 0.0623 | 0.0415 |

## D.4 LEAVE ONE OUT

**Evaluation.** The "leave-one-out" strategy is another widely adopted evaluation protocol in sequential recommendation. For each user's interaction sequence, the final item serves as the test instance, the penultimate item is reserved for validation, and the remaining preceding interactions are utilized for training. During testing, the ground-truth item of each sequence is ranked against a set of candidate items, allowing for a comprehensive assessment of the model's ranking capabilities. Performance is evaluated by computing ranking-based metrics over the test set, and the final reported result is the average metric across all users in the test set.

Table 7: Detailed Statistics of Datasets after Preprocessing in Leave-One-Out Setting.

| Datasets | Sports | Beauty | Toys | Automotive | Music | Office |
|---|---|---|---|---|---|---|
| #Sequences | 35,598 | 22,363 | 19,412 | 2,929 | 1,430 | 4,906 |
| #Items | 18,357 | 12,101 | 11,924 | 1,863 | 901 | 2,421 |
| #Interactions | 296,337 | 198,502 | 167,597 | 20,473 | 10,261 | 53,258 |
| Avg. Length | 8.32 | 8.87 | 8.63 | 6.99 | 7.17 | 10.86 |

**Datasets.** Except for the original three datasets (Sports, Toys and Beauty) in TIGER, we select three additional product review categories—namely, "Automotive", "Music Instrument" and "Office Product" from Amazon 2014 for a more comprehensive comparison. Here, we utilize the common five-core datasets, filtering out users and items with fewer than five interactions across all datasets.

**Baselines.** Here, we directly report baseline results (e.g., S$^3$-Rec (Zhou et al., 2020), P5 (Geng et al., 2022), FDSA (Hao et al., 2023)) from TIGER (Rajput et al., 2023) and evaluate DreamRec (Yang et al., 2023b) and the proposed PreferDiff.

**Results.** Tables 8 and Tables 9 present the performance of PreferDiff compared with six categories sequential recommenders. For breivty, R stands for Recall, and N stands for NDCG. The top-performing and runner-up results are shown in bold and underlined, respectively. "Improv" represents the relative improvement percentage of PreferDiff over the best baseline. We observe that in the leave-one-out setting, PreferDiff demonstrates competitive recommendation performance compared to the baselines. Specifically, on larger datasets (i.e., Sports and Beauty), PreferDiff performs on par with TIGER. However, on the Toys dataset and the three smaller datasets, PreferDiff achieves a significant lead.This may be due to PreferDiff adopting the same manner as DreamRec, where recommendation is not included in the training process. With a smaller number of items, this approach can result in more precise recommendation performance.

## D.5 GENERAL SEQUENTIAL RECOMMENDATION

**Pretraining Datasets.** Here, we introduce more details about Pretraining datasets. Following the previous work (Hou et al., 2022a; Li et al., 2023a), we select five different product reviews from Amazon 2018 (Ni et al., 2019), namely, "Automotive", "Cell Phones and Accessories", "Grocery and Gourmet Food", "Musical Instruments" and "Tools and Home Improvement", as pretraining datasets. "Cell Phones and Accessories" is selected as the validation dataset for early stopping when Recall@5

Table 8: Performance comparison on sequential recommendation under leave one out. The last row depicts % improvement with PreferDiff relative to the best baseline.

| Methods | Sports and Outdoors | | | | Beauty | | | | Toys and Games | | | |
|---|---|---|---|---|---|---|---|---|---|---|---|---|
| | R@5 | N@5 | R@10 | N@10 | R@5 | N@5 | R@10 | N@10 | R@5 | N@5 | R@10 | N@10 |
| P5 | 0.0061 | 0.0041 | 0.0095 | 0.0052 | 0.0163 | 0.0107 | 0.0254 | 0.0136 | 0.0070 | 0.0050 | 0.0121 | 0.0066 |
| Caser | 0.0116 | 0.0072 | 0.0194 | 0.0097 | 0.0205 | 0.0131 | 0.0347 | 0.0176 | 0.0166 | 0.0270 | 0.0141 | |
| HGN | 0.0189 | 0.0120 | 0.0313 | 0.0159 | 0.0325 | 0.0206 | 0.0540 | 0.0257 | 0.0266 | 0.0321 | 0.0497 | 0.0277 |
| GRU4Rec | 0.0129 | 0.0086 | 0.0204 | 0.0111 | 0.0164 | 0.0113 | 0.0283 | 0.0137 | 0.0137 | 0.0097 | 0.0176 | 0.0084 |
| BERT4Rec | 0.0115 | 0.0075 | 0.0191 | 0.0099 | 0.0263 | 0.0184 | 0.0407 | 0.0214 | 0.0170 | 0.0161 | 0.0310 | 0.0183 |
| FDSA | 0.0182 | 0.0128 | 0.0288 | 0.0156 | 0.0261 | 0.0201 | 0.0407 | 0.0228 | 0.0228 | 0.0150 | 0.0381 | 0.0199 |
| SASRec | 0.0233 | 0.0162 | 0.0412 | 0.0209 | 0.0462 | 0.0387 | 0.0605 | 0.0318 | 0.0463 | 0.0463 | 0.0675 | 0.0374 |
| S³-Rec | 0.0251 | 0.0161 | 0.0385 | 0.0204 | 0.0380 | 0.0244 | 0.0647 | 0.0327 | 0.0327 | 0.0294 | 0.0700 | 0.0376 |
| DreamRec | 0.0087 | 0.0071 | 0.0096 | 0.0075 | 0.0318 | 0.0257 | 0.0624 | 0.0273 | 0.0422 | 0.0347 | 0.0689 | 0.0362 |
| TIGER | 0.0264 | 0.0181 | 0.0400 | **0.0225** | 0.0454 | **0.0321** | 0.0648 | 0.0384 | 0.0521 | 0.0371 | 0.0712 | 0.0432 |
| **PreferDiff** | **0.0275** | **0.0190** | **0.0405** | 0.0218 | **0.0455** | 0.0317 | **0.0660** | **0.0388** | **0.0603** | **0.0403** | **0.0851** | **0.0483** |
| **Improve** | **4.16%** | **4.97%** | **1.25%** | -3.1% | **0.22%** | -1.25% | **1.85%** | **1.04%** | **15.73%** | **8.63%** | **19.52%** | **11.81%** |

Table 9: Performance comparison on sequential recommendation under leave one out. The last row depicts % improvement with PreferDiff relative to the best baseline.

| Methods | Automotive | | | | Music | | | | Office | | | |
|---|---|---|---|---|---|---|---|---|---|---|---|---|
| | R@5 | N@5 | R@10 | N@10 | R@5 | N@5 | R@10 | N@10 | R@5 | N@5 | R@10 | N@10 |
| DreamRec | 0.0543 | 0.0400 | 0.0683 | 0.0445 | 0.0622 | 0.0414 | 0.0783 | 0.0467 | 0.0523 | 0.0378 | 0.0699 | 0.0434 |
| TIGER | 0.0454 | 0.0290 | 0.0745 | 0.0383 | 0.0532 | 0.0358 | 0.0840 | 0.0456 | 0.0462 | 0.0299 | 0.0746 | 0.0390 |
| **PreferDiff** | 0.0649 | 0.0463 | 0.0864 | 0.0532 | 0.0650 | 0.0453 | 0.0874 | 0.0526 | 0.0538 | 0.0379 | 0.0850 | 0.0480 |
| **Improve** | **19.52%** | **15.75%** | **15.97%** | **19.55%** | **4.50%** | **9.42%** | **4.04%** | **12.63%** | **2.87%** | **0.26%** | **13.90%** | **10.60%** |

(i.e., **R@5**) shows no improvement for 20 consecutive epochs. The detailed statistics of each dataset used for pretraining are shown in Table 10. Clearly, the pretraining datasets have no domain overlap with the unseen datasets used in Section 4.2.

Table 10: Detailed Statistics of Pretraining Datasets.

| Datasets | Automotive | Phones | Tools | Instruments | Food |
|---|---|---|---|---|---|
| **#Sequences** | 193,651 | 157,212 | 240,799 | 27,530 | 127,496 |
| **#Items** | 18,703 | 12,839 | 22,854 | 2,494 | 11,778 |
| **#Interactions** | 806,939 | 544,339 | 1,173,154 | 110,151 | 623,940 |
| **Avg. Length** | 7.26 | 6.51 | 7.19 | 7.06 | 7.24 |

**Baselines.** Here, we introduce more details for baselines in General Sequential Recommendation tasks. Notably, for a fair comparison, we employ the `text-embedding-3-large` model (Liu et al., 2025a) from OpenAI (Neelakantan et al., 2022) as the text encoder instead of Bert (Devlin et al., 2019) in UniSRec and MoRec to convert identical item descriptions (e.g., title, category, brand) into vector representations, as it has been proven to deliver commendable performance in recommendation (Harte et al., 2023). Different of the Mixed-of-Experts (MoE) Whitening utilized in UniSRec, we employ identical ZCA-Whitening (Bell & Sejnowski, 1997) for the textual item embeddings for MoRec and Our proposed PreferDiff.

• **UniSRec** (Hou et al., 2022a) uses textual item embeddings from frozen text encoder and adapts to a new domain using an MoE-enhance adaptor. We adopt the author's public implementation [12].

• **MoRec** (Yuan et al., 2023) uses textual item embeddings from frozen text encoder and utilize dimension transformation technique. The architecture is the same as previously mentioned.

**Positive Correlation Between Training Data Scale and General Sequential Recommendation Performance.** Here, we explore how the scale of training data impacts the general sequential recommendation performance of PreferDiff-T. For brevity, we use the initials to represent each dataset. For example, "A" stands for Automotive, and "P" stands for Phones. "AP" indicates that the training data for pretraining includes both Automotive and Phones datasets' training set.

We observe that both NDCG and HR increase as the training data grows, indicating that PreferDiff-T can effectively learn general knowledge to model user preference distributions through pre-training on

---

[12]https://github.com/RUCAIBox/UniSRec

diverse datasets and transfer this knowledge to unseen datasets via advanced textual representations. Further studies can explore whether homogeneous datasets lead to greater performance improvements (e.g., whether Amazon Book data provides a larger boost for Goodreads compared to other datasets) and investigate the limits of data scalability for PreferDiff-T.

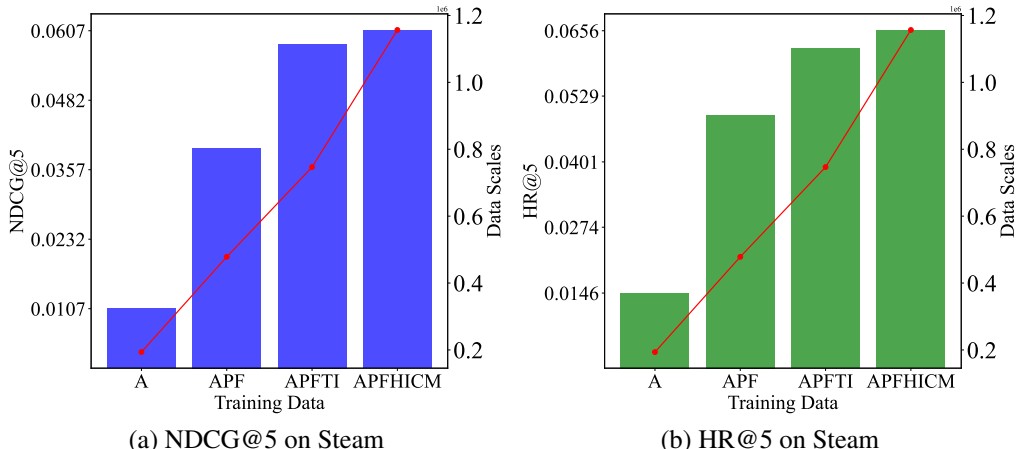

Figure 4: Positive Correlation Between Training Data Scale and General Sequential Recommendation Performance.

### D.6 HYPERPARAMETER SEARCH SPACE

Here, we introduce the hyperparamter search space for baselines and PreferDiff.

Table 11: Hyperparameters Search Space for Baselines.

| | Hyperparameter Seach Space |
|---|---|
| **GRU4Rec** | lr $\sim$ {1e-2, 1e-3, 1e-4, 1e-5}, weight decay=0 |
| **SASRec** | lr $\sim$ {1e-2, 1e-3, 1e-4, 1e-5}, weight decay=0 |
| **Bert4Rec** | lr $\sim$ {1e-2, 1e-3, 1e-4, 1e-5}, weight decay=0, mask probability$\sim$ {0.2,0.4,0.6,0.8} |
| **CL4SRec** | lr $\sim$ {1e-2, 1e-3, 1e-4, 1e-5}, weight decay=0, $\lambda\sim$ {0.1, 0.3, 0.5, 1.0, 3.0} |
| **DiffRec** | lr $\sim$ {1e-2, 1e-3, 1e-4, 1e-5}, weight decay=0, noise scale $\sim$ {1e-1, 1e-2, 1e-3, 1e-4, 1e-5}, T $\sim$ {2, 5, 20, 50, 100} |
| **DreamRec** | lr $\sim$ {1e-2, 1e-3, 1e-4, 1e-5}, weight decay=0, embedding size $\sim$ {64, 128, 256, 1024, 1536, 3072} , w $\sim$ {0, 2, 4, 6, 8, 10} |
| **DiffuRec** | lr $\sim$ {1e-2, 1e-3, 1e-4, 1e-5}, weight decay=0, embedding size $\sim$ {64, 128, 256, 1024, 1536, 3072} |
| **UniSRec** | lr $\sim$ {1e-2, 1e-3, 1e-4, 1e-5}, weight decay=0, $\lambda\sim$ {0.05, 0.1, 0.3, 0.5, 1.0, 3.0} |
| **TIGER** | lr $\sim$ {1e-2, 1e-3, 1e-4, 1e-5}, weight decay $\sim$ {0, 1e-1, 1e-2, 1e-3} |
| **MoRec** | lr $\sim$ {1e-2, 1e-3, 1e-4, 1e-5}, weight decay=0, text-encoder=text-embedding-3-large |
| **LLM2Bert4Rec** | lr $\sim$ {1e-2, 1e-3, 1e-4, 1e-5}, weight decay=0, text-encoder=text-embedding-3-large |
| **PreferDiff** | lr $\sim$ {1e-2, 1e-3, 1e-4, 1e-5}, $\lambda \sim$ {0.2, 0.4, 0.6, 0.8}, embedding size $\sim$ {64, 128, 256, 1024, 1536, 3072} , w $\sim$ {0, 2, 4, 6, 8, 10} |

Table 12: Best Hyperparameters for PreferDiff on Sports, Beauty, and Toys.

| Dataset | learning rate | weight decay | $\lambda$ | $w$ | embedding_size |
|---|---|---|---|---|---|
| **Sports** | 1e-4 | 0 | 0.4 | 2 | 3072 |
| **Beauty** | 1e-4 | 0 | 0.8 | 6 | 3072 |
| **Toys** | 1e-4 | 0 | 0.5 | 4 | 3072 |

## E HYPERPARAMETER ANALYSIS FOR PREFERDIFF

### E.1 THE NUMBER OF NEGATIVE SAMPLES FOR PREFERDIFF.

Here, we discuss the impact of the number of negative samples on PreferDiff. As shown in Figure 6, we observe that in cases where the number of items is relatively small (e.g., Beauty and Toys), 8

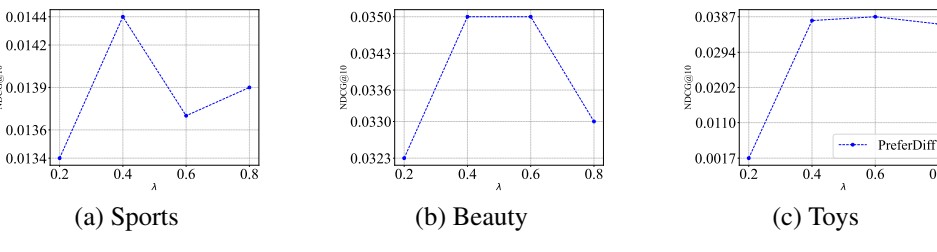

Figure 5: Effect of the $\lambda$ for PreferDiff.

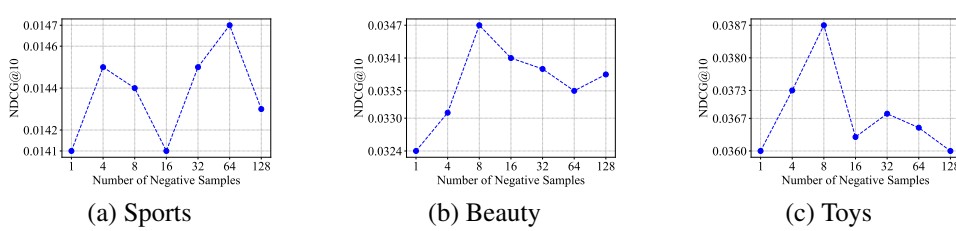

Figure 6: Effect of the Number of Negative Samples for PreferDiff.

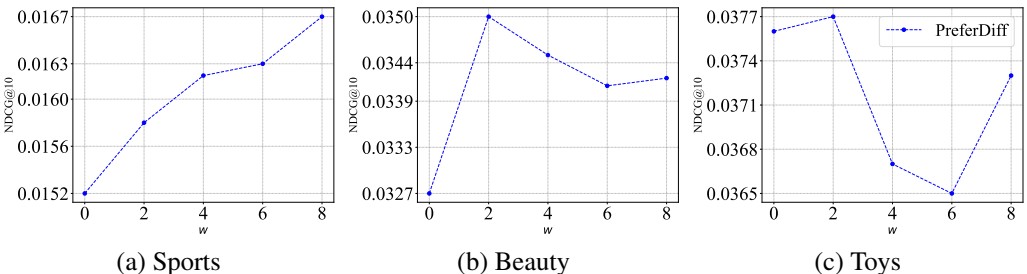

Figure 7: Effect of the $w$ for PreferDiff.

negative samples are sufficient. However, as the number of items increases, the required number of negative samples also grows (e.g., in Sports).

### E.2 IMPORTANCE OF GUIDANCE STRENGTH FOR PREFERDIFF

$w$ controls the weight of personalized guidance during the inference stage of PreferDiff. As shown in Figure 7, increasing $w$ can enhance recommendation performance. However, an excessively large $w$ may reduce the generalization capability of DMs, negatively impacting the recommender's performance. Therefore, we think setting $w \in [2, 4]$.

### E.3 DIFFERENT TEXT ENCODERS

**Obtaining Item Embedding from Advanced Text Encoder** Here, we introduce the process for obtaining item embeddings from current advanced text-encoders (Liu et al., 2025b). For encoder-based large language models, such as Bert (Devlin et al., 2019) and Robert (Liu et al., 2019), we leverage the final hidden state representation associated with the [CLS] token (Hou et al., 2024b). For convenient, we directly utilize the Sentence Transformers APIs [13]. As for other large language models, including T5 (Ni et al., 2022), Llama-7B (Touvron et al., 2023), Mistral-7B (Jiang et al., 2023), we utilize the output from the last transformer block corresponding to the final input token (Vaswani et al., 2017). Closed-source large language models like text-embedding-ada-v2 and text-embeddings-3-large, we obtain the item embeddings directly via OpenAI APIs [14] (Neelakantan et al., 2022).

---

[13] https://huggingface.co/sentence-transformers
[14] https://platform.openai.com/docs/guides/embeddings

Table 13: Comparison of the PreferDiff-T performance with different text-encoder.

| PreferDiff-T | Sports and Outdoors | | | | Beauty | | | | Toys and Games | | | |
|---|---|---|---|---|---|---|---|---|---|---|---|---|
| Text-Encoders | R@5 | N@5 | R@10 | N@10 | R@5 | N@5 | R@10 | N@10 | R@5 | N@5 | R@10 | N@10 |
| Bert | 0.0022 | 0.0020 | 0.0030 | 0.0023 | 0.0104 | 0.0128 | 0.0154 | 0.0148 | 0.0051 | 0.0022 | 0.0068 | 0.0044 |
| T5 | 0.0011 | 0.0009 | 0.0014 | 0.0011 | 0.0241 | 0.0198 | 0.0282 | 0.0212 | 0.0283 | 0.0240 | 0.0309 | 0.0248 |
| Robert | 0.0115 | 0.0098 | 0.0135 | 0.0102 | 0.0331 | 0.0256 | 0.0393 | 0.0276 | 0.0391 | 0.0303 | 0.0438 | 0.0319 |
| Mistral-7B | 0.0166 | 0.0130 | 0.0213 | 0.0146 | 0.0375 | 0.0287 | 0.0456 | 0.0312 | 0.0427 | 0.0328 | 0.0505 | 0.0353 |
| LLaMA-7B | 0.0171 | 0.0126 | 0.0205 | 0.0137 | 0.0402 | 0.0297 | 0.0483 | 0.0323 | 0.0397 | 0.0298 | 0.0494 | 0.0330 |
| OpenAI-Ada-V2 | 0.0160 | 0.0126 | 0.0183 | 0.0134 | 0.0407 | 0.0318 | 0.0469 | 0.0338 | 0.0396 | 0.0315 | 0.0467 | 0.0339 |
| OpenAI-3-large | 0.0182* | 0.0145* | 0.0222* | 0.0158* | 0.0429* | 0.0327* | 0.0532* | 0.0360* | 0.0460* | 0.0351* | 0.0525* | 0.0387* |

**Results.** Table 13 shows the PreferDiff-T employing different item embeddings encoded from text-encoders with varying parameter sizes and architectures. We can observe that

**Positive Correlation Between LLM Size and Recommendation Performance.** The results show that OpenAI-3-large outperforms all other models, indicating that larger language models (LLMs) yield better results in recommendation tasks. This is because larger models generate richer and more semantically stable embeddings, which improve PreferDiff's ability to capture user preferences. Thus, the larger the LLM, the better the embeddings perform within PreferDiff.

**High-Quality Embeddings Improve Generalization.** Models like Mistral-7B and LLaMA-7B, although smaller than OpenAI-3-large, still perform relatively well across metrics. This suggests that while model size is important, the quality of embeddings plays a crucial role. Especially in the Beauty, these models provide embeddings with sufficient semantic power to enhance recommendation quality.

## E.4 ANALYSIS OF LEARNED ITEM EMBEDDINGS

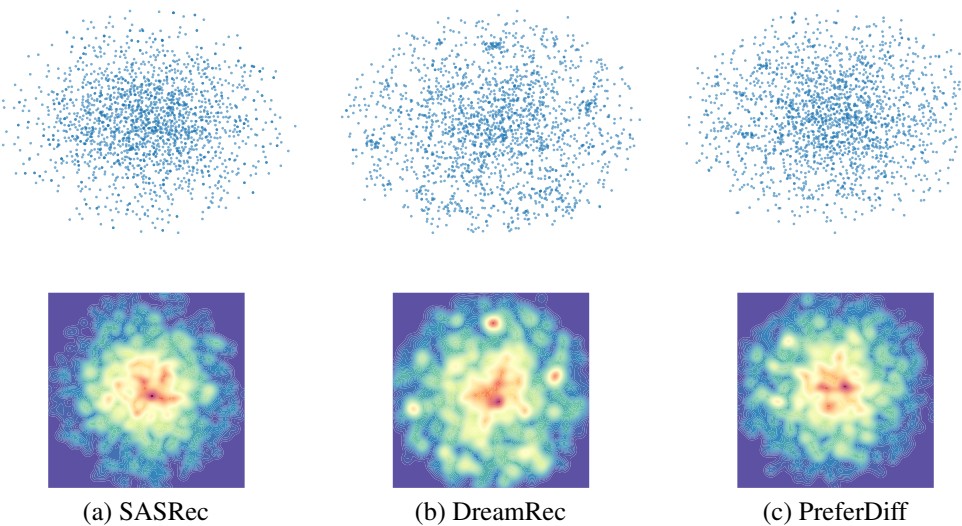

     (a) SASRec            (b) DreamRec            (c) PreferDiff

Figure 8: t-SNE Visualization and Gaussian Kernel Density Estimation of Learned Item Embeddings on Amazon Beauty.

To further analysis the item space learned by PreferDiff, we reduce the dimensionality of the learned item embeddings using T-SNE (Van der Maaten & Hinton, 2008; Liu et al., 2024a; Qian et al., 2024) [15] to visualize the underlying distribution of the item space learned by PreferDiff. Due to the large number of items in Amazon Beauty, we randomly select 2000 items as example. Then, we apply Gaussian kernel density estimation (Botev et al., 2010) [16] to analyze the density distribution of reduced item embeddings and visualize the results using contour plots. The red regions indicate areas where a

---

[15] https://scikit-learn.org/dev/modules/generated/sklearn.manifold.TSNE.html

[16] https://docs.scipy.org/doc/scipy/reference/generated/scipy.stats.gaussian_kde.html

high concentration of items is clustered. From figure 8, we can observe that comparing with SASRec, PreferDiff not only explores the item space more thoroughly (covering most regions). Comparing with DreamRec, PreferDiff exhibits a stronger clustering effect (with high-density regions concentrated in specific areas), better reflecting the similarities between items, result in better recommendation performance.

## F  DISCUSSION

### F.1  COMPARISON ON OTHER BACKGROUND DATASETS.

To further validate the effectiveness of PreferDiff, we include Yahoo! R1 (Music) as an additional dataset, along with two other commonly used datasets in sequential recommendation—Steam (Game) and ML-1M (Movie). These datasets provide a diverse set of user-item interaction patterns, allowing us to comprehensively evaluate the performance of our proposed PreferDiff.

We utilize the same data preprocessing technique and same evaluation setting as introduced in our paper for all three datasets, except Yahoo! R1. Due to its large size (over one million users), we are unable to provide results for the entire dataset during the rebuttal period. Instead, we randomly sampled 50,000 users for our experiments. We will include the full-scale results on Yahoo! R1 in the final revised version of the paper. The experimental results are shown in Table 14.

Table 14: Performance Comparison Across Background Datasets (Recall@5/NDCG@5)

| Datasets (Background) | Yahoo (Music) | Steam (Game) | ML-1M (Movie) |
|---|---|---|---|
| **GRU4Rec** | 0.0548 / 0.0491 | 0.0379 / 0.0325 | 0.0099 / 0.0089 |
| **SASRec** | 0.0996 / 0.0743 | 0.0695 / 0.0635 | 0.0132 / 0.0102 |
| **Bert4Rec** | 0.1028 / 0.0840 | 0.0702 / 0.0643 | 0.0215 / 0.0152 |
| **TIGIR** | 0.1128 / 0.0928 | 0.0603 / 0.0401 | 0.0430 / 0.0272 |
| **DreamRec** | 0.1302 / 0.1025 | 0.0778 / 0.0572 | 0.0464 / 0.0314 |
| **PreferDiff** | **0.1408 / 0.1106** | **0.0814 / 0.0680** | **0.0629 / 0.0439** |

We observe that the effectiveness of our proposed PreferDiff across datasets with different backgrounds are validated.

### F.2  COMPARISON ON VARIABLE USER HISTORY

we conduct additional experiments to evaluate the performance of PreferDiff under different maximum history lengths $\{10, 20, 30, 40, 50\}$. Notably, since the historical interaction sequences in the original three datasets (Sports, Beauty, Toys) are relatively short, with an average length of around 10, we select two additional commonly used datasets Kang & McAuley (2018); Sun et al. (2019), Steam and ML-1M, for further experiments. These datasets were processed and evaluated following the same evaluation settings and data preprocessing protocols in our paper, which is different from the leave-one-out split in Kang & McAuley (2018); Sun et al. (2019).

We choose another two datasets (Steam and ML-1M). The results are as follows:

Table 15: Performance Comparison on Steam Dataset (Recall@5/NDCG@5)

| Model | 10 | 20 | 30 | 40 | 50 |
|---|---|---|---|---|---|
| **SASRec** | 0.0698 / 0.0634 | 0.0676 / 0.0610 | 0.0663 / 0.0579 | 0.0668 / 0.0610 | 0.0704 / 0.0587 |
| **Bert4Rec** | 0.0702 / 0.0643 | 0.0689 / 0.0621 | 0.0679 / 0.0609 | 0.0684 / 0.0618 | 0.0839 / 0.0574 |
| **TIGIR** | 0.0603 / 0.0401 | 0.0704 / 0.0483 | 0.0676 / 0.0488 | 0.0671 / 0.0460 | 0.0683 / 0.0481 |
| **DreamRec** | 0.0778 / 0.0572 | 0.0746 / 0.0512 | 0.0741 / 0.0548 | 0.0749 / 0.0571 | 0.0846 / 0.0661 |
| **PreferDiff** | **0.0814 / 0.0680** | **0.0804 / 0.0664** | **0.0806 / 0.0612** | **0.0852 / 0.0643** | **0.0889 / 0.0688** |

From Table 15 and Table 16, we can observe that PreferDiff consistently outperforms other baselines across different lengths of user historical interactions.

Table 16: Performance Comparison on ML-1M Dataset (Recall@5/NDCG@5)

| Model | 10 | 20 | 30 | 40 | 50 |
|---|---|---|---|---|---|
| SASRec | 0.0201 / 0.0137 | 0.0242 / 0.0131 | 0.0306 / 0.0179 | 0.0217 / 0.0138 | 0.0205 / 0.0134 |
| Bert4Rec | 0.0215 / 0.0152 | 0.0265 / 0.0146 | 0.0331 / 0.0200 | 0.0248 / 0.0154 | 0.0198 / 0.0119 |
| TIGIR | 0.0451 / 0.0298 | 0.0430 / 0.0270 | 0.0430 / 0.0289 | 0.0364 / 0.0238 | 0.0430 / 0.0276 |
| DreamRec | 0.0464 / 0.0314 | 0.0480 / 0.0349 | 0.0514 / 0.0394 | 0.0497 / 0.0350 | 0.0447 / 0.0377 |
| **PreferDiff** | **0.0629 / 0.0439** | **0.0513 / 0.0365** | **0.0546 / 0.0408** | **0.0596 / 0.0420** | **0.0546 / 0.0399** |

### F.3 WHY DREAMREC AND PREFERDIFF ARE SENSITIVE TO THE EMBEDDING DIMENSION?

Here, we will try to explain the reason. Since there is no robust theoretical proof at this stage, we propose a hypothesis supported by simple theoretical reasoning and experimental validation.

We guess the challenge is inherent to the DDPM Ho et al. (2020) itself, as it is designed to be **variance-preserving** as introduced in the following diffusion models Song et al. (2021b). For one target item, the forward process formula with vector form is as follows:

**Forward Process:** $\mathbf{e}_0^t = \sqrt{\alpha_t}\mathbf{e}_0 + \sqrt{1 - \alpha_t}\epsilon$ Here, $\mathbf{e}_0 \in \mathbb{R}^{1 \times d}$ represents the target item embedding, $\mathbf{e}_0^t$ represents the noised target item embedding, $\alpha_t$ denotes the degree of noise added, and $\epsilon$ is the noise sampled from a standard Gaussian distribution.

Considering the whole item embeddings $\mathbf{E} \in \mathbb{R}^{N \times d}$, where $N$ represents the total number of items, we can rewrite the previous formula in matrix form as follows:

$$\mathbf{E}_0^t = \sqrt{\alpha_t}\mathbf{E}_0 + \sqrt{1 - \alpha_t}\epsilon$$

Then, we calculate the variance on both sides of the equation:

$$\text{Var}(\mathbf{E}_0^t) = \alpha_t \text{Var}(\mathbf{E}_0) + (1 - \alpha_t)\mathbf{I}$$

We can observe that the $\text{Var}(\mathbf{E}_0)$ is almost an identity matrix. This is relatively easy to achieve for data like images or text, as these data are fixed during the training process and can be normalized beforehand. However, in recommendation, the item embeddings are randomly initialized and updated dynamically during training. We empirically find that initializing item embeddings with a standard normal distribution is also a key factor for the success of DreamRec and PreferDiff. The results are shown as follows:

Table 17: Performance of Different Initialization methods on Various Datasets (Recall@5/NDCG@5).

| Embedding Initialization | Sports | Beauty | Toys |
|---|---|---|---|
| Uniform | 0.0039/0.0026 | 0.0013/0.0037 | 0.0015/0.0011 |
| Kaiming_Uniform | 0.0025/0.0019 | 0.0040/0.0027 | 0.0051/0.0028 |
| Kaiming_Normal | 0.0023/0.0021 | 0.0049/0.0028 | 0.0041/0.0029 |
| Xavier_Uniform | 0.0011/0.0007 | 0.0036/0.0021 | 0.0051/0.0029 |
| Xavier_Normal | 0.0014/0.0007 | 0.0067/0.0037 | 0.0042/0.0023 |
| **Standard Normal** | **0.0185/0.0147** | **0.0429/0.0323** | **0.0473/0.0367** |

**We can observe that the initializing item embeddings with a standard normal distribution is the key of success for Diffusion-based recommenders. This experiment validates the aforementioned hypothesis.**

Furthermore, we also calculate the final inferred item embeddings of DreamRec, PreferDiff, and SASRec. As shown in Figure 9, interestingly, we observe that the covariance matrices of the final item embeddings for DreamRec and PreferDiff are almost identity matrices, while SASRec does not exhibit this property. This indicates that DreamRec and PreferDiff rely on high-dimensional embeddings to adequately represent a larger number of items. The identity-like covariance structure suggests that diffusion-based recommenders distribute variance evenly across embedding dimensions, requiring more dimensions to capture the complexity and diversity of the item space effectively. This further validates our the hypothesis that maintaining a proper variance distribution of the item embeddings is crucial for the effectiveness of current diffusion-based recommenders.

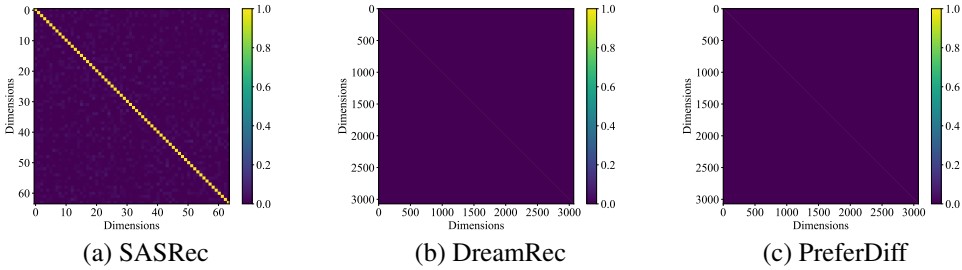

(a) SASRec            (b) DreamRec            (c) PreferDiff

Figure 9: Covariance Matrix Visualization of Learned Item Embeddings on Amazon Beauty.

We have tried several dimensionality reduction techniques (e.g., Projection Layers) and regularization techniques (e.g., enforcing the item embedding covariance matrix to be an identity matrix). However, these approaches empirically led to a significant drop in model performance.

We guess one possible solution to this issue is to explore the use of Variance Exploding (VE) diffusion models Song et al. (2021b). Unlike Variance Preserving diffusion models, which maintain a constant variance throughout the diffusion process, VE diffusion models increase the variance over time.

### F.4 TRAINING AND INFERENCE TIME COMPARISON

Table 18: Training and Inference Time Comparison for PreferDiff and Baselines.

| Dataset | Model | Training Time (s/epoch)/(s/total) | Inference Time (s/epoch) |
|---------|-------|-----------------------------------|--------------------------|
| **Sports** | SASRec | 2.67 / 35 | 0.47 |
| | Bert4Rec | 7.87 / 79 | 0.65 |
| | TIGIR | 11.42 / 1069 | 24.14 |
| | DreamRec | 24.32 / 822 | 356.43 |
| | PreferDiff | 29.78 / 558 | 6.11 |
| **Beauty** | SASRec | 1.05 / 36 | 0.37 |
| | Bert4Rec | 3.66 / 80 | 0.40 |
| | TIGIR | 5.41 / 1058 | 10.19 |
| | DreamRec | 14.78 / 525 | 297.06 |
| | PreferDiff | 18.05 / 430 | 3.80 |
| **Toys** | SASRec | 0.80 / 56 | 0.22 |
| | Bert4Rec | 3.11 / 93 | 0.23 |
| | TIGIR | 3.76 / 765 | 4.21 |
| | DreamRec | 15.43 / 552 | 309.45 |
| | PreferDiff | 16.07 / 417 | 3.29 |

In this subsection, we endeavor to illustrate the training and inference time comparison between PreferDiff and baseline methods, as efficiency is critically important for the practical application of recommenders in real-world scenarios. As shown in Table 18, Figure 10 and Figure 11, we can observe that

● In PreferDiff, thanks to our adoption of DDIM for skip-step sampling, requires less training time and significantly shorter inference time compared to DreamRec, another diffusion-based recommender.

● Compared to traditional deep learning methods like SASRec and Bert4Rec, PreferDiff has longer training and inference times but achieves much better recommendation performance.

● Furthermore, compared to recent generative recommendation methods, such as TIGIR, which rely on autoregressive models and use beam search during inference, PreferDiff also demonstrates shorter training and inference times, highlighting its efficiency and practicality in real-world scenarios.

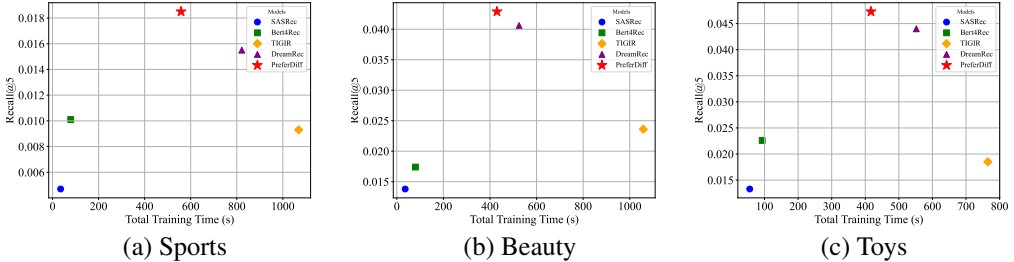

Figure 10: Recall@5 and Total Training Time for PreferDiff and Baselines.

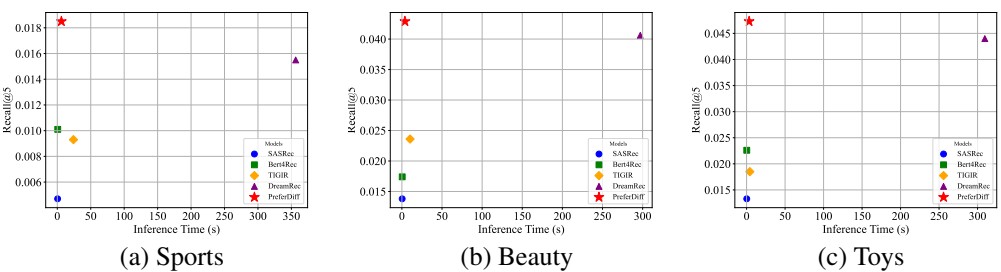

Figure 11: Recall@5 and Inference Time for PreferDiff and Baselines.

## F.5 TRADE-OFF BETWEEN RECOMMENDATION PERFORMANCE AND INFERENCE TIME

As introduced in Subsection F.4, PreferDiff demonstrates significantly lower inference time compared to DreamRec, averaging around 3 seconds per batch. However, this may still be unacceptable for real-time recommendation scenarios with strict latency constraints. **In this subsection, we aim to show how adjusting the number of denoising steps can effectively balance recommendation performance and inference time.**

As shown in Figure 12 and Table 19, we observe that by adjusting the number of denoising steps, PreferDiff can ensure practicality for real-time recommendation tasks. This flexibility allows for a trade-off between inference speed and recommendation performance, making PreferDiff adaptable to various latency constraints while maintaining competitive effectiveness.

## F.6 CONNECTION OF PREFERDIFF AND DPO

In Preferdiff, we aim to redesign a diffusion optimization objective that is specially tailored to model user preference distributions for personalized ranking. Therefore, we reformulate the classic recommendation objective Bayesian personalized ranking Rendle et al. (2009) to log-likelihood rankings

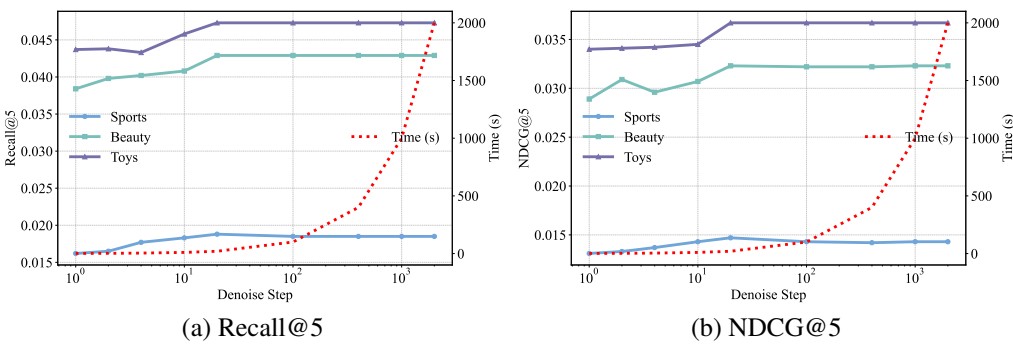

Figure 12: Relationship of Denoising Steps and Recommendation Performance.

Table 19: Adjusting Denoising Steps for Trade-Off Between Recommendation Performance and Inference Time.

| Datasets | Sports | Beauty | Toys |
|---|---|---|---|
| **SASRec (0.33s)** | 0.0047 / 0.0036 | 0.0138 / 0.0090 | 0.0133 / 0.0097 |
| **BERT4Rec (0.42s)** | 0.0101 / 0.0060 | 0.0174 / 0.0112 | 0.0226 / 0.0139 |
| **TIGER (12.85s)** | 0.0093 / 0.0073 | 0.0236 / 0.0151 | 0.0185 / 0.0135 |
| **DreamRec (320.98s)** | 0.0155 / 0.0130 | 0.0406 / 0.0299 | 0.0440 / 0.0323 |
| **PreferDiff (Denoising Step=1, 0.35s)** | 0.0162 / 0.0131 | 0.0384 / 0.0289 | 0.0437 / 0.0340 |
| **PreferDiff (Denoising Step=2, 0.43s)** | 0.0165 / 0.0133 | 0.0398 / 0.0309 | 0.0438 / 0.0341 |
| **PreferDiff (Denoising Step=4, 0.65s)** | 0.0177 / 0.0137 | 0.0402 / 0.0296 | 0.0433 / 0.0342 |
| **PreferDiff (Denoising Step=20, 3s)** | **0.0185 / 0.0147** | **0.0429 / 0.0323** | **0.0473 / 0.0367** |

Table 20: Comparison with DPO and Diffusion-DPO (Recall@5/NCDG@5)

| Models | Sports | Beauty | Toys |
|---|---|---|---|
| **DreamRec + DPO ($\beta = 1$)** | 0.0031 / 0.0015 | 0.0067 / 0.0053 | 0.0030 / 0.0022 |
| **DreamRec + DPO ($\beta = 5$)** | 0.0036 / 0.0026 | 0.0053 / 0.0034 | 0.0036 / 0.0023 |
| **DreamRec + DPO ($\beta = 10$)** | 0.0019 / 0.0011 | 0.0075 / 0.0056 | 0.0046 / 0.0034 |
| **DreamRec + Diffusion-DPO ($\beta = 1$)** | 0.0129 / 0.0101 | 0.0308 / 0.0244 | 0.0324 / 0.0261 |
| **DreamRec + Diffusion-DPO ($\beta = 5$)** | 0.0132 / 0.0113 | 0.0321 / 0.0251 | 0.0340 / 0.0272 |
| **DreamRec + Diffusion-DPO ($\beta = 10$)** | 0.0133 / 0.0115 | 0.0281 / 0.0223 | 0.0345 / 0.0281 |
| **PreferDiff** | **0.0185 / 0.0147** | **0.0429 / 0.0323** | **0.0473 / 0.0367** |

which meet the requirement of generative modeling in diffusion models. We are also surprisingly and delightedly discovering that the one-negative-sample version of PreferDiff's formulation, $\mathcal{L}_{\text{BPR-Diff}}$, is indeed related to the recent well-known DPO Rafailov et al. (2023) which stems from Reinforcement Learning with Human Feedback, as you have mentioned. To further validate the rationality of our proposed $\mathcal{L}_{\text{BPR-Diff}}$, we intentionally aligned some aspects of our final formulation with DPO in terms of mathematical expression.

However, there are significant distinctions between PreferDiff and DPO.

•First, PreferDiff is an optimization objective specifically tailored to model user preferences in diffusion-based recommenders. It is designed to align with the unique characteristics of the diffusion process, ensuring its effectiveness in recommendation tasks. We also replace the MSE loss with Cosine loss

• Second, unlike DPO and Diffusion-DPO Wallace et al. (2024), PreferDiff incorporates multiple negative samples and proposes a theoretically guaranteed, efficient strategy to reduce the computational overhead of denoising caused by the increased number of negative samples in diffusion models. This innovation allows PreferDiff to scale effectively while maintaining high performance, making it well-suited for large-negative-sample scenarios in recommendation tasks.

• Third, unlike DPO and Diffusion-DPO, PreferDiff is utilized in an end-to-end manner without relying on a reference model. In contrast, DPO and Diffusion-DPO require a two-stage process, where the first step involves training a reference model. This significantly increases training overhead, which is often unacceptable in practical recommendation scenarios.

To further validate the aforementioned distinctions, we conduct experiments on three datasets using DPO and Diffusion-DPO. Specifically, we select $\beta$, a crucial hyperparameter in DPO, with values of 1, 5, and 10, and integrate it with DreamRec for a fair comparison. The results are shown in Table 20

We can observe that PreferDiff outperforms DPO and Diffusion-DPO by a large margin on all three datasets. This further validates the effectiveness of our proposed PreferDiff, demonstrating that it is specifically tailored to model user preferences in diffusion-based recommenders.

