# OpenReview forum: "Preference Diffusion for Recommendation"
_ICLR.cc/2025/Conference — ICLR 2025 Poster_

### Official Review · Reviewer_den3 · 2024-10-28

**Soundness:** 3
**Presentation:** 3
**Contribution:** 3
**Rating:** 6
**Confidence:** 3

**Summary:**

This paper proposes a novel recommendation approach, PreferDiff, that optimizes diffusion models specifically for personalized ranking. By reformulating BPR as a log-likelihood objective and incorporating multiple negative samples, PreferDiff enhances the model’s ability to capture nuanced user preferences and better distinguish between positive and negative interactions. The authors demonstrate PreferDiff’s effectiveness through extensive evaluations, showing improved performance in sequential recommendation tasks, faster convergence, and superior handling of complex user preference distributions.

**Strengths:**

1. The paper introduces a unique objective tailored for DM-based recommenders by transforming BPR into a log-likelihood ranking format. This innovation effectively uses the generative capacity of DMs in recommendation tasks.

2. The model incorporates a balance between generative modeling and preference learning, which enhances stability and performance.

3. The model's gradient-weighted approach to negative samples allows it to focus on challenging cases where negative samples are mistakenly ranked high. This focus on "hard negatives" is particularly valuable in recommendation.

**Weaknesses:**

1. PreferDiff can be very slow in both the training and inference stages. It would be useful if the authors could show the efficiency-effectiveness comparison between PreferDiff and other baselines in a 2-D figure.

2. As shown in Appendix D, the maximum length of interaction history is small (i.e., 10 or 20) following prior works, however, as we know that users may generally have a much longer interaction history. Is this a general limitation of DM-based recommenders that they are too expensive to train on larger sequences? What would PreferDiff perform if training with a longer sequence?

**Questions:**

Could you share more details on why high embedding dimensions are necessary for PreferDiff’s performance? Have you experimented with any regularization techniques or embedding pruning methods to mitigate this dependence?

---

> ### Author Response · Authors · 2024-11-23
> **Response to Reviewer den3 - Part (1/4)**
>
> Thank you for your insightful review and for highlighting the strengths of our work, as well as your deep understanding of PreferDiff. We truly appreciate the thoughtful feedback and the suggestions you raised. These points are essential and have prompted us to carefully address them, significantly improving the quality of our paper.
>
> > **Comment 1: Training and Inference Cost Concerns** 一一  "PreferDiff can be very slow in both the training and inference stages. It would be useful if the authors could show the efficiency-effectiveness comparison between PreferDiff and other baselines in a 2-D figure."
>
>
> Thank you for your valuable questions. We completely understand your concerns about the efficiency of our proposed PreferDiff. To address your concerns more thoroughly, we firstly provide comparisons of training time and inference time between PreferDiff and other baselines.
>
>
>
> **Table 1: Comparison of Training Time and Inference Times.**
>
>
> | Dataset | Model      | Training Time (s/epoch)/(s/total) | Inference Time (s/epoch) |
> |---------|------------|-----------------------------------|--------------------------|
> | Sports  | SASRec     | 2.67 / 35                        | 0.47                     |
> |         | Bert4Rec   | 7.87 / 79                        | 0.65                     |
> |         | TIGIR      | 11.42 / 1069                     | 24.14                    |
> |         | DreamRec   | 24.32 / 822                      | 356.43                   |
> |         | PreferDiff | 29.78 / 558                      | 6.11                     |
> | Beauty  | SASRec     | 1.05 / 36                        | 0.37                     |
> |         | Bert4Rec   | 3.66 / 80                        | 0.40                     |
> |         | TIGIR      | 5.41 / 1058                      | 10.19                    |
> |         | DreamRec   | 15 / 525                         | 297.06                   |
> |         | PreferDiff | 18 / 430                         | 3.80                     |
> | Toys    | SASRec     | 0.80 / 56                        | 0.22                     |
> |         | Bert4Rec   | 3.11 / 93                        | 0.23                     |
> |         | TIGIR      | 3.76 / 765                       | 4.21                     |
> |         | DreamRec   | 15.43 / 552                      | 309.45                   |
> |         | PreferDiff | 16.07 / 417                      | 3.29                     |
>
> **Results**. We can observe that
>
> $\bullet$ Thanks to our adoption of DDIM for skip-step sampling, requires less training time and significantly shorter inference time compared to DreamRec, another diffusion-based recommender.
>
> $\bullet$ While PreferDiff incurs longer training and inference times than traditional deep learning models like SASRec and Bert4Rec, it achieves much better recommendation performance.
>
> $\bullet$ Compared to recent generative methods like TIGIR, which rely on autoregressive models and beam search, PreferDiff demonstrates shorter training and inference times, emphasizing its practicality for real-world scenarios.
>
> Following your suggestions, **we have plotted 2-D figures to illustrate the training time, testing time, Recall@5, and NDCG@5 on the three datasets—Sports, Beauty, and Toys in the Appendix F.3 of revision paper.**
> These visualizations help us provide a clearer understanding of the trade-offs and performance metrics.
> Thanks for your suggestion.

---

> ### Author Response · Authors · 2024-11-23
> **Response to Reviewer den3 - Part (2/4)**
>
> > **Comment 2: Length of User Interaction History** 一一  "As shown in Appendix D, the maximum length of interaction history is small (i.e., 10 or 20) following prior works, however, as we know that users may generally have a much longer interaction history. Is this a general limitation of DM-based recommenders that they are too expensive to train on larger sequences? What would PreferDiff perform if training with a longer sequence?"
>
>
> Thank you for your valuable suggestion. In our experiments, for fairness, we followed the experimental settings of previous diffusion-based recommenders (i.e., DreamRec), where, as you mentioned, the maximum length of interaction history is small (i.e., 10 or 20).
>
> We fully understand your concern about limitation of DM-based recommenders that they are too expensive to train on larger sequences. Based on your suggestion, we **conduct additional experiments** to evaluate the performance of PreferDiff under different maximum history lengths (10, 20, 30, 40, 50). Notably, since the historical interaction sequences in the original three datasets (Sports, Beauty, Toys) are relatively short, with an average length of around 10, we select two additional commonly used datasets, Steam and ML-1M [1][2], for further experiments. These datasets were processed and evaluated following the same evaluation settings and data preprocessing protocols in our paper, which is different from the leave-one-out split settings in their original papers [1][2].
> The results are as follows:
>
> **Table 2: Recommendation Performance with varied length of user history on Steam.**
>
> | Model (Recall@5/NDCG@5)      | 10 | 20 | 30 | 40| 50 |
> |------------|-----------------------|-----------------------|-----------------------|-----------------------|-----------------------|
> | **SASRec**     | 0.0698 / 0.0634      | 0.0676 / 0.0610      | 0.0663 / 0.0579      | 0.0668 / 0.0610      | 0.0704 / 0.0587      |
> | **Bert4Rec**   | 0.0702 / 0.0643      | 0.0689 / 0.0621      | 0.0679 / 0.0609      | 0.0684 / 0.0618      | 0.0839 / 0.0574      |
> | **TIGIR**      | 0.0603 / 0.0401      | 0.0704 / 0.0483      | 0.0676 / 0.0488      | 0.0671 / 0.0460      | 0.0683 / 0.0481      |
> | **DreamRec**   | 0.0778 / 0.0572      | 0.0746 / 0.0512      | 0.0741 / 0.0548      | 0.0749 / 0.0571      | 0.0846 / 0.0661      |
> | **PreferDiff** | **0.0814 / 0.0680**      | **0.0804 / 0.0664**      | **0.0806 / 0.0612**      | **0.0852 / 0.0643**      | **0.0889 / 0.0688**      |
>
>
>
> **Table 3: Recommendation Performance with varied length of user history on ML-1M.**
>
> | Model (Recall@5/NDCG@5)      | 10                  | 20                  | 30                  | 40                  | 50                  |
> |------------------------------|---------------------|---------------------|---------------------|---------------------|---------------------|
> |**SASRec**                       | 0.0201 / 0.0137    | 0.0242 / 0.0131    | 0.0306 / 0.0179    | 0.0217 / 0.0138    | 0.0205 / 0.0134    |
> | **Bert4Rec**                     | 0.0215 / 0.0152    | 0.0265 / 0.0146    | 0.0331 / 0.0200    | 0.0248 / 0.0154    | 0.0198 / 0.0119    |
> | **TIGIR**                        | 0.0451 / 0.0298    | 0.0430 / 0.0270    | 0.0430 / 0.0289    | 0.0364 / 0.0238    | 0.0430 / 0.0276    |
> | **DreamRec**                     | 0.0464 / 0.0314    | 0.0480 / 0.0349    | 0.0514 / 0.0394    | 0.0497 / 0.0350    | 0.0447 / 0.0377    |
> | **PreferDiff**                   | **0.0629 / 0.0439**    | **0.0513 / 0.0365**    | **0.0546 / 0.0408**    | **0.0596 / 0.0420**    | **0.0546 / 0.0399**    |
>
> **Results**.
> We can observe that PreferDiff consistently outperforms other baselines across different lengths of user historical interactions.
> We **incorporate the this discussion in Appendix F.2** of revision paper.
>
>
> [1] Kang, Wang-Cheng, and Julian McAuley. "Self-attentive sequential recommendation." 2018 IEEE international conference on data mining (ICDM). IEEE, 2018.
>
>
> [2] Sun, Fei, et al. "BERT4Rec: Sequential recommendation with bidirectional encoder representations from transformer." Proceedings of the 28th ACM international conference on information and knowledge management. 2019.

---

> ### Author Response · Authors · 2024-11-23
> **Response to Reviewer den3 - Part (3/4)**
>
> > **Comment 3: Embedding Dimension Sensitivity** 一一   "Could you share more details on why high embedding dimensions are necessary for PreferDiff’s performance? Have you experimented with any regularization techniques or embedding pruning methods to mitigate this dependence?"
>
>
> Thank you for your valuable question. We also believe that understanding why diffusion-based recommenders like DreamRec and PreferDiff require high-dimensional item embeddings is both important and meaningful.
>
> Next, we will try to explain this in detail. If you still have any questions or concerns afterward, please feel free to ask further!
>
> Here, we guess the challenge is inherent to the DDPM [1] itself, as it is designed to be **variance-preserving** as introducd in the following diffusion models [2].  For one target item, the forward process formula with vector form is as follows:
>
> **Foward Process:** $\mathbf{e}_{0}^{t}=\sqrt{\alpha_t}\mathbf{e}_0+\sqrt{1-\alpha_t}\epsilon$
> Here, $\mathbf{e}_0 \in \mathbb{R}^{1 \times d}$ represents the target item embedding, $\mathbf{e}_0^t$ represents the noised target item embedding, $\alpha_t$ denotes the degree of noise added, and $\epsilon$ is the noise sampled from a standard Gaussian distribution.
>
>  Considering the whole item embeddings $\mathbf{E} \in \mathbb{R}^{N \times d}$, where $N$ represents the total number of items, we can rewrite the previous formula in matrix form as follows:
>
> **Foward Process:** $\mathbf{E}_{0}^{t}=\sqrt{\alpha_t}\mathbf{E}_0+\sqrt{1-\alpha_t}\epsilon$
>
> Then, we calculate the variance on both sides of the equation:
>
> $\text{Var}(\mathbf{E}_{0}^{t})=\alpha_t\text{Var}(\mathbf{E}_0)+(1-\alpha_t)\mathbf{I}$
>
> We can observe that the $\text{Var}(\mathbf{E}_0)$ is almost an identity matrix. This is relatively easy to achieve for data like images or text, as these data are fixed during the training process and can be normalized beforehand. However, in recommendation, the item embeddings are randomly initialized and updated dynamically during training. We empirically find that initializing item embeddings with a standard normal distribution is also a key factor for the success of DreamRec and PreferDiff.  The results are shown as follows:
>
>
>
> **Table 4: Effect of Different Initilization Methods**
>
> | PreferDiff (Recall@5/NDCG@5) | Uniform           | Kaiming_Uniform    | Kaiming_Normal     | Xavier_Uniform     | Xavier_Normal      | Standard Normal     |
> |------------------------------|-------------------|--------------------|--------------------|--------------------|--------------------|---------------------|
> | **Sports**                       | 0.0039/0.0026     | 0.0025/0.0019      | 0.0023/0.0021      | 0.0011/0.0007      | 0.0014/0.0007      | **0.0185/0.0147**       |
> | **Beauty**                       | 0.0013/0.0037     | 0.0040/0.0027      | 0.0049/0.0028      | 0.0036/0.0021      | 0.0067/0.0037      | **0.0429/0.0323**       |
> | **Toys**                         | 0.0015/0.0011     | 0.0051/0.0028      | 0.0041/0.0029      | 0.0051/0.0029      | 0.0042/0.0023      | **0.0473/0.0367**       |
>
> **Results**. We can observe that the initializing item embeddings with a standard normal distribution is the key of success for Diffusion-based recommenders. **This experiment validates the aforementioned hypothesis.**

---

> ### Author Response · Authors · 2024-11-23
> **Response to Reviewer den3 - Part (4/4)**
>
> Furthermore, we also calculate the final inferred item embeddings of DreamRec, PreferDiff, and SASRec. Interestingly, we observed that the covariance matrices of the final item embeddings for DreamRec and PreferDiff are almost identity matrices, while SASRec does not exhibit this property.
>
> This indicates that DreamRec and PreferDiff rely on high-dimensional embeddings to adequately represent a larger number of items. The identity-like covariance structure suggests that  diffusion-based recommenders distribute variance evenly across embedding dimensions, requiring more dimensions to capture the complexity and diversity of the item space effectively.
>
> This further validates our the hypothesis that maintaining a proper variance distribution of the item embeddings is crucial for the effectiveness of current diffusion-based recommenders.
>
> We have tried several dimensionality reduction techniques (e.g., Projection Layers) and regularization techniques (e.g., enforcing the item embedding covariance matrix to be an identity matrix). However, these approaches empirically led to a significant drop in model performance.
>
> **Possible Solution** We guess one possible solution to this issue is to explore the use of Variance Exploding (VE) diffusion models [2]. Unlike Variance Preserving diffusion models, which maintain a constant variance throughout the diffusion process, VE models increase the variance over time.
>
> We **added all the discusion in the Appendix F.3** of revision paper.
>
> [1] Ho, Jonathan, Ajay Jain, and Pieter Abbeel. "Denoising diffusion probabilistic models." Advances in neural information processing systems 33 (2020): 6840-6851.
>
> [2] Song, Yang, et al. "Score-Based Generative Modeling through Stochastic Differential Equations." International Conference on Learning Representations.
>
> **If there are any issues, please feel free to reply, and we will respond promptly.**

---

> ### Author Response · Authors · 2024-11-25
> **looking forward to your reply**
>
> Dear Reviewer den3,
>
> Thank you again for your constructive comments. We have carefully addressed your concerns by providing a detailed comparison of training and inference times between our methods and the baselines, using tables and 2D figures as per your suggestions. Furthermore, we conducte experiments on two datasets (Steam and ML-1M) to demonstrate the performance of our approach under different lengths of user interaction history. Additionally, we provide theoretical and experimental evidence on the sensitivity of diffusion models to embedding dimensions. All these revisions have been incorporated into the manuscript.
>
> We look forward to further discussion with you and would greatly appreciate your positive feedback on our rebuttal.
>
> Best regards,
>
> The Authors

---

> ### Author Response · Authors · 2024-11-29
> **Kind Reminder**
>
> Dear Reviewer den3,
>
> As the deadline approaches, we sincerely hope to address your concerns and discuss the rebuttal with you further. If you have any questions, please feel free to ask directly!
>
> Best regards,
>
> Authors

---

> ### Author Response · Authors · 2024-12-03
> **Kind Reminder**
>
> Dear Reviewer den3,
>
> As today marks the final day of the discussion period, we sincerely hope to address your concerns thoroughly and engage in further discussion regarding our rebuttal. Should you have any questions or require additional clarification, please don’t hesitate to reach out.
> Moreover, if you find our responses satisfactory, we would greatly appreciate it if you could kindly consider the possibility of revising your score. Thank you once again for your valuable feedback and thoughtful suggestions.
>
> Best regards,
>
> Authors

---

### Official Review · Reviewer_SvZm · 2024-10-31

**Soundness:** 3
**Presentation:** 3
**Contribution:** 2
**Rating:** 5
**Confidence:** 4

**Summary:**

This paper introduces PreferDiff, a model designed to capture user preferences using diffusion model-based recommenders. PreferDiff primarily focuses on enhancing loss functions. The core contributions are twofold: First, the authors incorporate negative samples into the model training process through the use of ranking loss. Second, they transform the loss function in DM-based methods from rating estimation of samples to distribution estimation, formulated as the L_{BPR-Diff} loss, which is approximated by an upper bound. Experiments are conducted across multiple dataset settings to validate the approach.

**Strengths:**

1. The paper is well-written and easy to follow. It is well-structured and provides a detailed description of the experimental setup, which will aid the community in reimplementation.
2. The authors conducted experiments under various settings to validate the effectiveness of their method.

**Weaknesses:**

1. The evaluation difference between the full-ranking approach and the leave-one-out method is not clearly described. There is no mention of a full-ranking approach in [1]. The 'leave-one-out' evaluation setting is the most mainstream approach. The authors need to provide a justification for using the full-ranking manner as the primary evaluation setting in the main text. Additionally, do these two evaluation methods affect the ranking of different approaches?
2. Performance of other recommenders in Tab.1 is confusing. For TIGER, the performance in Tab.1 is much lower than the performance reported by the original paper. The authors should explain the reasons behind these performance gap. One reason maybe contribute to the performance gap is the ID quantifer. Authors take the PQcode of VQREC[2] as the ID quantifier instead of the RQVAE used in TIGER[3]。However, the authors should clearly specify whether these performance metrics are reproduced results or those reported in the original paper. For LLM2BERT4Rec, the performance on Beauty dataset is also inconsistent with the performance in original paper.
3. The explanation in Section 3.2 is somewhat unconvincing. L_{BPR-Diff} is essentially a ranking loss in metric learning.
4. For generative models, timestamps is crucial. The authors should conduct an ablation study on timestamps to evaluate their impact on the generation of positive and negative samples. Additionally, given that generative models are typically time-consuming, it would be beneficial to assess how PreferDiff performs in terms of inference speed compared to other sequence models. Specifically, can it achieve a favorable trade-off between hit rate/NDCG and speed?

[1] Generate What You Prefer: Reshaping Sequential Recommendation via Guided Diffusion.
[2] Learning Vector-Quantized Item Representation for Transferable Sequential Recommenders
[3] Recommender Systems with Generative Retrieval

**Questions:**

see in Weaknesses.
I am open to revising my score based on further clarifications or additional information provided by the authors during the rebuttal process.

---

> ### Author Response · Authors · 2024-11-23
> **Response to Reviewer SvZm - Part (1/5)**
>
> Thank you for your insightful review and for highlighting the strengths of our work, as well as your deep understanding of PreferDiff. We truly appreciate the thoughtful feedback and the suggestions you raised. These points are essential and have prompted us to carefully address them, significantly improving the quality of our paper.
>
> > **Comment 1:  Clarification of Evaluation Settings** 一一  "The evaluation difference between the full-ranking approach and the leave-one-out method is not clearly described. There is no mention of a full-ranking approach in [1]. The 'leave-one-out' evaluation setting is the most mainstream approach. The authors need to provide a justification for using the full-ranking manner as the primary evaluation setting in the main text. Additionally, do these two evaluation methods affect the ranking of different approaches?"
>
> Thank you for your valuable question. Following your suggestion, we have revised the paper to provide a clearer description of evaluation metrics and justify our choice of evaluation settings.
>
> We first want to clarify the full-ranking evaluation in recommendation.
> The standard full-ranking evalution implies that, for each user, the candidate set includes the entire item space without any special selection.
> We have added and modified the description of full-ranking to the revised paper for better clarity.
>
> Then, we justify our choice of dataset settings for evaluation.
> We use both user-split and leave-one-out settings for a comprehensive comparison, and both are evaluated in a full-ranking manner:
>
> 1. **User-split:** This splits the dataset by users, meaning that the user sequences in the validation and test datasets are unseen during training. The last item in each sequence is considered the ground truth.
>
> 2. **Leave-one-out:** In this method, the second-to-last interaction of each user is used for the validation set, and the last interaction is used for the test set.
>
> The user-split setting is more challenging because the test sequences are entirely unseen during training, potentially making them out-of-distribution.
> As you noted, the choice of evaluation setting can affect the ranking of different recommenders.
> PreferDiff is evaluated under both settings for a comprehensive comparison.
> Table 1 in the main paper follows the evaluation protocol of DreamRec, demonstrating PreferDiff’s effectiveness under user-split settings.
> Tables 8 and 9 in the Appendix follow the leave-one-out protocol used in TIGIR. Baseline results are reused for a fair comparison and to provide more insights.
> Across both settings, PreferDiff consistently outperforms state-of-the-art baselines, validating its robustness and generalizability.
>
>
>
> > **Comment 2:  Results Explaination** 一一"Performance of other recommenders in Tab.1 is confusing. For TIGER, the performance in Tab.1 is much lower than the performance reported by the original paper. For LLM2BERT4Rec, the performance on Beauty dataset is also inconsistent with the performance in original paper. The authors should explain the reasons behind these performance gap."
>
> As mentioned above, the inconsistent performance of Tiger and LLM2BERT4Rec is actually caused **by the differences in evaluation settings.**
> Both of these papers use the Leave-one-out evaluation method, which differs from the User-split used in the table 1.
> A comparison under leave-one-out setting could be found in Table 8 & 9.
> **We have revised our submission in line 1501-1503 for better clarification.**

---

> ### Author Response · Authors · 2024-11-23
> **Response to Reviewer SvZm - Part (2/5)**
>
> > **Comment 3:  Clarification of ID Quantifer** 一一  "One reason maybe contribute to the performance gap is the ID quantifer. Authors take the PQcode of VQREC[2] as the ID quantifier instead of the RQVAE used in TIGER[3]."
>
> Thanks for your careful reading.
> In our implementation of TIGER, as shown in Appendix line 1475-1480, "For quantization, we employ FAISS, which is widely used in recent studies of recommendation." Therefore, we utilize RQVAE, the same as in the original TIGER paper, rather than the PQ code utilized in VQRec.
>
> We fully agree that the quality of the codebook significantly impacts TIGER's performance, and RQVAE training involves many tricks and details, such as using hierarchical k-means for initialization.
> To ensure more stable performance, we directly use the `faiss.IndexResidualQuantizer` API from the FAISS library, which has been applied in codebook-related recommenders, including VQRec.
>
> We want to emphasize that our reproduced TIGER achieves comparable or even slightly better results than those reported in the original paper on the three datasets (Sports, Beauty, and Toys) under the leave-one-out setting with T5. Please see the results below.
>
> **Table 1: Comparison of Original TIGIR and Our Reproduced Version**
>
> | Model                    | Recall@5/NDCG@5 (Sports) | Recall@5/NDCG@5 (Beauty) | Recall@5/NDCG@5 (Toys) |
> |--------------------------|--------------------------|--------------------------|-------------------------|
> | **TIGER (Original Paper)**   | 0.0264 / 0.0181         | 0.0454 / 0.0321         | 0.0521 / 0.0371         |
> | **TIGER (Reproduce)**        | 0.0245 / 0.0154         | 0.0447 / 0.0312         | 0.0544 / 0.0397         |
>
>
> **Results**. This shows the effectiveness of our implementation while maintaining consistency with the original paper.
> We also find that the weight decay significantly affects TIGER's performance across different datasets, requiring careful hyperparameter tuning.
>
> To further validate the effectiveness of the proposed PreferDiff, we evaluate it under the leave-one-out setting, and the results are shown in Table 8 and Table 9. As observed, PreferDiff consistently achieves better recommendation performance even under this evaluation setting.
>
> In our evaluation setting, namely User-split, we surprisingly find that TIGER does not perform well, despite our careful tuning of the weight decay. We hypothesize that TIGER, which splits an item into four semantic IDs using a codebook, is more fine-grained, therefore, requires a larger amount of user history and stable user history distribution during inference to ensure accurate recommendations.
> We are not sure and just guess that this fine-grained nature of TIGER's codebook-based method may be a potential drawback, as it could fail when the user's history distribution is out-of-distribution.
>
> [1] Rajput, Shashank, et al. "Recommender systems with generative retrieval." Advances in Neural Information Processing Systems 36 (2023): 10299-10315.
>
> [2] Hou, Yupeng, et al. "Learning vector-quantized item representation for transferable sequential recommenders." Proceedings of the ACM Web Conference 2023. 2023.
>
> > **Comment 4:  Result Source** 一一  "However, the authors should clearly specify whether these performance metrics are reproduced results or those reported in the original paper."
>
>
> Thank you for your valuable feedback! **We have revised our submission and highlighted in the line 339-349.**
> For fair comparision, we describe how to  implement these methods under the same backbone. Additionally, in Appendix D.4, lines 1500-1502, we note that the results in Table 8 are derived from the original TIGER paper.

---

> ### Author Response · Authors · 2024-11-23
> **Response to Reviewer SvZm - Part (3/5)**
>
> > **Comment 5:  Ablation Study on Timesteps** 一一  "For generative models, timestamps is crucial. The authors should conduct an ablation study on timestamps to evaluate their impact on the generation of positive and negative samples. "
>
> Thank you very much for your feedback. As you mentioned, timesteps are indeed an important hyperparameter for diffusion models. To ensure fairness in our previous experiments, we used the same timestep setting of 2000 for all diffusion-based recommenders, which is a commonly used configuration in diffusion models. Your suggestion has inspired us to explore the impact of timesteps on our method. We **conducted new experiments** on three datasets (Sports, Beauty and Toys), and the results are as follows:
>
>
>
> **Table 2: Comparison of Original TIGIR and Our Reproduced Version**
>
>
> | PreferDiff (Recall@5/NDCG@5) | 10           | 20           | 50           | 100          | 200          | 500          | 1000         | 2000         |
> |------------------------------|--------------|--------------|--------------|--------------|--------------|--------------|--------------|--------------|
> | **Sports**                  | 0.0025/0.0023 | 0.0028/0.0025 | 0.0028/0.0023 | 0.0042/0.0033 | 0.0087/0.0073 | 0.0146/0.0120 | 0.0174/0.0135 | **0.0185/0.0147** |
> | **Beauty**                  | 0.0098/0.0073 | 0.0103/0.0075 | 0.0094/0.0075 | 0.0134/0.0105 | 0.0264/0.0224 | 0.0353/0.0287 | 0.0379/0.0289 | **0.0429/0.0323** |
> | **Toys**                    | 0.0139/0.0128 | 0.0169/0.0154 | 0.0149/0.0134 | 0.0273/0.0223 | 0.0397/0.0300 | 0.0438/0.0335 | 0.0433/0.0324 | **0.0473/0.0367** |
>
> **Results**. We can observe that the diffusion timestep significantly affects the performance of PreferDiff. Typically, a timestep of 2000 achieves more stable and better results. This aligns with the observations in the original DDPM paper, which states that with 2000 timesteps, the noised embedding becomes pure noise, closely matching the Gaussian distribution assumption.

---

> ### Author Response · Authors · 2024-11-23
> **Response to Reviewer SvZm - Part (4/5)**
>
> > **Comment 6:  Inference Time Concerns** 一一  "Additionally, given that generative models are typically time-consuming, it would be beneficial to assess how PreferDiff performs in terms of inference speed compared to other sequence models. Specifically, can it achieve a favorable trade-off between hit rate/NDCG and speed?"
>
> Thank you for your valuable questions. We completely understand your concerns about the efficiency of our proposed PreferDiff. To address your concerns more thoroughly, we firstly provide how to trade-off between hit rate/NDCG and speed.
>
> **For diffusion-based recommenders, the most time-consuming factor during the inference stage is the number of denoising steps.** The previous DM-based recommender, DreamRec, adopts the initial denoising method from DDPM, resulting in 2000 denoising steps, which requires a significant amount of time. In contrast, PreferDiff adopts DDIM, which enables skip-step sampling. Empirically, we find that a denoising step count of around 20 achieves a favorable trade-off between hit rate/NDCG and inference speed. We **report the new following table** to illustrate the relationship between the number of denoising steps and recommendation performance:
>
> **Table 3: Effect of Different Denoising Steps**
>
> | PreferDiff (Recall@5/NDCG@5) | 1 (<1s)         | 2 (<1s)         | 4 (1s)          | 10 (2s)         | 20 (3s)         | 100 (23s)       | 400 (47s)       | 1000 (120s)     | 2000 (180s)     |
> |------------------------------|-----------------|-----------------|-----------------|-----------------|-----------------|-----------------|-----------------|-----------------|-----------------|
> | **Sports**                       | 0.0162/0.0131   | 0.0165/0.0133   | 0.0177/0.0137   | 0.0183/0.0143   | 0.0188/0.0147   | 0.0185/0.0143   | 0.0185/0.0142   | 0.0185/0.0143   | 0.0185/0.0143   |
> | **Beauty**                       | 0.0384/0.0289   | 0.0398/0.0309   | 0.0402/0.0296   | 0.0408/0.0307   | 0.0429/0.0323   | 0.0429/0.0322   | 0.0429/0.0322   | 0.0429/0.0323   | 0.0429/0.0323   |
> | **Toys**                         | 0.0437/0.0340   | 0.0438/0.0341   | 0.0433/0.0342   | 0.0458/0.0345   | 0.0473/0.0367   | 0.0473/0.0367   | 0.0473/0.0367   | 0.0473/0.0367   | 0.0473/0.0367   |
>
> **Results**. We can observe that with around 20 denoising steps, we only need approximately 3 seconds to infer recommendations for a batch of 32 users. This achieves a very good trade-off between recommendation performance and inference efficiency, making PreferDiff highly practical for real-world applications where both speed and accuracy are crucial.

---

> ### Author Response · Authors · 2024-11-23
> **Response to Reviewer SvZm - Part (5/5)**
>
> **We also draw a figure to visualize the trade-off between recommendation performance and inference efficiency in Appendix F.4 of revision paper.**
>
> Furthermore, we also **make comparisons** of training time and inference time between PreferDiff and other baselines.
>
>
> **Table 4: Comparison of Training Time and Inference Times.**
>
> | Dataset | Model      | Training Time (s/epoch)/(s/total) | Inference Time (s/epoch) |
> |---------|------------|-----------------------------------|--------------------------|
> | Sports  | SASRec     | 2.67 / 35                        | 0.47                     |
> |         | Bert4Rec   | 7.87 / 79                        | 0.65                     |
> |         | TIGIR      | 11.42 / 1069                     | 24.14                    |
> |         | DreamRec   | 24.32 / 822                      | 356.43                   |
> |         | PreferDiff | 29.78 / 558                      | 6.11                     |
> | Beauty  | SASRec     | 1.05 / 36                        | 0.37                     |
> |         | Bert4Rec   | 3.66 / 80                        | 0.40                     |
> |         | TIGIR      | 5.41 / 1058                      | 10.19                    |
> |         | DreamRec   | 15 / 525                         | 297.06                   |
> |         | PreferDiff | 18 / 430                         | 3.80                     |
> | Toys    | SASRec     | 0.80 / 56                        | 0.22                     |
> |         | Bert4Rec   | 3.11 / 93                        | 0.23                     |
> |         | TIGIR      | 3.76 / 765                       | 4.21                     |
> |         | DreamRec   | 15.43 / 552                      | 309.45                   |
> |         | PreferDiff | 16.07 / 417                      | 3.29                     |
>
> **Results**. We can observe that
>
> $\bullet$ Thanks to our adoption of DDIM for skip-step sampling, requires less training time and significantly shorter inference time compared to DreamRec, another diffusion-based recommender.
>
> $\bullet$ Compared to traditional deep learning methods like SASRec and Bert4Rec, PreferDiff has longer training and inference times but achieves much better recommendation performance.
>
> $\bullet$ Furthermore, compared to recent generative recommendation methods, such as TIGIR, which rely on autoregressive models and use beam search  during inference, PreferDiff also demonstrates shorter training and inference times, highlighting its efficiency and practicality in real-world scenarios.
>
>
> Thank you for your valuable comments. Your insights have greatly contributed to strengthening our submission.
>
> **If there are any issues, please feel free to reply, and we will respond promptly.**

---

> ### Author Response · Authors · 2024-11-25
> **looking forward to your reply**
>
> Dear Reviewer SvZm,
>
> Thank you again for your constructive comments. We have carefully addressed your concerns by clarifying the evaluation settings of our proposed methods, resolving confusion regarding baseline results or ID Quantifier through experimental evidence, and incorporating an ablation study on timesteps. Additionally, we provide a detailed comparison of training and inference times between our methods and the baselines. Furthermore, we demonstrate that inference time can be further reduced by adjusting the number of denoising steps, offering a flexible trade-off between latency and performance. All these revisions have been incorporated into the manuscript.
>
> We look forward to further discussion with you and would greatly appreciate your positive feedback on our rebuttal.
>
> Best regards,
>
> The Authors

---

> ### Author Response · Authors · 2024-11-29
> **Kind Reminder**
>
> Dear Reviewer SvZm,
>
> As the deadline approaches, we sincerely hope to address your concerns and discuss the rebuttal with you further. If you have any questions, please feel free to ask directly!
>
> Best regards,
>
> Authors

---

> ### Author Response · Authors · 2024-12-02
> **Alignment of Proposed  $\mathcal{L} _ {\text{BPR-Diff}}$ with Score Function (1/2)**
>
> > **Comment 7: Explanation for $\mathcal{L} _ {\text{BPR-Diff}}$** 一一 "The explanation in Section 3.2 is somewhat unconvincing. L_{BPR-Diff} is essentially a ranking loss in metric learning."
>
>
> Thank you for your valuable questions. We fully understand your concerns. The proposed $\mathcal{L}_{\text{BPR-Diff}}$ is not merely a ranking loss but is also fundamentally aligned with the core principles of diffusion models.
>
> Here, we prove that the goal of our proposed tailored diffusion optimization objective $\mathcal{L} _ {\text{BPR-Diff}}$ for personalized rankings is deeply connected with recent well-known score-based diffusion models. Optimizing $\mathcal{L} _ {\text{BPR-Diff}}$ can more effectively learn the landscape of the score function through personalized ranking. As introduced in recent studies [1][3], score function is the key component which guide the Langevin dynamics sampling process of diffusion models. Thus, we can utilize the trained score function $\nabla _ {\mathbf{e} _ 0} p _ {\theta}(\mathbf{e} _ 0 \mid \mathbf{c})$ to sample higher quality item embeddings with high ratings via Langevin dynamics [1][2], given certain user historical conditions.
>
> **Step 1: From Ratings to Probability Distribution**
>
> $$
> \mathcal{L} _ {\text{BPR}} = -\mathbb{E} _ {(\mathbf{e} _ 0 ^ + , \mathbf{e} _ 0 ^ - , \mathbf{c})}\left[ \log \sigma \left( f _ {\theta}(\mathbf{e} _ 0 ^ + \mid \mathbf{c}) -  f _ {\theta}(\mathbf{e} _ 0 ^ - \mid \mathbf{c}) \right) \right] \,
> $$
>
> The primary objective is to maximize the rating margin between positive items and the sampled negative items, where $f(\cdot)$ is a rating function that indicates how much the user likes the item given the historical interaction sequence. Here, we employ softmax normalization to transform the rating ranking into a log-likelihood ranking.
>
> We begin by expressing the rating $f _ {\theta}(\mathbf{e} _ 0 \mid \mathbf{c})$ in terms of the probability distribution $p _ {\theta}(\mathbf{e} _ 0 \mid \mathbf{c})$. This relationship is established through the following equations:
>
> $$
> p _ {\theta}(\mathbf{e} _ 0 \mid \mathbf{c}) = \frac{\exp(f _ {\theta}(\mathbf{e} _ 0 \mid \mathbf{c}))}{Z _ {\theta}} \,
> $$
> $$
> \log p _ {\theta}(\mathbf{e} _ 0 \mid \mathbf{c}) = f _ {\theta}(\mathbf{e} _ 0 \mid \mathbf{c}) - \log Z _ {\theta} \,
> $$
> $$
> f _ {\theta}(\mathbf{e} _ 0 \mid \mathbf{c}) = \log p _ {\theta}(\mathbf{e} _ 0 \mid \mathbf{c}) + \log Z _ {\theta} \,.
> $$
>
> Substituting the above equations into the BPR loss, we get:
>
> $$
> \mathcal{L} _ {\text{BPR-Diff}} = -\mathbb{E} _ {(\mathbf{e} _ 0 ^ + , \mathbf{e} _ 0 ^ - , \mathbf{c})}\left[ \log \sigma \left( \log p _ {\theta}(\mathbf{e} _ 0 ^ + \mid \mathbf{c}) - \log p _ {\theta}(\mathbf{e} _ 0 ^ - \mid \mathbf{c}) \right) \right] \.
> $$
>
> ---
>
> **Step 2: Connecting the Rating Function to the Score Function**
>
> The relationship between the rating function $f _ {\theta}(\mathbf{e} _ 0 \mid \mathbf{c})$ and the score function is given by the following derivation:
>
> Starting from:
> $$
> f _ {\theta}(\mathbf{e} _ 0 \mid \mathbf{c}) = \log p _ {\theta}(\mathbf{e} _ 0 \mid \mathbf{c}) + \log Z _ {\theta} \,
> $$
> where $Z _ {\theta}$ is the partition function:
> $$
> Z _ {\theta} = \int \exp(f _ {\theta}(\mathbf{e} \mid \mathbf{c})) \, d\mathbf{e} \.
> $$
>
> Taking the gradient of $f _ {\theta}(\mathbf{e} _ 0 \mid \mathbf{c})$ with respect to $\mathbf{e} _ 0$, we have:
> $$
> \nabla _ {\mathbf{e} _ 0} f _ {\theta}(\mathbf{e} _ 0 \mid \mathbf{c}) = \nabla _ {\mathbf{e} _ 0} \log p _ {\theta}(\mathbf{e} _ 0 \mid \mathbf{c}) + \nabla _ {\mathbf{e} _ 0} \log Z _ {\theta} \.
> $$
>
> Since $Z _ {\theta}$ does not depend on $\mathbf{e} _ 0$:
> $$
> \nabla _ {\mathbf{e} _ 0} \log Z _ {\theta} = 0 \.
> $$
>
> Thus:
> $$
> \nabla _ {\mathbf{e} _ 0} f _ {\theta}(\mathbf{e} _ 0 \mid \mathbf{c}) = \nabla _ {\mathbf{e} _ 0} \log p _ {\theta}(\mathbf{e} _ 0 \mid \mathbf{c}) \.
> $$
>
> In score-based models, the score function is defined as:
> $$
> \mathbf{s} _ {\theta}(\mathbf{e} _ 0, \mathbf{c}) \triangleq \nabla _ {\mathbf{e} _ 0} \log p _ {\theta}(\mathbf{e} _ 0 \mid \mathbf{c}) \.
> $$
>
> Thus, we have:
> $$
> \nabla _ {\mathbf{e} _ 0} f _ {\theta}(\mathbf{e} _ 0 \mid \mathbf{c}) = \mathbf{s} _ {\theta}(\mathbf{e} _ 0, \mathbf{c}) \.
> $$
>
> This equivalence connects the rating function and the score function, bridging the goal of recommendation systems and generative modeling in score-based diffusion models.

---

> ### Author Response · Authors · 2024-12-02
> **Alignment of Proposed  $\mathcal{L} _ {\text{BPR-Diff}}$ with Score Function (2/2)**
>
> **Step 3: Score Function Analysis for the Proposed $\mathcal{L} _ {\text{BPR-Diff}}$**
>
> The BPR-Diff loss is given by:
> $$
> \mathcal{L} _ {\text{BPR-Diff}} = -\mathbb{E} _ {(\mathbf{e} _ 0 ^ + , \mathbf{e} _ 0 ^ - , \mathbf{c})}\left[ \log \sigma \left( f _ {\theta}(\mathbf{e} _ 0 ^ + \mid \mathbf{c}) - f _ {\theta}(\mathbf{e} _ 0 ^ - \mid \mathbf{c}) \right) \right] \,.
> $$
>
> Taking the gradient with respect to $\mathbf{e} _ 0 ^ +$:
> $$
> \nabla _ {\mathbf{e} _ 0 ^ +} \mathcal{L} _ {\text{BPR-Diff}} = -\mathbb{E}\left[ \sigma(-s) \cdot \nabla _ {\mathbf{e} _ 0 ^ +} f _ {\theta}(\mathbf{e} _ 0 ^ + \mid \mathbf{c}) \right] \,,
> $$
> where$s = f _ {\theta}(\mathbf{e} _ 0 ^ + \mid \mathbf{c}) - f _ {\theta}(\mathbf{e} _ 0 ^ - \mid \mathbf{c})$.
>
> Similarly, for $\mathbf{e} _ 0 ^ - $:
> $$
> \nabla _ {\mathbf{e} _ 0 ^ -} \mathcal{L} _ {\text{BPR-Diff}} = \mathbb{E}\left[ \sigma(-s) \cdot \nabla _ {\mathbf{e} _ 0 ^ -} f _ {\theta}(\mathbf{e} _ 0 ^ - \mid \mathbf{c}) \right] \,.
> $$
>
> The gradients drive the optimization to:
> 1. Increasing the rating $f _ {\theta}(\mathbf{e} _ 0^+ \mid \mathbf{c})$ of the positive item by moving $\mathbf{e} _ 0^+$ in the direction of $\nabla _ {\mathbf{e} _ 0^+} f _ {\theta}$.
>
> 2. Decrease Decreasing the rating $f _ {\theta}(\mathbf{e} _ 0^- \mid \mathbf{c})$ of the negative item by moving $\mathbf{e} _ 0^-$ opposite to $\nabla _ {\mathbf{e} _ 0^-} f _ {\theta}$.
>
>  Therefore, optimizing $\mathcal{L} _ {\text{BPR-Diff}}$ can more effectively learn the landscape of the score function through personalized ranking. Thus, we can utilize $\nabla _ {\mathbf{e} _ 0} f _ {\theta}(\mathbf{e} _ 0 \mid \mathbf{c})$ to sample high quality item embeddings with high ratings via Langevin dynamics [1][2], given certain user historical conditions. In summary, our proposed loss function not only integrates user preferences into the training process of diffusion models but also ensures that item embeddings generated during inference better align with user preferences.
>
> [1] Song, Yang, and Stefano Ermon. "Improved techniques for training score-based generative models." Advances in neural information processing systems 33 (2020): 12438-12448.
>
>
> [2] Ho, Jonathan, Ajay Jain, and Pieter Abbeel. "Denoising diffusion probabilistic models." Advances in neural information processing systems 33 (2020): 6840-6851.
>
> [3] Song, Yang, et al. "Score-Based Generative Modeling through Stochastic Differential Equations." International Conference on Learning Representations.

---

> ### Author Response · Authors · 2024-12-03
> **Kind Reminder**
>
> Dear Reviewer SvZm,
>
> As today marks the final day of the discussion period, we sincerely hope to address your concerns thoroughly and engage in further discussion regarding our rebuttal. Should you have any questions or require additional clarification, please don’t hesitate to reach out.
> Moreover, if you find our responses satisfactory, we would greatly appreciate it if you could kindly consider the possibility of revising your score. Thank you once again for your valuable feedback and thoughtful suggestions.
>
> Best regards,
>
> Authors

---

### Official Review · Reviewer_5ztS · 2024-11-03

**Soundness:** 3
**Presentation:** 3
**Contribution:** 3
**Rating:** 6
**Confidence:** 4

**Summary:**

This paper presents PreferDiff, a novel optimization objective designed specifically for diffusion model (DM)-based recommenders in sequential recommendation tasks. PreferDiff aims to address the limitations of traditional objectives by incorporating a log-likelihood ranking approach, allowing the model to better capture user preferences through the integration of multiple negative samples. Extensive experiments demonstrate that PreferDiff outperforms baseline models in both performance and convergence speed, validating its effectiveness in modeling complex preference distributions.

**Strengths:**

- Innovative Objective Design: PreferDiff introduces a log-likelihood ranking objective tailored to diffusion models, marking a significant advancement in personalized ranking for DM-based recommenders.
- Comprehensive Negative Sampling: The incorporation of multiple negative samples enhances user preference understanding, leading to better separation between positive and negative items and improved recommendation performance.
-  Effective Performance and Convergence: PreferDiff demonstrates superior recommendation accuracy and faster convergence compared to existing models, showcasing both theoretical and practical value.

**Weaknesses:**

1.Limited Originality: The formulation of PreferDiff shows considerable overlap with Direct Preference Optimization (DPO), as several of its mathematical expressions and objective functions appear directly inspired or derived from DPO's original framework​. This raises concerns about the novelty of PreferDiff's contribution to preference learning within diffusion models, as the paper does not introduce substantial modifications or unique approaches that deviate meaningfully from DPO's foundational equations.


2.Dependency on High Embedding: PreferDiff’s performance is highly dependent on large embedding sizes, which may limit its scalability and increase computational costs.

**Questions:**

1. How does PreferDiff handle real-time recommendation scenarios where embedding dimensions need to be minimized due to latency constraints?
2. Could the authors provide more insights into the optimal range for the hyperparameter \lamda, especially in varied recommendation domains?

---

> ### Author Response · Authors · 2024-11-23
> **Response to Reviewer 5ztS - Part (1/3)**
>
> Thank you for your insightful review and for highlighting the strengths of our work, as well as your deep understanding of PreferDiff.
> We truly appreciate the thoughtful feedback and the two fundamental suggestions you raised.
> These points are essential and have prompted us to carefully address them, significantly improving the quality of our paper.
> Below are our detailed responses.
> We would be delighted to engage further with you if you have additional questions or feedback.
>
> > **Comment 1:  Clarification of Originality** 一一  "Limited Originality: The formulation of PreferDiff shows considerable overlap with Direct Preference Optimization (DPO), as several of its mathematical expressions and objective functions appear directly inspired or derived from DPO's original framework. This raises concerns about the novelty of PreferDiff's contribution to preference learning within diffusion models, as the paper does not introduce substantial modifications or unique approaches that deviate meaningfully from DPO's foundational equations."
>
> Thank you for your careful reading and thoughtful comments.
> We appreciate the opportunity to clarify the distinctions between PreferDiff and DPO.
> While PreferDiff’s formulation may overlap with DPO in certain aspects, this similarity arises because we reformulate the classical BPR loss as a log-likelihood ranking objective, which revealed a connection to DPO.
> To enhance readability and conceptual consistency, we referenced DPO’s formulation.
> However, PreferDiff is fundamentally different from DPO in several critical aspects:
> 1. **Formulation Differences**: First, PreferDiff incorporates dual objectives (cosine loss for generative modeling and preference loss for ranking), which is specially tailored to ranking tasks instead of preference learning.
> Second, unlike DPO [2] and Diffusion-DPO [3], PreferDiff incorporates multiple negative samples and proposes a theoretically guaranteed, making it well-suited for large-negative-sample scenarios in recommendation tasks.
> Third, unlike DPO or Diffusion-DPO, PreferDiff is implemented in an end-to-end manner without requiring a pre-trained reference model.
> 2.**Empirical Validation**: We conducted experiments comparing PreferDiff, DPO, and Diffusion-DPO across three datasets, varying
> $\beta$, a key DPO hyperparameter with values of 1, 5, and 10, and integrating it with DreamRec for a fair comparison.
> The results, summarized below, demonstrate PreferDiff’s superior performance in recommendation tasks:
>
>
> **Table 1: Comparsion with DPO and Diffusion-DPO**
>
> | Metric (Recall@5/NCDG@5）                                 | Sports | Beauty| Toys |
> |--------------------------------------|--------------------------|--------------------------|-------------------------|
> | **DreamRec + DPO (beta=1)**              | 0.0031 / 0.0015          | 0.0067 / 0.0053          | 0.0030 / 0.0022         |
> | **DreamRec + DPO (beta=5)**              | 0.0036 / 0.0026          | 0.0053 / 0.0034          | 0.0036 / 0.0023         |
> | **DreamRec + DPO (beta=10)**             | 0.0019 / 0.0011          | 0.0075 / 0.0056          | 0.0046 / 0.0034         |
> | **DreamRec + Diffusion-DPO (beta=1)**    | 0.0129 / 0.0101          | 0.0308 / 0.0244          | 0.0324 / 0.0261         |
> | **DreamRec + Diffusion-DPO (beta=5)**    | 0.0132 / 0.0113          | 0.0321 / 0.0251          | 0.0340 / 0.0272         |
> | **DreamRec + Diffusion-DPO (beta=10)**   | 0.0133 / 0.0115          | 0.0281 / 0.0223          | 0.0345 / 0.0281         |
> | **PreferDiff**                           | **0.0185 / 0.0147**          | **0.0429 / 0.0323**          | **0.0473 / 0.0367**         |
>
> **Results**. We can observe that PreferDiff outperforms DPO and Diffusion-DPO by a large margin across all three datasets. This further validates the effectiveness of our proposed PreferDiff, demonstrating that it is specifically tailored to model user preferences in diffusion-based recommenders. We hope this clarification, along with the supporting evidence, addresses your concerns.
>
> [1] Rendle, Steffen, et al. "BPR: Bayesian personalized ranking from implicit feedback." Proceedings of the Twenty-Fifth Conference on Uncertainty in Artificial Intelligence. 2009.
>
> [2] Rafailov, Rafael, et al. "Direct preference optimization: Your language model is secretly a reward model." Advances in Neural Information Processing Systems 36 (2024).
>
> [3] Wallace, Bram, et al. "Diffusion model alignment using direct preference optimization." Proceedings of the IEEE/CVF Conference on Computer Vision and Pattern Recognition. 2024.

---

> ### Author Response · Authors · 2024-11-23
> **Response to Reviewer 5ztS - Part (2/3)**
>
> > **Comment 2: Embedding Dimension Sensitivity** 一一 "Dependency on High Embedding: PreferDiff’s performance is highly dependent on large embedding sizes, which may limit its scalability and increase computational costs."
>
> Thank you for your valuable question.
> We fully understand and appreciate this concern, which we also identified as a limitation in our paper.
> Here, we want to firstly **clarify the cause of dimension sensitivity** and then **report a detailed computational efficiency comparison**.
>
> 1.**Clarification on the Cause**.
> The sensitivity to embedding size is not unique to PreferDiff but is inherently tied to the variance-preserving nature of DDPM and the learnable nature of embeddings in recommendation systems.
> This is a common challenge for current DM-based recommenders, and we are the first to investigate and highlight it.
> We believe the default use of small embedding sizes (e.g., 128) in prior research may have limited the exploration of DM-based recommenders.
> A detailed discussion and analysis on this is provided in Appendix F.3.
> 2. **Computational Efficiency**.
> While embedding size affects dimension sensitivity, our results show that it introduces only acceptable computational costs.
> Specifically, we measured the training and inference times for PreferDiff and several baselines on three datasets, as shown below:
>
>
> **Table 2: Comparison of Training Time and Inference Time**
>
> | Dataset | Model      | Training Time (s/epoch)/(s/total) | Inference Time (s/epoch) |
> |---------|------------|-----------------------------------|--------------------------|
> | Sports  | SASRec     | 2.67 / 35                        | 0.47                     |
> |         | Bert4Rec   | 7.87 / 79                        | 0.65                     |
> |         | TIGIR      | 11.42 / 1069                     | 24.14                    |
> |         | DreamRec   | 24.32 / 822                      | 356.43                   |
> |         | PreferDiff | 29.78 / 558                      | 6.11                     |
> | Beauty  | SASRec     | 1.05 / 36                        | 0.37                     |
> |         | Bert4Rec   | 3.66 / 80                        | 0.40                     |
> |         | TIGIR      | 5.41 / 1058                      | 10.19                    |
> |         | DreamRec   | 15 / 525                         | 297.06                   |
> |         | PreferDiff | 18 / 430                         | 3.80                     |
> | Toys    | SASRec     | 0.80 / 56                        | 0.22                     |
> |         | Bert4Rec   | 3.11 / 93                        | 0.23                     |
> |         | TIGIR      | 3.76 / 765                       | 4.21                     |
> |         | DreamRec   | 15.43 / 552                      | 309.45                   |
> |         | PreferDiff | 16.07 / 417                      | 3.29                     |
>
> We can observe that
>
> $\bullet$ Efficiency Over DreamRec: By adopting DDIM for skip-step sampling (20 denoising steps), PreferDiff achieves significantly shorter inference times compared to DreamRec, another diffusion-based recommender.
>
> $\bullet$ Performance Trade-offs: PreferDiff incurs slightly higher training and inference times than traditional methods like SASRec and Bert4Rec but delivers substantially better recommendation performance.
>
> $\bullet$ Competitive Practicality: Compared to recent generative methods like TIGIR, which rely on autoregressive models and beam search, PreferDiff demonstrates shorter training and inference times, making it efficient and practical for real-world applications.
>
> **This analysis, along with the time complexity details, has been included in Appendix F.4 of the revised paper.**

---

> ### Author Response · Authors · 2024-11-23
> **Response to Reviewer 5ztS - Part (3/3)**
>
> > **Comment 3:  Handling Real-Time Recommendations** 一一  "How does PreferDiff handle real-time recommendation scenarios where embedding dimensions need to be minimized due to latency constraints?"
>
> We speculate that your concern might be that increasing the embedding dimension could lead to longer inference times, potentially making PreferDiff unsuitable for real-time recommendation scenarios with latency constraints.
>
> However, as shown in the table above, our inference time is relatively short, demonstrating that PreferDiff is efficient even with large embedding dimensions. Additionally, we can **further reduce inference time by adjusting the number of denoising steps**, offering a flexible trade-off between latency and performance.
>
>
> **Table 3: Comparison of Inference Time and Recommendation Performance**
>
> | Datasets (Recall@5/NDCG@5) | Sports       | Beauty      | Toys      |
> |------------------------------------------|---------------------|--------------------|--------------------|
> | **SASRec (0.33s)**                                    | 0.0047 / 0.0036     | 0.0138 / 0.0090    | 0.0133 / 0.0097    |
> | **BERT4Rec (0.42s)**                                  | 0.0101 / 0.0060     | 0.0174 / 0.0112    | 0.0226 / 0.0139    |
> | **TIGER (12.85s)**                                     | 0.0093 / 0.0073     | 0.0236 / 0.0151    | 0.0185 / 0.0135    |
> | **DreamRec (320.98s)**                                  | 0.0155 / 0.0130     | 0.0406 / 0.0299    | 0.0440 / 0.0323    |
> | **PreferDiff (Denoising Step=1, 0.35s)**                                | 0.0162/0.0131 | 0.0384/0.0289| 0.0437/0.0340 |
> | **PreferDiff (Denoising Step=2, 0.43s)**                                | 0.0165/0.0133 | 0.0398/0.0309| 0.0438/0.0341  |
> | **PreferDiff (Denoising Step=4, 0.65s)**                                | 0.0177/0.0137  | 0.0402/0.0296| 0.0433/0.0342  |
> | **PreferDiff (Denoising Step=20, 3s)**                                | **0.0185 / 0.0147** | **0.0429 / 0.0323**| **0.0473 / 0.0367** |
>
> **Results**. We can observe that by adjusting the number of denoising steps, PreferDiff can ensure practicality for real-time recommendation tasks.
> This flexibility allows for a trade-off between inference speed and recommendation performance, making PreferDiff adaptable to various latency constraints while maintaining competitive effectiveness.
>
> We **added this discussion in Appendix F.5** of revision paper.
>
> > **Comment 4:  Optimal Range of Hyperparamters** 一一 "Could the authors provide more insights into the optimal range for the hyperparameter \lamda, especially in varied recommendation domains?"
>
>
> Thank you for the valuable question.
> According to Figure 4 in the paper, the optimal range for $\lambda$ is approximately between 0.4 and 0.6.
> Notably, the $\lambda$ value for the Sports dataset is the smallest, indicating that a larger proportion of learning generation is required.
> Interestingly, the Sports dataset also has the largest number of items.
> This observation may suggest that when the number of items is larger, learning generation might become increasingly important. **Therefore, we recommend setting a smaller $\lambda$ when dealing with recommendation domains with a large number of items to ensure a better balance between learning generation and learning preference.**
>
> **If there are any issues, please feel free to reply, and we will respond promptly.**

---

> ### Author Response · Authors · 2024-11-25
> **looking forward to your reply**
>
> Dear Reviewer 5ztS,
>
> Thank you again for your constructive comments. We have carefully addressed your concerns by clarifying the novelty of our approach and its connection to DPO through comprehensive analysis and experiments. Additionally, we provide a detailed comparison of training and inference times between our proposed methods and the baselines. Furthermore, we demonstrate that inference time can be further reduced by adjusting the number of denoising steps, offering a flexible trade-off between latency and performance. All these revisions have been incorporated into the manuscript.
>
> We look forward to further discussion with you and would greatly appreciate your positive feedback on our rebuttal.
>
> Best regards,
>
> The Authors

---

> ### Author Response · Authors · 2024-11-27
>
> Thank you for your prompt and positive feedback, as well as for appreciating the value of our work and raising the score. We are delighted to have addressed your questions and incorporated all your suggested modifications into the revised version based on your comments. If you have any further questions or concerns, please don’t hesitate to reach out.

---

### Official Review · Reviewer_E8Vd · 2024-11-04

**Soundness:** 3
**Presentation:** 3
**Contribution:** 2
**Rating:** 6
**Confidence:** 4

**Summary:**

This paper presents PreferDiff, an improved diffusion based recommendation model. The key contribution is the incorporation of an adapted  BPR objective and in turn jointly optimize the traditional MSE objective together with this new ranking loss. Variational method is employed to make the optimization tractable. The authors also show that the proposed approach is connected to DPO in formulation.
Experiments on Amazon review data set show that the proposed methods outperform a number of different baselines.

**Strengths:**

+ The intuition of incorporating ranking loss into the DM recommender makes sense. The development of the proposed method looks reasonable and (to my knowledge) correct.
+ The paper is clearly presented and easy to follow.
+ Experiments and ablation studies are mostly solid.

**Weaknesses:**

- There is essentially only 1 data set being used (amazon review), no matter how many categories you include, this data set may not be representative enough which may raise concerns regarding the generalizability of your findings
- Some of the questions remain unanswered (or observations without explanation) , e.g,: 1) what caused PreferDiff to be faster than DreamRec? 2) why diffusion models are more sensitive to d_model?
- Novelty seems to be minimum, the overall approach makes sense but is also straightforward. The connection to DPO is rather weak and the claim of this as a theoretical contribution (1 of the 3 contributions) is not very sound.

**Questions:**

1. Consider use a few other data sets, especially data sets with diverse background (e.g, Yahoo!, Criteo)
2. At least some efforts should be made to explain unexpected observations, e.g, e.g,: 1) what caused PreferDiff to be faster than DreamRec? 2) why diffusion models are more sensitive to d_model?
3. The connection to DPO is rather weak and the claim of this as a theoretical contribution (1 of the 3 contributions) is not very sound.
4. Eq(12) added the MSE loss back to the formula, the authors claimed that this is to mitigate the learning stability issues, it would be interesting to the readers if the authors could report that instability observations directly. It would also be worthy looking into this instability issue to root-cause it. Since PreferDiff converges faster, would this indicate that ranking loss itself might be more stable than the MSE loss and could it be possible that there are other ways to mitigate the instability issue without taking the hybrid path?

---

> ### Author Response · Authors · 2024-11-23
> **Response to Reviewer E8Vd - Part (1/3)**
>
> Thank you for your insightful review and for highlighting the strengths of our work, as well as your deep understanding of PreferDiff. We truly appreciate the thoughtful feedback and the suggestions you raised. These points are essential and have prompted us to carefully address them, significantly improving the quality of our paper.
>
> > **Comment 1: Diverse Datasets** 一一 "There is essentially only 1 data set being used (amazon review), no matter how many categories you include, this data set may not be representative enough which may raise concerns regarding the generalizability of your findings." "Consider use a few other data sets, especially data sets with diverse background (e.g, Yahoo!, Criteo)"
>
>
> Thank you for your valuable suggestion.
> We agree that validating our approach on datasets with diverse backgrounds strengthens the paper's credibility.
> In response, we incorporated **three new datasets: Yahoo! R1 (Music), Steam (Game), and ML-1M (Movie)**. These datasets cover diverse domains, aligning with your suggestion.
> However, as Criteo is more suited for CTR tasks rather than sequential recommendation, we decided not to include it.
>
> To maintain consistency, we applied the same data preprocessing and evaluation protocols outlined in our paper.
> For Yahoo! R1, due to its large size (over one million users), we conducted experiments on a randomly sampled subset of 50,000 users during the rebuttal period.
> The full-scale results will be included in the final revision.
> The experimental results are as follows:
>
>
> **Table 1: Additional experiments on three new Dataset**
>
> | Datasets (Background) (Recall@5/NDCG@5) | Yahoo (Music)       | Steam (Game)       | ML-1M (Movie)      |
> |------------------------------------------|---------------------|--------------------|--------------------|
> | **GRU4Rec**                                   | 0.0548 / 0.0491     | 0.0379 / 0.0325    | 0.0099 / 0.0089    |
> | **SASRec**                                    | 0.0996 / 0.0743     | 0.0695 / 0.0635    | 0.0132 / 0.0102    |
> | **Bert4Rec**                                  | 0.1028 / 0.0840     | 0.0702 / 0.0643    | 0.0215 / 0.0152    |
> | **TIGIR**                                     | 0.1128 / 0.0928                  | 0.0603 / 0.0401    | 0.0430 / 0.0272    |
> | **DreamRec**                                  | 0.1302 / 0.1025     | 0.0778 / 0.0572    | 0.0464 / 0.0314    |
> |**PreferDiff**                                | **0.1408 / 0.1106**     | **0.0814 / 0.0680**    | **0.0629 / 0.0439**    |
>
> Consistent with our findings, these results confirm that PreferDiff outperforms baselines across datasets from diverse domains, demonstrating its strong generalizability.
> We have **added this discussion to Appendix F.1** of the revised paper.

---

> ### Author Response · Authors · 2024-11-23
> **Response to Reviewer E8Vd - Part (2/3)**
>
> > **Comment 2: Observations without explanation** 一一 "why diffusion models are more sensitive to d_model?"
>
> Thank you for raising this insightful question. Understanding why diffusion-based recommenders, such as DreamRec and PreferDiff, are more sensitive to embedding dimensionality is indeed important for advancing this line of research.
>
> We hypothesize that this sensitivity stems from the **inherent variance-preserving nature** of (DDPM) [1,2].
> In recommendation, the forward process formula for item embeddings $\mathbf{E} \in \mathbb{R}^{N \times d}$ is:
>
> $\mathbf{E}_{0}^{t}=\sqrt{\alpha_t}\mathbf{E}_0+\sqrt{1-\alpha_t}\epsilon$,
>
> where $N$ represents the total number of items, $\alpha_t$ the noise level, and $\epsilon$ is a standard Gaussian noise.
> The variance on both sides is:
>
> $\text{Var}(\mathbf{E}_{0}^{t})=\alpha_t\text{Var}(\mathbf{E}_0)+(1-\alpha_t)\mathbf{I}$.
>
> For diffusion-based recommenders, the item embeddings' covariance matrix $\text{Var}(\mathbf{E}_0)$ must approach an identity-like structure to ensure effective variance distribution.
> This is straightforward for fixed data like images or text, but in recommendation, item embeddings are dynamically updated during training.
> High-dimensional embeddings are thus critical to capture the diversity of items.
>
> We also **empirically observed that initializing item embeddings with a standard normal distribution is key** to the success of DreamRec and PreferDiff, as shown below:
>
> **Table 2: Effect of Different Initilization Methods**
>
> | PreferDiff (Recall@5/NDCG@5) | Uniform           | Kaiming_Uniform    | Kaiming_Normal     | Xavier_Uniform     | Xavier_Normal      | Standard Normal     |
> |------------------------------|-------------------|--------------------|--------------------|--------------------|--------------------|---------------------|
> | **Sports**                       | 0.0039/0.0026     | 0.0025/0.0019      | 0.0023/0.0021      | 0.0011/0.0007      | 0.0014/0.0007      | **0.0185/0.0147**       |
> | **Beauty**                       | 0.0013/0.0037     | 0.0040/0.0027      | 0.0049/0.0028      | 0.0036/0.0021      | 0.0067/0.0037      | **0.0429/0.0323**       |
> | **Toys**                         | 0.0015/0.0011     | 0.0051/0.0028      | 0.0041/0.0029      | 0.0051/0.0029      | 0.0042/0.0023      | **0.0473/0.0367**       |
>
> Furthermore, we calculated the covariance matrices of the final item embeddings.
> Interestingly, for diffusion-based recommenders, the covariance matrices were nearly identity matrices, while models like SASRec did not exhibit this property.
> This suggests that current diffusion-based recommenders distribute variance evenly across dimensions, requiring higher dimensions to represent item complexity effectively.
> Please refer to Appendix F.3 for details.
> All these experimental results validate our hypothesis that inherent nature of Variance-Preserving is main reason behind for dimension sensitivity for current diffusion-based recommenders.
>
> We also explored dimensionality reduction (e.g., Projection Layers) and regularization techniques (e.g., enforcing the item embedding covariance matrix to be an identity matrix) but found that these approaches significantly reduced performance.
> We guess that Variance Exploding (VE) diffusion models [2], which allow variance to grow over time, may mitigate this sensitivity and provide a promising research direction worth further exploration.
>
> We have **incorporated this discussion, along with all supporting evidence**, in Appendix F.3 of the revised paper. Thank you again for your valuable comment, which has deepened the insights presented in our work.
>
> [1] Ho, Jonathan, Ajay Jain, and Pieter Abbeel. "Denoising diffusion probabilistic models." Advances in neural information processing systems 33 (2020): 6840-6851.
>
> [2] Song, Yang, et al. "Score-Based Generative Modeling through Stochastic Differential Equations." International Conference on Learning Representations.
>
> > **Comment 3: Convergence Speed** "Some of the questions remain unanswered (or observations without explanation) , e.g,: 1) what caused PreferDiff to be faster than DreamRec?"  "At least some efforts should be made to explain unexpected observations, e.g, e.g,: 1) what caused PreferDiff to be faster than DreamRec?"
>
>
> In Section 3.2, lines 260-271, we analyze the property of the proposed $\mathcal{L} _ {\text{BPR-Diff}}$, demonstrating its capability to handle hard negative samples. Specifically, if the diffusion model  incorrectly assigns higher likelihood to negative items than positive items for certain users' historical sequences, the gradient weight $w_\theta$ of $\mathcal{L} _ {\text{BPR-Diff}}$ will increase.  This means that during backpropagation, the model receives larger update steps to correct the misclassification of hard negative samples. Therefore, $\mathcal{L} _ {\text{BPR-Diff}}$ can adaptively increase the learning emphasis on difficult samples, promoting faster convergence of the model.

---

> ### Author Response · Authors · 2024-11-23
> **Response to Reviewer E8Vd - Part (3/3)**
>
> > **Comment 4: Connection with DPO**一一 "The connection to DPO is rather weak and the claim of this as a theoretical contribution (1 of the 3 contributions) is not very sound." "Novelty seems to be minimum, the overall approach makes sense but is also straightforward."
>
> Thanks for your thoughtful feedback regarding the novelty of our approach and connection to DPO.
> We appreciate the opportunity to clarify these points and refine our submission.
>
> Our primary contribution lies in designing a diffusion optimization objective tailored for modeling user preference distributions in personalized ranking.
> Instead of positioning a theoretical contribution, our intent is to explore PrefDiff's properties from three perspectives:
> 1. Modeling user behavior distribution by integrating both positive and multiple negative items.
> 2. Hard negative mining through gradient analysis.
> 3. Enhancing generative abilities by connecting to DPO.
>
> We acknowledge that the statement, “we theoretically prove that PreferDiff is equivalent to Direct Preference Optimization (DPO) under certain conditions,” may have been overstated and potentially misleading.
> Based on your suggestion, we **have revised this claim** to:
> “from a preference learning perspective, we find that PreferDiff is connected to Direct Preference Optimization~\citep{DPO} under certain conditions, indicating its potential to align user preferences through generative modeling in diffusion-based recommenders.”
> This revision more accurately reflects our intent to highlight the connection to DPO as a means of validating PreferDiff’s rationality, rather than claiming novelty solely through this association.
> We hope these revisions address your concerns and improve the clarity of our submission.
>
>
>
>
> **Comment 5: Hybird Loss**一一  "Eq(12) added the MSE loss back to the formula, the authors claimed that this is to mitigate the learning stability issues, it would be interesting to the readers if the authors could report that instability observations directly. It would also be worthy looking into this instability issue to root-cause it. Since PreferDiff converges faster, would this indicate that ranking loss itself might be more stable than the MSE loss and could it be possible that there are other ways to mitigate the instability issue without taking the hybrid path?"
>
>
> Thanks for your valuable question.
> In Eq 12: $\mathcal{L} _ {\text{PerferDiff}}= \underbrace{\lambda \mathcal{L} _ {\text{Simple}}}_{\text{Learning Generation}} + \underbrace{(1-\lambda) \mathcal{L} _ {\text{BPR-Diff-C}}} _ {\text{Learning Preference}}$.
>
> The first term can be seen as learning generation, and the second term can be seen as learning preference. Notably, like other recommenders, the input of PreferDiff is the learnable item embedding, which will vary during the training stage. This means that exclusively learning the preference can result in unstable training because errors introduced in the early stage may accumulate over time.
>
>
> **Table 3: Effect of Different $\lambda$**
>
> | Model                     | Recall@5/NDCG@5 (Sports) | Recall@5/NDCG@5 (Beauty) | Recall@5/NDCG@5 (Toys) |
> |---------------------------|--------------------------|--------------------------|-------------------------|
> | PreferDiff ($\lambda=0$)          | 0.0129 / 0.0101         | 0.0308 / 0.0244         | 0.0324 / 0.0261         |
> | PreferDiff                | 0.0185 / 0.0147         | 0.0429 / 0.0323         | 0.0473 / 0.0367         |
>
> We can observe that, empirically, when $\lambda=0$, meaning only the ranking loss is used, the performance drops significantly. This finding partially validates our hypothesis that balancing between learning generation and learning preference is crucial for achieving optimal recommendation performance in PreferDiff.
>
> We also report the recommendation performance of PreferDiff with different values of $\lambda$ in Figure 3, where a larger $\lambda$ indicates a greater emphasis on learning preference. You can observe that when $\lambda$ is very large, the performance degrades, reflecting the issue we discussed earlier. Conversely, when $\lambda$ is too small, the performance also declines, highlighting that learning preference is crucial for the recommendation performance of PreferDiff.
> These results demonstrate the importance of carefully tuning $\lambda$ to achieve a balanced trade-off for optimal performance.
>
> **If there are any issues, please feel free to reply, and we will respond promptly.**

---

> ### Author Response · Authors · 2024-11-25
> **looking forward to your reply**
>
> Dear Reviewer E8Vd,
>
> Thank you again for your constructive comments. We have carefully addressed your concerns by adding new datasets (Yahoo!R1, Steam, and ML-1M), providing theoretical and experimental evidence on the sensitivity of diffusion models to embedding dimensions, and clarifying the novelty of our approach and its connection to DPO. All these revisions have been incorporated into the manuscript.
>
> We look forward to further discussion with you and would greatly appreciate your positive feedback on our rebuttal.
>
> Best regards,
>
> The Authors

---

> ### Author Response · Authors · 2024-11-29
> **Kind Reminder**
>
> Dear Reviewer E8Vd,
>
> As the deadline approaches, we sincerely hope to address your concerns and discuss the rebuttal with you further. If you have any questions, please feel free to ask directly!
>
> Best regards,
>
> Authors

---

> ### Author Response · Authors · 2024-12-02
> **More Theoretical Justification (1 / 2)**
>
> Dear Reviewer E8Vd,
>
> Here, we aim to provide a more detailed theoretical justification for why our proposed method works and its implications for both recommendation systems and diffusion models.
>
> we prove that the goal of our proposed tailored diffusion optimization objective $\mathcal{L} _ {\text{BPR-Diff}}$ for personalized rankings is deeply connected with recent well-known score-based diffusion models. Optimizing $\mathcal{L} _ {\text{BPR-Diff}}$ can more effectively learn the landscape of the score function through personalized ranking. As introduced in recent studies [1][3], score function is the key component which guide the Langevin dynamics sampling process of diffusion models. Thus, we can utilize the trained score function $\nabla _ {\mathbf{e} _ 0} p _ {\theta}(\mathbf{e} _ 0 \mid \mathbf{c})$ to sample higher quality item embeddings with high ratings via Langevin dynamics [1][2], given certain user historical conditions.
>
> ---
>
>
> **Step 1: From Ratings to Probability Distribution**
>
> $$
> \mathcal{L} _ {\text{BPR}} = -\mathbb{E} _ {(\mathbf{e} _ 0 ^ + , \mathbf{e} _ 0 ^ - , \mathbf{c})}\left[ \log \sigma \left( f _ {\theta}(\mathbf{e} _ 0 ^ + \mid \mathbf{c}) -  f _ {\theta}(\mathbf{e} _ 0 ^ - \mid \mathbf{c}) \right) \right] \,
> $$
>
> The primary objective is to maximize the rating margin between positive items and the sampled negative items, where $f(\cdot)$ is a rating function that indicates how much the user likes the item given the historical interaction sequence. Here, we employ softmax normalization to transform the rating ranking into a log-likelihood ranking.
>
> We begin by expressing the rating $f _ {\theta}(\mathbf{e} _ 0 \mid \mathbf{c})$ in terms of the probability distribution $p _ {\theta}(\mathbf{e} _ 0 \mid \mathbf{c})$. This relationship is established through the following equations:
>
> $$
> p _ {\theta}(\mathbf{e} _ 0 \mid \mathbf{c}) = \frac{\exp(f _ {\theta}(\mathbf{e} _ 0 \mid \mathbf{c}))}{Z _ {\theta}} \,
> $$
> $$
> \log p _ {\theta}(\mathbf{e} _ 0 \mid \mathbf{c}) = f _ {\theta}(\mathbf{e} _ 0 \mid \mathbf{c}) - \log Z _ {\theta} \,
> $$
> $$
> f _ {\theta}(\mathbf{e} _ 0 \mid \mathbf{c}) = \log p _ {\theta}(\mathbf{e} _ 0 \mid \mathbf{c}) + \log Z _ {\theta} \,.
> $$
>
> Substituting the above equations into the BPR loss, we get:
>
> $$
> \mathcal{L} _ {\text{BPR-Diff}} = -\mathbb{E} _ {(\mathbf{e} _ 0 ^ + , \mathbf{e} _ 0 ^ - , \mathbf{c})}\left[ \log \sigma \left( \log p _ {\theta}(\mathbf{e} _ 0 ^ + \mid \mathbf{c}) - \log p _ {\theta}(\mathbf{e} _ 0 ^ - \mid \mathbf{c}) \right) \right] \.
> $$
>
> ---
>
> **Step 2: Connecting the Rating Function to the Score Function**
>
> The relationship between the rating function $f _ {\theta}(\mathbf{e} _ 0 \mid \mathbf{c})$ and the score function is given by the following derivation:
>
> Starting from:
> $$
> f _ {\theta}(\mathbf{e} _ 0 \mid \mathbf{c}) = \log p _ {\theta}(\mathbf{e} _ 0 \mid \mathbf{c}) + \log Z _ {\theta} \,
> $$
> where $Z _ {\theta}$ is the partition function:
> $$
> Z _ {\theta} = \int \exp(f _ {\theta}(\mathbf{e} \mid \mathbf{c})) \, d\mathbf{e} \.
> $$
>
> Taking the gradient of $f _ {\theta}(\mathbf{e} _ 0 \mid \mathbf{c})$ with respect to $\mathbf{e} _ 0$, we have:
> $$
> \nabla _ {\mathbf{e} _ 0} f _ {\theta}(\mathbf{e} _ 0 \mid \mathbf{c}) = \nabla _ {\mathbf{e} _ 0} \log p _ {\theta}(\mathbf{e} _ 0 \mid \mathbf{c}) + \nabla _ {\mathbf{e} _ 0} \log Z _ {\theta} \.
> $$
>
> Since $Z _ {\theta}$ does not depend on $\mathbf{e} _ 0$:
> $$
> \nabla _ {\mathbf{e} _ 0} \log Z _ {\theta} = 0 \.
> $$
>
> Thus:
> $$
> \nabla _ {\mathbf{e} _ 0} f _ {\theta}(\mathbf{e} _ 0 \mid \mathbf{c}) = \nabla _ {\mathbf{e} _ 0} \log p _ {\theta}(\mathbf{e} _ 0 \mid \mathbf{c}) \.
> $$
>
> In score-based models, the score function is defined as:
> $$
> \mathbf{s} _ {\theta}(\mathbf{e} _ 0, \mathbf{c}) \triangleq \nabla _ {\mathbf{e} _ 0} \log p _ {\theta}(\mathbf{e} _ 0 \mid \mathbf{c}) \.
> $$
>
> Thus, we have:
> $$
> \nabla _ {\mathbf{e} _ 0} f _ {\theta}(\mathbf{e} _ 0 \mid \mathbf{c}) = \mathbf{s} _ {\theta}(\mathbf{e} _ 0, \mathbf{c}) \.
> $$
>
> This equivalence connects the rating function and the score function, bridging the goal of recommendation systems and generative modeling in score-based diffusion models.

---

> ### Author Response · Authors · 2024-12-02
> **More Theoretical Justification (2/2)**
>
> **Step 3: Score Function Analysis for the Proposed $\mathcal{L} _ {\text{BPR-Diff}}$**
>
> The BPR-Diff loss is given by:
> $$
> \mathcal{L} _ {\text{BPR-Diff}} = -\mathbb{E} _ {(\mathbf{e} _ 0 ^ + , \mathbf{e} _ 0 ^ - , \mathbf{c})}\left[ \log \sigma \left( f _ {\theta}(\mathbf{e} _ 0 ^ + \mid \mathbf{c}) - f _ {\theta}(\mathbf{e} _ 0 ^ - \mid \mathbf{c}) \right) \right] \,.
> $$
>
> Taking the gradient with respect to $\mathbf{e} _ 0 ^ +$:
> $$
> \nabla _ {\mathbf{e} _ 0 ^ +} \mathcal{L} _ {\text{BPR-Diff}} = -\mathbb{E}\left[ \sigma(-s) \cdot \nabla _ {\mathbf{e} _ 0 ^ +} f _ {\theta}(\mathbf{e} _ 0 ^ + \mid \mathbf{c}) \right] \,,
> $$
> where$s = f _ {\theta}(\mathbf{e} _ 0 ^ + \mid \mathbf{c}) - f _ {\theta}(\mathbf{e} _ 0 ^ - \mid \mathbf{c})$.
>
> Similarly, for $\mathbf{e} _ 0 ^ - $:
> $$
> \nabla _ {\mathbf{e} _ 0 ^ -} \mathcal{L} _ {\text{BPR-Diff}} = \mathbb{E}\left[ \sigma(-s) \cdot \nabla _ {\mathbf{e} _ 0 ^ -} f _ {\theta}(\mathbf{e} _ 0 ^ - \mid \mathbf{c}) \right] \,.
> $$
>
> The gradients drive the optimization to:
> 1. Increasing the rating $f _ {\theta}(\mathbf{e} _ 0^+ \mid \mathbf{c})$ of the positive item by moving $\mathbf{e} _ 0^+$ in the direction of $\nabla _ {\mathbf{e} _ 0^+} f _ {\theta}$.
>
> 2. Decrease Decreasing the rating $f _ {\theta}(\mathbf{e} _ 0^- \mid \mathbf{c})$ of the negative item by moving $\mathbf{e} _ 0^-$ opposite to $\nabla _ {\mathbf{e} _ 0^-} f _ {\theta}$.
>
>  Therefore, optimizing $\mathcal{L} _ {\text{BPR-Diff}}$ can more effectively learn the landscape of the score function through personalized ranking. Thus, we can utilize $\nabla _ {\mathbf{e} _ 0} f _ {\theta}(\mathbf{e} _ 0 \mid \mathbf{c})$ to sample high quality item embeddings with high ratings via Langevin dynamics [1][2], given certain user historical conditions. In summary, our proposed loss function not only integrates user preferences into the training process of diffusion models but also ensures that item embeddings generated during inference better align with user preferences.
>
> [1] Song, Yang, and Stefano Ermon. "Improved techniques for training score-based generative models." Advances in neural information processing systems 33 (2020): 12438-12448.
>
>
> [2] Ho, Jonathan, Ajay Jain, and Pieter Abbeel. "Denoising diffusion probabilistic models." Advances in neural information processing systems 33 (2020): 6840-6851.
>
> [3] Song, Yang, et al. "Score-Based Generative Modeling through Stochastic Differential Equations." International Conference on Learning Representations.

---

> ### Author Response · Authors · 2024-12-03
> **Kind Reminder**
>
> Dear Reviewer E8Vd,
>
> As today marks the final day of the discussion period, we sincerely hope to address your concerns thoroughly and engage in further discussion regarding our rebuttal. Should you have any questions or require additional clarification, please don’t hesitate to reach out.
> Moreover, if you find our responses satisfactory, we would greatly appreciate it if you could kindly consider the possibility of revising your score. Thank you once again for your valuable feedback and thoughtful suggestions.
>
> Best regards,
>
> Authors

---

### Meta-Review · Area_Chair_t7uV · 2024-12-22

**Metareview:**

This paper, based on diffusion models, proposes preference diffusion for recommendation. Reviewers considered this a borderline paper, with scores of 6 or 5. The main concerns raised were: 1) The experimental datasets and baselines were insufficient, and there was a lack of efficiency analysis experiments; 2) The novelty of the model and the differences from existing methods were not clearly articulated. During the rebuttal phase, the authors responded to the reviewers' questions in detail, but no further feedback from reviewers was provided. After reviewing the authors' responses and the paper, I think this paper can be accepted as a poster.

**Additional Comments On Reviewer Discussion:**

During the rebuttal phase, the authors responded to the reviewers' questions in detail, but no further feedback from reviewers was provided.

---

### Decision · Program_Chairs · 2025-01-22

Accept (Poster)